# Clustering via Hedonic Games: New Concepts and Algorithms

**Gergely Csáji**
ELTE Centre of Economic and Regional Studies
Budapest, Hungary
`csaji.gergely@krtk.elte.hu`

**Alexander Gundert**
Heinrich-Heine-Universität Düsseldorf
Düsseldorf, Germany
`Alexander.Gundert@hhu.de`

**Jörg Rothe**
Heinrich-Heine-Universität Düsseldorf
Düsseldorf, Germany
`rothe@hhu.de`

**Ildikó Schlotter**
ELTE Centre of Economic and Regional Studies
Budapest, Hungary
`schlotter.ildiko@krtk.elte.hu`

## Abstract

We study fundamental connections between coalition formation games and clustering, illustrating the cross-disciplinary relevance of these concepts. We focus on graphical hedonic games where agents' preferences are compactly represented by a friendship graph and an enmity graph. In the context of clustering, friendship relations naturally align with data point similarities, whereas enmity corresponds to dissimilarities. We consider two stability notions based on single-agent deviations: local popularity and local stability. Exploring these concepts from an algorithmic viewpoint, we design efficient mechanisms for finding locally stable or locally popular partitions. Besides gaining theoretical insight into the computational complexity of these problems, we perform simulations that demonstrate how our algorithms can be successfully applied in clustering and community detection.

## 1 Introduction

Coalition formation and clustering are two research areas that capture different aspects of grouping a set of entities in a meaningful or optimal way. While coalition formation studies autonomous agents who have preferences over the various ways in which they can be partitioned into groups, clustering aims to unearth hidden structure and similarities within a set of datapoints. Despite the shared theme of group formation, there is no substantial interplay between these two research fields.

This study aims to investigate how different solution concepts emerging from coalition formation—in particular, from hedonic games—can be used in data clustering. Traditional clustering metrics measure the quality of the boundaries between the clusters or the average similarities within clusters. We propose a criterion that is based on a form of stability: we aim for a clustering where no deviation by a single agent (or data point) should lead to a more preferred outcome. We aim to investigate how such stability notions can be harnessed in clustering algorithms.

Similarities and dissimilarities, the baseline concepts used by all clustering algorithms, can be associated with relations of compatibility and of conflict between two agents. Leveraging this idea, we focus on a model in coalition formation due to Dimitrov et al. [16] where agents' preferences over different coalitions are determined by their friendships and their enmities, represented as two graphs. More precisely, we adopt the model of hedonic *friends–enemies–neutrals* (FEN) games [33, 37, 27] where agents classify all other agents as a friend, an enemy, or a neutral, and each agent evaluates a given partition based on the number of friends and the number of enemies in their coalition.

39th Conference on Neural Information Processing Systems (NeurIPS 2025).

Unfortunately, finding "optimal" or, in some sense, *stable* partitions often turns out to be a computationally intractable task. Since computational efficiency is often crucial in clustering applications, we focus our attention on *local* stability concepts requiring only that re-assigning a single agent to a different coalition does not yield a partition deemed better for the majority of agents. Such single-agent deviations were studied in FEN games by Kerkmann and Rothe [28] and, more recently, in friends–enemies (FE) games—a model excluding neutrality—by Brandt et al. [10] who showed that in a model where two partitions are compared by taking the majority vote of the agents, such deviations converge into a *locally stable* partition in a polynomial number of rounds. We extend these results to FEN games and also to a slightly more stringent stability concept, *local popularity*. As opposed to the FE games studied by Brandt et al. [10], the option of neutrality introduces a flexibility that allows focusing on relations between points that are either very similar, or very different. Furthermore, neutrality yields smaller degrees in the friendship and enmity graphs which, as we show, has a huge benefit in terms of the convergence speed. Depending on the application under focus, different emphasis might be given to grouping together similar data points versus separating dissimilar data points. In the context of a FEN game, agents may put a significantly larger weight on the objective of being grouped together with friends than being separated from enemies, or vice versa; a third, more balanced approach is when agents treat these two objectives with the same importance. These options give rise to three preference domains—friend-appreciating, enemy-averse, and balanced—whose study may facilitate the application of our results to different clustering domains.

By identifying "similarity" with "friendship" and "difference" with "enmity," we demonstrate via simulations how our algorithms for finding locally stable or locally popular partitions can be used in the context of data clustering. The stability property enforced by our algorithm is particularly useful in clustering tasks where agents have meaningful local interactions or preferences. This includes standard benchmarks such as citation networks (e.g., Cora [36]), email communication graphs (e.g., Enron [30]), and social networks, where misclassifying a node can distort community structure. By ensuring that no agent prefers to switch to a different cluster based on local relationships, our method yields clusters that are robust against local deviations. This distinguishes our approach from traditional methods like $k$-means or DBSCAN, which may prioritize global fit over local consistency. As our simulations attest, this interdisciplinary approach can be surprisingly successful when compared to the most prevalent clustering algorithms. Thus we hope our work can spark a stronger cooperation between these fields to explore the applicability of coalition formation games in clustering.

**Related work.** Although both hedonic games and clustering are widely studied, broad subjects, the only work connecting these two areas we are aware of is by Feldman et al. [20] who study hedonic clustering games in a noncooperative framework where clustering is obtained via decisions made by independent agents. They propose two types of hedonic games for which they characterize Nash equilibria and analyze the price of anarchy and the price of stability. Although some ideas appear both in our model and theirs—e.g., the utilities of agents depending on their similarities with agents within their own cluster and on dissimilarities with agents in other clusters—the models proposed by Feldman et al. significantly differ from ours. Most importantly, while our work is rooted in cooperative game theory, Feldman et al. take a noncooperative approach. In addition, although they establish the existence of Nash equilibria for certain specific domains, their results do not seem to lead to efficient, scalable algorithms for general clustering.

Regarding the broader literature related to our study, we only point out the most important precursors of our work and a few key references for more background on the relevant research areas. Coalition formation is a central topic in cooperative game theory. Starting with the seminal work of Drèeze and Greenberg [17], researchers' focus shifted to so-called hedonic games where agents' preferences depend only on the coalition they are contained in; see also [5, 8, 13] and the book chapters by Aziz and Savani [4] and Bullinger et al. [11, Section 3.6]. FE games were introduced by Dimitrov et al. [16] as a subclass of hedonic games with a concise description (namely, the agents' friendship relations). FEN games, the extension of their model that allows for neutrality, have also been studied [14, 27, 28, 33, 37, 40]. Single-agent deviations and their induced dynamics were investigated in FE games by Brandt et al. [10] who, in various settings, showed that such deviations converge to a partition that fulfills certain stability concepts. Among the stability concept they considered is what we call *local stability*; our work extends their results on this notion to FEN games. Our stability concepts stem from the idea of popularity as introduced by Gärdenfors [22]. Popularity has been widely studied in matching and allocation problems [1, 6, 15, 25, 26] and has also been considered in

the context of hedonic games by, e.g., Aziz et al. [3], Kerkmann et al. [27], Brandt and Bullinger [9], Kerkmann and Rothe [29], and Bullinger and Gilboa [12].

For a recent survey on clustering—a fundamental problem in unsupervised learning aimed at discovering structure in data by grouping similar elements—see [19]. Classical clustering methods such as $k$-means, introduced by MacQueen [34], optimize intra-cluster similarity under geometric assumptions, while DBSCAN by Ester et al. [18] identifies dense regions in data and is robust to noise. The most relevant subfield of clustering for our purposes is community detection, which seeks to partition the nodes of a graph into densely connected groups with sparse interconnections. Here, the Louvain algorithm of Blondel et al. [7] and its improved variant Leiden by Traag et al. [41] are state-of-the-art methods that optimize modularity to reveal community structure in networks. Our main metric to compare our algorithms with $k$-means, DBSCAN, Louvain, and Leiden is the Rand distance [39], which—assuming some true labeling for the data points—computes the fraction of pairs of points that are correctly clustered together or separately.

**Our contribution.** We study two stability concepts in hedonic FEN games: local popularity and local stability. The former—introduced in this paper—assures that no single agent can deviate such that more agents improve than get worse, while the latter requires this only for Nash deviations (where the deviating agent improves). Our main result is a simple algorithm that always finds a locally popular partition in polynomial time if friendship and enmity relationships are symmetric in all three settings (friend-appreciative, enemy-averse, and balanced). For asymmetric relationships, we show that a locally stable partition can still be found efficiently in the balanced case, but all other settings yield NP-hardness. See Table 1 for a summary of our results.

We also conduct an experimental study comparing our algorithms to common clustering and community detection algorithms, in particular, to $k$-means, DBSCAN, Louvain, and Leiden. Our simulations provided sufficient data to show that our algorithms are well suited for clustering and community detection tasks, often outperforming existing solutions.

## 2   Preliminaries

Let $N$ denote the set of agents. A subset $C \subseteq N$ is called a *coalition*; in particular, $N$ is the *grand coalition*. In this paper, *partition* refers to a family of pairwise disjoint nonempty subsets of $N$ whose union is $N$. Given a partition $\pi$, the coalition in $\pi$ that contains $i \in N$ is denoted by $\pi(i)$.

A *hedonic game* is a pair $(N, \succeq)$, where $\succeq = (\succeq_i)_{i \in N}$ is a preference profile that contains for each agent $i$ a weak preference list $\succeq_i$ over the possible coalitions containing $i$. That is, for any two coalitions $C, C'$ with $i \in C \cap C'$, either $C' \succ_i C$ (meaning that $C'$ *is preferred to* $C$ *by* $i$), or $C \succ_i C'$ (meaning that $C$ is preferred to $C'$ by $i$), or $C' \sim_i C$ (meaning that $i$ *is indifferent between the two coalitions*). In a hedonic game, the preference lists of the agents only depend on the coalitions they are in. Hence, $\succeq_i$ also induces a weak ranking of agent $i$ over the possible partitions.

Given an agent $i \in N$, a partition $\pi$, and a coalition $C \in \pi \cup \{\emptyset\}$, the deviation where $i$ leaves its original coalition $\pi(i)$ and joins $C$ (or forms a new coalition $\{i\}$ in the case when $C = \emptyset$) is referred to as the *switch* $i \to C$. We use $\pi_{i \to C}$ to denote the partition resulting from $\pi$ by the switch $i \to C$, that is, $\pi_{i \to C} = (\pi \setminus \{\pi(i), C\}) \cup \{\pi(i) \setminus \{i\}, C \cup \{i\}\}$. We say that $i$ *has an incentive to join* $C$ if $C \cup \{i\} \succ_i \pi(i)$; in this case, $i \to C$ is a *Nash deviation*. More generally, extending this notion to partitions, we say that an agent $j$ *votes for* (or, $j$ *votes against*) a switch $i \to C$ if $j$ prefers $\pi_{i \to C}$ to $\pi$ (or, $j$ prefers $\pi$ to $\pi_{i \to C}$, respectively). Further, we define

$$\text{vote}_j(\pi, \pi') = \begin{cases} +1 & \text{if } \pi'(j) \succ_j \pi(j), \\ -1 & \text{if } \pi(j) \succ_j \pi'(j), \\ 0 & \text{if } \pi(j) \sim_j \pi'(j) \end{cases}$$

for partitions $\pi$ and $\pi'$ over $N$. Let $\Lambda(\pi, \pi') = \sum_{j \in N} \text{vote}_j(\pi, \pi')$ be the sum of the agents' votes for $\pi'$ against $\pi$. We will further write $\Lambda_{-i}(\pi, \pi') = \sum_{j \in N \setminus \{i\}} \text{vote}_j(\pi, \pi')$ for the sum of votes for $\pi'$ against $\pi$ among all agents excluding agent $i$.

**Definition 1.** Given a partition $\pi$, an agent $i$, and a coalition $C \in \pi \cup \{\emptyset\}$, we say that the switch $i \to C$ *locally dominates* $\pi$ if $\Lambda(\pi, \pi_{i \to C}) > 0$; and the switch $i \to C$ *locally blocks* $\pi$ if it is a Nash deviation that locally dominates $\pi$. A partition $\pi$ is *popular* if for any partition $\pi'$, we have that

$\Lambda(\pi, \pi') \leq 0$. Moreover, $\pi$ is *locally popular* if there is no switch that locally dominates $\pi$, and $\pi$ is *locally stable* if there is no switch that locally blocks $\pi$.

Note that a single deviation by one agent can also change the situation—and thus the voting behavior on these two partitions—of other agents, giving rise to the above notions of local popularity and local stability. Note further that local popularity implies local stability.

**Preferences in FEN games.** Dimitrov et al. [16] introduced a preference domain for hedonic games where agents' preferences over the coalitions containing them depend only on the number of friends and enemies they have in their coalition. Thus, in a *FEN game* over agent set $N$, each agent $i$ classifies all remaining agents into one of three disjoint categories: those that $i$ considers either *friends*, *enemies*, or *neutrals*; we denote these agent sets by $F(i)$, $E(i)$, and $N(i)$, respectively. The friendship and enmity relations can be represented by two directed graphs, the *friendship digraph* $G^F = (N, A^F)$ and the *enmity digraph* $G^E = (N, A^E)$, where an arc $(i, j) \in A^F$ means that $i$ considers $j$ a friend, whereas an arc $(i, j) \in A^E$ means that $i$ considers $j$ an enemy. We denote by $\Delta$ the maximum number or arcs incident to any agent in $G^F$ or in $G^E$. A FEN game is *symmetric* if friendship and enmity relations are both symmetric; it is *asymmetric* otherwise.

FEN games are subclasses of so-called *additively separable hedonic games* [5] where each agent $i$ has a *valuation function* $v_i$ over $N \setminus \{i\}$ that is defined as

$$v_i(j) = \begin{cases} \alpha_f & \text{if } i \text{ considers } j \text{ a friend, i.e., } j \in F(i), \\ \alpha_e & \text{if } i \text{ considers } j \text{ an enemy, i.e., } j \in E(i), \\ 0 & \text{if } i \text{ considers } j \text{ a neutral, i.e., } j \in N(i) \end{cases}$$

for some integers $\alpha_f > 0$ and $\alpha_e < 0$, and agent $i$'s *utility in a given coalition $C$ containing $i$* is defined as $u_i(C) = \sum_{j \in C \setminus \{i\}} v_i(j)$; we refer to the pair $(\alpha_f, \alpha_e)$ as the *type* of the FEN game. Then the preferences of agent $i$ are based on $i$'s utilities: for two coalitions $C$ and $C'$ both containing $i$, $C \succeq_i C'$ if and only if $u_i(C) \geq u_i(C')$. For a coalition $C \subseteq N$ with $i \in C$, we let $f_C^i = |F(i) \cap C|$ and $e_C^i = |E(i) \cap C|$ denote the number of $i$'s friends and enemies within $C$, respectively; then $u_i(C) = f_C^i \cdot \alpha_f + e_C^i \cdot \alpha_e$. The *utility of agent $i$ in a partition $\pi$* is $u_i(\pi) = u_i(\pi(i))$.

**Friend-appreciating, enemy-averse, and balanced preferences.** Following Dimitrov et al. [16], we introduce the following preference domains for FEN games. A *FEN game with appreciation of friends* (or a *FEN-AF game*, for short) is a FEN game over agent set $N$ whose type is $(|N|, -1)$. That is, $C \succeq_i C'$ holds for two coalitions $C, C'$ with $i \in C \cap C'$ if and only if either $f_C^i > f_{C'}^i$, or $f_C^i = f_{C'}^i$ and $e_C^i \leq e_{C'}^i$. By contrast, a *FEN game with aversion to enemies* (or a *FEN-AE game*, for short) is a FEN game over agent set $N$ whose type is $(1, -|N|)$. That is, $C \succeq_i C'$ holds for two coalitions $C, C'$ with $i \in C \cap C'$ if and only if either $e_C^i < e_{C'}^i$, or $e_C^i = e_{C'}^i$ and $f_C^i \geq f_{C'}^i$. Finally, a *FEN game with balanced preferences* (or a *FEN-B game*, for short) is a FEN game with type $(1, -1)$.

That is, $C \succeq_i C'$ holds fo two coalitions $C, C'$ with $i \in C \cap C'$ if and only if $f_C^i - e_C^i \geq f_{C'}^i - e_{C'}^i$. We will call $u_i^B(C) = f_C^i - e_C^i$ the *balanced utility of $i$ in $C$*.

**Problem definitions.** Let us define the decision problems related to the existence of locally popular and locally stable partitions.

Given a FEN game $I$ of type $(\alpha_f, \alpha_e)$ defined over a set $N$ of agents with a friendship digraph $G^F$ and an enmity digraph $G^E$, the LOCAL-POPULARITY-EXISTENCE problem asks if there exists a locally popular partition for $I$; analogously, the LOCAL-STABILITY-EXISTENCE problem—given the same input—asks if there exists a locally stable partition for $I$.

## 3 Local heuristics

We present two local heuristics, one to find a locally popular partition, the other to find a locally stable partition. Both heuristics start from an arbitrary partition and iteratively modify it using only single-agent deviations; the conditions that determine whether or not to perform a given switch is a local condition that can be checked efficiently.

**LocPop:**
1. Start from an arbitrary partition $\pi$.
2. While there exists a coalition $C \in \pi \cup \{\emptyset\}$ and an agent $i \notin C$ such that $\Lambda(\pi, \pi_{i \to C}) > 0$, replace $\pi$ with $\pi_{i \to C}$ and continue.

Table 1: Summary of our results on the computational complexity of LOCAL-POPULARITY-EXISTENCE and LOCAL-STABILITY-EXISTENCE. We remark that a locally stable or locally popular partition always exists in the settings where the corresponding problem is proven to be in P. By contrast, a locally stable or locally popular partition may not exist in those settings where the corresponding problem is proven to be NP-complete (abbreviated as "NP-c").

| | LOCAL-STABILITY-EXISTENCE | | LOCAL-POPULARITY-EXISTENCE | |
| | symmetric | asymmetric | symmetric | asymmetric |
|---|---|---|---|---|
| FEN-B game | P (Thm. 2) | P (Thm. 4) | P (Thm. 2) | NP-c (Thm. 9) |
| FEN-AF game | P (Thm. 2) | NP-c (Thm. 7) | P (Thm. 2) | NP-c (Thm. 7) |
| FEN-AE game | P (Thm. 2) | NP-c (Thm. 13) | P (Thm. 2) | NP-c (Thm. 13) |

**LocStab:**   1. Start from an arbitrary partition $\pi$.

2. While there exists a coalition $C \in \pi \cup \{\emptyset\}$ and an agent $i \notin C$ such that $\Lambda(\pi, \pi_{i \to C}) > 0$ and $C \cup \{i\} \succ_i \pi(i)$, replace $\pi$ with $\pi_{i \to C}$ and continue.

LocPop and LocStab can both be modified such that the number of coalitions within the partition remains fixed: it suffices to consider switches of the form $i \to C$ where $C \in \pi$ and $\{i\} \notin \pi$ in Step 2.

In the rest of this section, we show that the above heuristics converge in certain FEN games, that is, they arrive at a locally popular or locally stable partition in a finite number of steps. We remark that LocPop and LocStab cannot guarantee to achieve popularity; in fact, we show in Appendix A that deciding whether there exists a popular partition—or even just verifying popularity—is coNP-hard.

Our convergence results both for symmetric and asymmetric FEN games rely on a *potential function* that we define as the total balanced utility of agents: $f(\pi) = \sum_{i \in N} u_i^{\mathrm{B}}(\pi(i)) = \sum_{i \in N} f_{\pi(i)}^i - e_{\pi(i)}^i$.

### 3.1 Symmetric FEN games

**Lemma 1.** *For a FEN game $I$ over agent set $N$, let $\pi$ be a partition, $i \in N$ an agent, and $C \in \pi \cup \{\emptyset\}$ a coalition with $\Lambda(\pi, \pi_{i \to C}) > 0$. Then (a) $\sum_{j \in N \setminus \{i\}} u_j^{\mathrm{B}}(\pi_{i \to C}) - u_j^{\mathrm{B}}(\pi) = \Lambda_{-i}(\pi_i, \pi_{i \to C})$. Moreover, if $I$ is a symmetric FEN game, then (b) $u_i^{\mathrm{B}}(C) - u_i^{\mathrm{B}}(\pi(i)) = \Lambda_{-i}(\pi_i, \pi_{i \to C})$ and (c) $f(\pi_{i \to C}) = f(\pi_i) + 2\Lambda_{-i}(\pi_i, \pi_{i \to C})$.*

*Proof.* First, consider the agents in $N \setminus \{i\}$. Each such agent $j$ may gain or lose a friend or an enemy by the switch $i \to C$, and thus its balanced utility increases exactly by the value of $\mathrm{vote}_j(\pi, \pi_{i \to C}) \in \{-1, 0, 1\}$ as a result of the switch $i \to C$. Summing up the increase in the balanced utility of all agents in $N \setminus \{i\}$, we therefore obtain $\Lambda_{-i}(\pi, \pi_{i \to C})$, proving statement (a) of the lemma.

Assume now that $I$ is symmetric. Consider the contribution of agent $i$ to $f(\pi_{i \to C}) - f(\pi)$. The change in the balanced utility of $i$ as a result of the switch $i \to C$ is exactly $u_i^{\mathrm{B}}(C) - u_i^{\mathrm{B}}(\pi(i)) = \Lambda_{-i}(\pi, \pi_{i \to C})$ as friendships are symmetric, so each agent who gains (or loses) a friend or an enemy due to $i \to C$ is in turn a friend or an enemy gained (or lost) by $i$; this yields statement (b).

Statement (c) then follows by summing up (a) and (b). □

**Theorem 2.** *In a symmetric FEN-AF, FEN-AE, or FEN-B game, irrespective of the initial partition, heuristic LocPop always converges in $\mathcal{O}(n\Delta^2)$ steps (in $\mathcal{O}(n\Delta)$ steps for FEN-B games) to a locally popular (and thus also locally stable) partition, and heuristic LocStab always converges in $\mathcal{O}(n\Delta)$ steps to a locally stable partition.*

*Proof.* Consider a step in heuristic LocPop or LocStab where a partition $\pi$ is replaced by $\pi_{i \to C}$ for some agent $i \in N$ and coalition $C \in \pi \cup \{\emptyset\}$. Then $\Lambda(\pi, \pi_{i \to C}) > 0$ implies $\Lambda_{-i}(\pi, \pi_{i \to C}) \geq 0$. By statement (b) of Lemma 1, this immediately yields that $f(\pi_{i \to C}) \geq f(\pi)$, meaning that $f$ is nondecreasing during the execution of the heuristic. As the total balanced utility is within the range $[-n\Delta, n\Delta]$, there can be at most $2n\Delta$ steps when $f(\pi)$ strictly increases.

Let us now consider the case when $f(\pi_{i \to C}) = f(\pi)$, i.e., the potential function does not increase; our aim is to bound the number of such steps during the execution of the heuristic. By statement (b) of Lemma 1, by $f(\pi_{i \to C}) = f(\pi)$ we must have $\Lambda_{-i}(\pi, \pi_{i \to C}) = 0$. By statement (a) of Lemma 1,

$$\Lambda_{-i}(\pi, \pi_{i \to C}) = f_C^i - e_C^i - f_{\pi(i)}^i + e_{\pi(i)}^i = 0. \tag{1}$$

We introduce $g^F(\pi) = |\{(i,j) \in A^F : \pi(i) = \pi(j)\}|$ and $g^E(\pi) = -|\{(i,j) \in A^E : \pi(i) = \pi(j)\}|$. We distinguish between the following three cases depending on the type of the FEN game.

**Case 1: Balanced preferences.** By Lemma 1, the switch $i \to C$ increases the balanced utility of $i$ by exactly $\Lambda_{-i}(\pi, \pi_{i \to C}) = 0$. Since agent $i$ has balanced preferences, this implies $\text{vote}_i(\pi, \pi_{i \to C}) = 0$. However, then $\Lambda(\pi, \pi_{i \to C}) = 0$ follows, contradicting our assumption that the condition $\Lambda(\pi, \pi_{i \to C}) > 0$ holds. This proves $f(\pi_{i \to C}) > f(\pi)$ for balanced preferences.

**Case 2: Friend-appreciating preferences.** Since we assumed friendships to be symmetric, $g^F(\pi_{i \to C}) - g^F(\pi) = 2f^i_C - 2f^i_{\pi(i)}$. We claim that $g^F(\pi_{i \to C}) > g^F(\pi)$. Indeed, as the switch $i \to C$ was performed by the heuristic based on the condition $\Lambda(\pi, \pi_{i \to C}) > 0$, it follows that agent $i$ must prefer $\pi_{i \to C}$ to $\pi$; hence, either $f^i_C > f^i_{\pi(i)}$ holds as promised, or $f^i_C = f^i_{\pi(i)}$ and $e^i_C < e^i_{\pi(i)}$; however, (1) excludes the latter.

**Case 3: Enemy-averse preferences.** Since we assumed enmities to be symmetric, we have $g^E(\pi_{i \to C}) - g^E(\pi) = 2e^i_{\pi(i)} - 2e^i_C$. We claim that $g^E(\pi_{i \to C}) > g^E(\pi)$. As in Case 2, we have $\Lambda(\pi, \pi_{i \to C}) > 0$, which implies that agent $i$ must prefer $\pi_{i \to C}$ to $\pi$; hence, either $e^i_C < e^i_{\pi(i)}$ holds as promised, or $e^i_C = e^i_{\pi(i)}$ and $f^i_C > f^i_{\pi(i)}$; however, (1) again excludes the latter.

**Total number of steps.** We have shown that the number of steps when the potential function $f$ strictly increases is at most $2n\Delta$. In each such step, $g^F$ and $g^E$ might each decrease by at most $\Delta$, yielding an upper bound of $2n\Delta^2$ on the total decrease during the algorithm. Since $g^F$ takes values within $[0, n\Delta]$ and $g^E$ between $[-n\Delta, 0]$, the number of steps when $f(\pi)$ does not increase is at most $2n\Delta^2 + n\Delta$. This gives a bound of $\mathcal{O}(n\Delta^2)$ on the number of steps taken by the heuristics.

It remains to show that heuristic LocStab (and LocPop for FEN-B games) converges in $\mathcal{O}(n\Delta)$ steps. In FEN-B games, $f$ always strictly increases, as we showed, hence we need $\mathcal{O}(n\Delta)$ steps. Otherwise, in FEN-AE and FEN-AF games, a switch $i \to C$ is only performed in LocStab if $f^i_C \geq f^i_{\pi(i)}$ (or if $e^i_C \leq e^i_{\pi(i)}$); hence $g^F$ (or $g^E$, respectively) never decreases. Thus the number of steps when $f$ does not increase is at most $n\Delta$ in FEN-AF and FEN-AE games, proving the claim for LocStab. □

By applying an efficient way to decide if a switch is locally dominating and by using a carefully maintained data structure for quickly finding such switches, our heuristics can be implemented in near-linear time if the maximum degree $\Delta$ of the friendship and enmity digraphs is constant; see Appendix B. As efficiency is often paramount in clustering, we believe that this result significantly widens the possibilities for applying our heuristics in practical clustering instances. Theorem 3 is proven in Appendix B, while all further results marked with ($\star$) are proven in Appendix D.

**Theorem 3 ($\star$).** LocPop *can be implemented to run in* $\mathcal{O}(n\Delta^3 \log n)$ *time (or* $\mathcal{O}(n\Delta^2 \log n)$ *for FEN-B games) and* LocStab *can be implemented to run in* $\mathcal{O}(n\Delta^2 \log n)$ *time.*

### 3.2 Asymmetric FEN games

The following example shows that heuristics LocPop and LocStab may not stop in an asymmetric FEN game with friend-appreciating or enemy-averse preferences.

**Example 1.** In an *asymmetric* FEN-AF or FEN-AE game, heuristics LocPop and LocStab can run into a cycle and never stop. Let there be seven agents: $0, 1, \ldots, 6$. Each agent $i$ considers $(i+1) \mod 7$ a friend but $(i+2) \mod 7$ and $(i+3) \mod 7$ enemies; see Figure 1a for an illustration. (Here and anywhere else, all relations that are not explicitly mentioned are tacitly assumed to be neutral.)

In the AF domain, let the initial partition be $\{\{0, 1, 2, 3\}, \{4, 5, 6\}\}$. Then agent 3 has an incentive to join $\{4, 5, 6\}$ and this switch takes place, as agents 0, 1, and 3 vote for it and only agent 2 votes against it. By symmetry, this creates a loop: agent 6 switches next, then agent 2, then agent 5, etc.

In the AE domain, let the initial partition be $\{\{0, 1\}, \{2, 3\}, \{4, 5, 6\}\}$. Then agent 4 has an incentive to join $\{2, 3\}$ and this switch takes place, as agents 4 and 3 vote for it and only agent 2 votes against it. Again, this creates a loop: agent 2 switches next, then agent 0, then agent 5, etc.

**Example 2.** In an *asymmetric* FEN-B game, heuristic LocPop can run into a cycle and never stop. Let the agent set be $\{0, 1, 2\}$, where each agent $i$ considers $(i-1) \mod 3$ a friend and $(i+1) \mod 3$ an enemy. Let the initial partition consist of two coalitions: $\{0, 1\}$ and $\{2\}$. Then both 0 and 2 vote

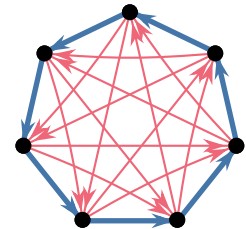
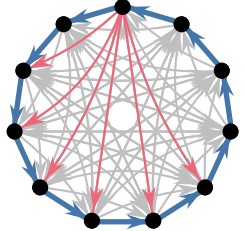
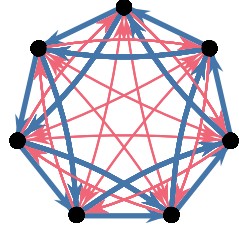

(a) The FEN game in Example 1.     (b) An AF-gadget on 11 agents.     (c) A B-gadget.

Figure 1: Illustration for Example 1 and Definitions 2 and 3. We use thick, blue arrows for friendship arcs and thin, red ones for enmity arcs (some enmity arcs in (b) are shown in gray for better visibility).

for the switch of $1 \to \{2\}$, which hence takes place. Next, the switch $2 \to \{0\}$ takes place, followed by the switch $0 \to \{1\}$, and the heuristic arrives back at the initial partition.

Contrasting Examples 1 and 2, heuristic LocStab always stops if agents' preferences are balanced.

**Theorem 4** (⋆). *In a possibly asymmetric FEN-B game, the total balanced utility of all agents strictly increases in each step of heuristic* LocStab*; consequently, heuristic* LocStab *always converges in* $\mathcal{O}(n\Delta)$ *steps to a locally stable partition, irrespective of the initial partition.*

## 4   Hardness results for local popularity and local stability

**Friend-appreciating FEN games.** Let us introduce the gadget which lies at the heart of our hardness results on FEN-AF games; see Figure 1b for an illustration.

**Definition 2.** Let $k \geq 5$ be an integer with $(2k+1) \bmod 3 \neq 0$. A set $\{0, 1, \ldots, 2k\}$ of agents forms an *AF-gadget* of size $2k+1$ if each agent $i$ considers $(i+1) \bmod (2k+1)$ a friend (so that the friendship graph is a directed cycle) and considers $(i+2), (i+3), \ldots, (i-4) \bmod (2k+1)$ enemies.

**Theorem 5** (⋆). *The asymmetric FEN-AF game consisting solely of an AF-gadget admits no locally popular partition.*

Building on AF-gadgets, a more involved construction shows that even a locally stable solution may fail to exist. In fact, Theorem 7 shows that it is NP-hard to decide whether a locally popular or a locally stable partition exists; our reductions are from 5-SAT and make ample use of AF-gadgets.

**Theorem 6** (⋆). *There exists an asymmetric FEN-AF game that admits no locally stable partition.*

**Theorem 7** (⋆). LOCAL-POPULARITY-EXISTENCE *and* LOCAL-STABILITY-EXISTENCE *are* NP-*complete for FEN-AF games.*

**Balanced FEN games.** Turning our attention to balanced FEN games, recall that given a FEN-B game $I$, heuristic LocStab always finds a locally stable partition for $I$ in polynomial time, as stated in Theorem 4. By contrast, we introduce a gadget that is similar to the AF-gadget from Definition 2 but is tailored for showing that FEN-B games may not admit a locally popular partition. Relying on B-gadgets, by a reduction from the NP-complete problem 3-COLORING we can show the NP-hardness of the related decision problem, as stated in Theorem 9.

**Definition 3.** A set $\{0, 1, \ldots, 6\}$ of agents forms a *B-gadget* if each agent $i$ considers $(i+1) \bmod 7$ and $(i+2) \bmod 7$ a friend and considers $(i+3), (i+4), (i+5) \bmod 7$ enemies.

**Theorem 8** (⋆). *The asymmetric FEN-B game consisting solely of a B-gadget admits no locally popular partition.*

**Theorem 9** (⋆). LOCAL-POPULARITY-EXISTENCE *is* NP-*complete for FEN-B games.*

As we have shown in Theorem 4, heuristic LocStab increases the agents' total balanced utility—the potential function $f$—in each step when running on a FEN-B game. Hence, a partition $\pi$ maximizing $f(\pi)$ is locally stable but, as Theorem 8 shows, not necessarily locally popular. Interestingly, not

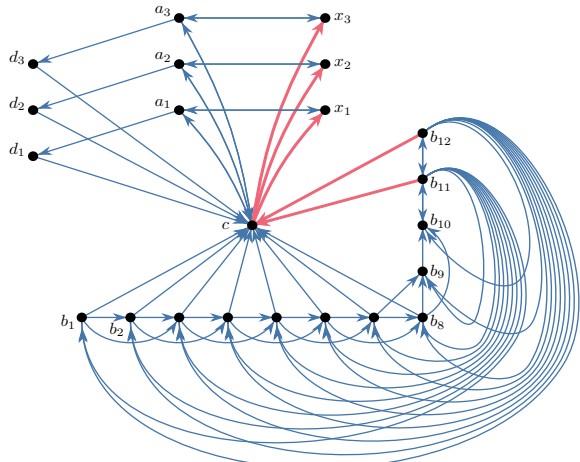

Figure 2: Illustration of the AE-gadget as defined in Definition 4. Here we denote friendship arcs as thin, blue lines and enemy arcs as thick, red lines.

even a popular partition is guaranteed to reach the maximum total balanced utility, even in symmetric FEN-B games; see Proposition 10. Moreover, deciding whether there exists a partition that achieves a given total balanced utility is NP-hard, as shown by Theorem 11.

**Proposition 10** ($\star$). *There exists a symmetric FEN-B game that admits a partition that is popular but does not maximize the agents' total balanced utility.*

**Theorem 11** ($\star$). *Given a symmetric FEN-B game $I$ and an integer $t$, the problem of deciding whether $I$ admits a partition $\pi$ whose total balanced utility is at least $t$ is NP-complete.*

**Enemy-averse FEN games.** Similarly to the gadgets from Definitions 2 and 3, we introduce an AE-gadget for FEN-AE games for proving that such games may not admit a locally stable partition. A reduction from 3-SAT based on AE-gadgets shows the intractability of deciding the existence of locally popular or locally stable partitions. For $k \in \mathbb{N} \setminus \{0\}$, we write $[k]$ as a shorthand for $\{1, 2, \ldots, k\}$.

**Definition 4.** An *AE-gadget* over a player set $K = \{c\} \cup \{b_i : i \in [12]\} \cup \{a_i, d_i, x_i : i \in [3]\}$ has the following friendship and enemy relations: $c$ considers $a_i$ and $d_i$ for $i \in [3]$ friends and agents $x_1, x_2, x_3, b_{11}$, and $b_{12}$ enemies; $b_i$ considers $c$, $b_{i+1}$, and $b_{i+2}$ a friend for each $i \in [8]$; $b_9$ considers $b_{10}$ a friend, and $b_{10}$ considers $b_{11}$ a friend; $b_{11}$ and $b_{12}$ consider each other and all agents in $\{b_i : i \in [10]\}$ friends and $c$ an enemy; $a_i$ considers $x_i$, $d_i$, and $c$ a friend for each $i \in [3]$; $d_1$, $d_2$, and $d_3$ consider $c$ a friend; and $x_i$ considers $a_i$ a friend for each $i \in [3]$. See Figure 2.

**Theorem 12** ($\star$). *The asymmetric FEN-AE game consisting solely of an AE-gadget admits no locally stable partition.*

**Theorem 13** ($\star$). LOCAL-POPULARITY-EXISTENCE *and* LOCAL-STABILITY-EXISTENCE *are* NP-complete for FEN-AE games.

## 5 Simulations

We complement our theoretical study with a range of experiments.

**Setup.** We implemented LocPop and LocStab in Python and run the simulations on a computer with AMD Ryzen 7735HS CPU and 16GB RAM. All codes are available in the supplementary material.

For our algorithms, we chose parameters $f \leq e \in [0, 1]$ to create the friendship and enmity graphs: for two data points $x$ and $y$ at distance $d(x, y)$ in an instance $I$,[1] we added $(x, y)$ to the friendship graph

---

[1] When points are described via their coordinates, we take the Euclidean distance, while in an instance described as a network, the distance of $x$ an $y$ is measured by the number of edges of a shortest $xy$-path.

if $d(x, y) \leq f \cdot \mathrm{diam}(I)$ and to the enmity graph if $d(x, y) > e \cdot \mathrm{diam}(I)$, where $\mathrm{diam}(I)$ denotes the *diameter of* $I$, i.e., the maximum distance between any two points in $I$. We used the parameter values $(f, e) \in \{(0.2, 0.2), (0.25, 0.35), (0.4, 0.4)\}$. For each parameterization, we considered the *appreciation-of-friends* (AF), *aversion-to-enemies* (AE), and the *balanced* (B) preference domains.

**Community Detection.** For community detection, we used four different datasets.

1. **Karate club** [23, 31, 43]: a 34-node benchmark dataset for community detection.
2. **Jazz musicians** [24, 32]: collaboration network of 198 jazz musicians; nodes represent musicians, edges represent co-membership in a band.
3. **Cora dataset** [36, 35]: a citation network of 2708 machine learning papers classified into seven classes; edges denote citation links.
4. **Random-25**: an instance containing 25 disjoint Erdős-Rényi graphs (10 nodes each, with $p = 0.2$) as communities; inter-community edges added independently with probability 0.05.

We compared our heuristics with two state-of-the-art algorithms, called Louvain and Leiden [2]. We tested our heuristics with respect to several different parameters. In particular, we considered the following variations for the initial clustering: (i) putting each agent into a singleton cluster (LocPop-S, LocStab-S), (ii) dividing agents randomly into $k$ clusters where $k$ is the predicted number of clusters (LocPop-P, LocStab-P), and (iii) using the output of the Leiden algorithm (LocPop-Ld, LocStab-Ld).

For evaluation, we used *Rand index* and *modularity*, both commonly used metrics. The Rand index assumes underlying true labels for the data points, and measures the similarity of the obtained clustering to the true clustering. More precisely, it computes the fraction of such $(x, y)$ pairs where $x$ and $y$ are correctly put together or into different clusters. By contrast, modularity does not necessitate a true labeling, but instead evaluates how well the graph underlying the instance is divided into clusters, based on the number of edges within the clusters in between different clusters. See Figure 3 and Appendix C.1.

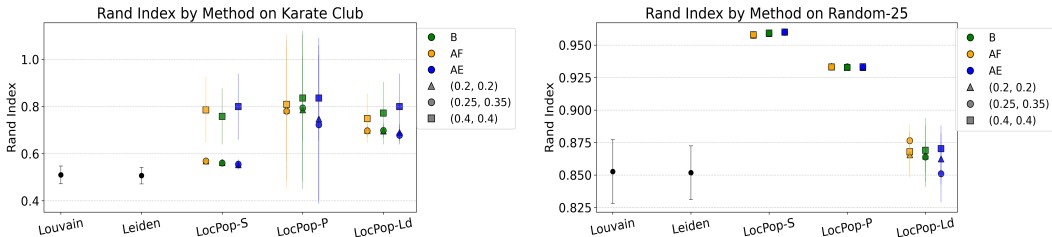

Figure 3: Comparing Rand index of Leiden, Louvain, and LocPop variants on the Karate club dataset (left) and the synthetic dataset Random-25 (right). In all figures henceforth, 2-sigma error bars show the standard deviation of the plotted data.

**Clustering.** For our clustering simulations, we used four datasets.

1. **Iris dataset** [21]: 150 samples from three Iris species, features are sepal and petal sizes.
2. **Breast cancer Wisconsin dataset** [42]: 569 samples from diagnostic images (30 dimensional datapoints).
3. **Moons dataset** [38]: two half-moons generated by `make_moons` (300 points, 0.05 noise).
4. **3-Circles dataset**: 300 points in three slightly overlapping circles centered at $(0.5, 0.5)$, $(0.7, 0.3)$, and $(0.1, 0.7)$ with radius 0.2 and Gaussian noise (std $= 0.05$).

We compared our algorithms to two widely used clustering algorithms, $k$-means and DBSCAN, using their initial parameters. We tested our heuristic with respect to several different parameters. In particular, we considered the following variations for the initial clustering: (i) putting every agent into a singleton cluster (LocPop-S, LocStab-S), (ii) dividing the agents randomly into $k$ clusters where $k$ is the predicted number of clusters (LocPop-P, LocStab-P), (iii) using the output of $k$-means (LocPop-$k$M, LocStab-$k$M), and (iv) using the output of DBSCAN (LocPop-D, LocStab-D).

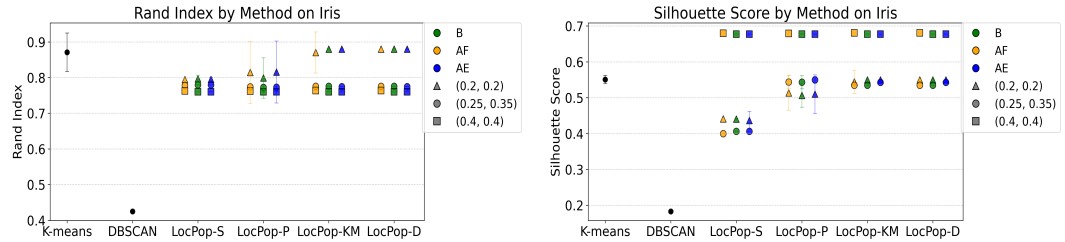

Figure 4: Comparing Rand index (left) and silhouette score (right) of $k$-means, DBSCAN, and LocPop variants for the Iris dataset.

To evaluate the outputs, we considered the Rand index and *silhouette score*, both being commonly used metrics in clustering. As opposed to Rank index, silhouette score does not assume the existence of a true labeling of the given data points, but evaluates the quality of the obtained clustering by measuring the cohesion within clusters as well as the separation between different clusters. See Figure 4 and Appendix C.2.

**Key insights from experimental results.** Our algorithms run within seconds, except for Cora, where computation takes 3-5 minutes, probably due to high degrees in the friendship and enmity graphs. LocPop and LocStab performed very similarly, with LocStab depending more on the preference domain, so we discuss LocPop here. A more detailed evaluation can be found in Appendix C.3.

LocPop demonstrated strong and consistent empirical performance, performing very similarly over the three preference domains (AF, AE, and B) . We observed that parameter tuning can significantly affect outcomes, usually either the setting $(0.2, 0.2)$ or $(0.4, 0.4)$ performed best, and $(0.25, 0.35)$ performed consistently between them. Interestingly, stronger performance on standard clustering metrics (silhouette score) often did not align with being closer to the true labels (Rand index).

In community detection tasks, LocPop matched or outperformed Louvain and Leiden with respect to the Rand index, particularly excelling on the Karate club ($\sim 50\%$ better) and Random-25 ($\sim 10\%$ better) datasets. The LocPop-Ld variant also matched them with respect to modularity. In clustering benchmarks, it mostly surpassed DBSCAN in both silhouette score and Rand index, and also matched or outperformed $k$-means (with up to $20\%$ in 3 Circles and Iris in silhouette score). These findings indicate that LocPop is both theoretically grounded and competitive in practice, offering a flexible and efficient approach for structure discovery while providing additional robustness against deviations.

## 6 Conclusion

We have investigated the computational complexity of finding a popular, a locally popular, or a locally stable partition in FEN-AF, FEN-AE, and FEN-B games and identified which cases were solvable efficiently and which were NP-hard. Our algorithms can also be interpreted as dynamics of agents based on single-agent deviations and majority-based decision making.

Our experimental study provided sufficient background to show that social-choice concepts like local stability and local popularity may lead to efficient clustering and community detection techniques. Furthermore, by their nature, these concepts may often provide fairer solutions that are also more stable against potential deviations, which—as argued in the introduction—is particularly relevant in scenarios where data points correspond to agents capable of making independent actions.

## Acknowledgments and Disclosure of Funding

This work was supported in part by Deutsche Forschungsgemeinschaft under DFG research grant RO-1202/21-2 (project 438204498). Csáji was also supported by the Hungarian Scientific Research Fund, OTKA, Grant No. K143858 and by the National Research, Development and Innovation fund, under the KDP-2023 funding scheme (grant number C2258525). Schlotter was also supported by the Hungarian Academy of Sciences under its Momentum Programme (LP2021-2) and its János Bolyai Research Scholarship.

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

# Appendix A  Existence of a popular partition in FEN games

Recall that Theorems 2 and 3 say that *locally* popular partitions *always* exist and can be efficiently found by a simple heuristic in symmetric FEN-AF and FEN-AE games. In this section, we show that the same does not hold for popularity: there exist symmetric FEN games where for each possible partition $\pi$, there is a deviation involving multiple agents that yields a partition $\pi'$ with $\Lambda(\pi, \pi') > 0$. Specifically, we show that the verification problem for popularity in symmetric FEN games is coNP-complete. Furthermore, we show that deciding whether a popular partition exists in such games is computationally intractable as well; specifically, the existence problem for popularity in symmetric FEN games is coNP-hard. These results on the complexity of popularity verification and existence are the same as those known for other types of hedonic games, such as the altruistic hedonic games studied by Kerkmann and Rothe [29]. Interestingly, Bullinger and Gilboa [12] recently settled the complexity of whether there exist popular partitions in either additively separable or fractional hedonic games by showing that these two problems are complete for $\mathrm{NP}^{\mathrm{NP}}$, the second level of the polynomial hierarchy—the first results closing the gap between the computational upper and lower bounds for popularity existence in hedonic games. For our problems, we leave this question open.

We prove the following result for all FEN games, among them FEN-AF, FEN-AE, and FEN-B games.

**Theorem 14.** *Given a FEN game $I$ of some fixed, arbitrary type with symmetric preferences, verifying whether a partition is popular in $I$ is* coNP-*complete, while deciding whether a popular partition exists for $I$ is* coNP-*hard.*

*Proof.* It is clear that verifying whether a given partition $\pi$ is popular in a given FEN game $I$ of some fixed, arbitrary type with symmetric preferences is in coNP, because we can nondeterministically guess a partition $\pi'$ and check in polynomial time whether $\Lambda(\pi, \pi') > 0$. This is the case for some partition $\pi'$ exactly if $\pi$ is not popular in $I$.

To prove coNP-hardness, we present a reduction from the EXACT 3-COVER problem whose input is a set $U = \{u_1, \ldots, u_{3n}\}$ and a set family $\mathcal{F} = \{F_1, \ldots, F_m\}$ with each $F_i$ being a size-3 subset of $U$; the task is to decide whether there exists a set family $\mathcal{S} \subseteq \mathcal{F}$ of size $n$ such that $U = \bigcup_{F_i \in \mathcal{S}} F_i$.

We construct a FEN game $I$ of the given type as follows; our reduction is correct irrespective of the type of $I$. For each element $u_i \in U$, we create an agent $\overline{u}_i$, and for each set $F_j \in \mathcal{F}$, we create an agent $\overline{F}_j$; we use the notation $\overline{U} = \{\overline{u} : u \in U\}$ and $\overline{\mathcal{F}} = \{\overline{F}_j : F_j \in \mathcal{F}\}$. We further add two sets $A$ and $A'$ of *anchor* agents with $|A| = |A'| = 5n + m$ and a set $B$ of *guard* agents with $|B| = 2n - 1$. Hence, the set of agents in $I$ is $A \cup A' \cup B \cup \overline{U} \cup \overline{\mathcal{F}}$.

The relationships for $I$ are defined as follows; recall that all relationships in $I$ are symmetric:

- Each anchor agent in $A$ is friends with all other anchor agents in $A$ and all agents in $\overline{U}$.

- Each anchor agent in $A'$ is friends with all other anchor agents in $A'$ and all guard agents.

- Each guard agent is friends with all agents in $\overline{\mathcal{F}}$.

- An agent $\overline{u}_i \in \overline{U}$ is friends with an agent $\overline{F}_j \in \overline{\mathcal{F}}$ if and only if $u_i \in F_j$.

- Each anchor agent in $A$ and each anchor agent in $A'$ are enemies.

We claim that $I$ admits a popular partition if and only if $\pi^\star = \{A \cup \overline{U}, A' \cup B \cup \overline{\mathcal{F}}\}$ is a popular coalition, which in turn happens if and only if our input instance $I_{\text{X3C}} = (U, \mathcal{F})$ of EXACT 3-COVER does *not* have a solution, i.e., there is no exact cover of $B$ in $\mathcal{F}$.

First, let us show that if $\mathcal{S} \subseteq \mathcal{F}$ is a solution to $I_{\text{X3C}}$, then $\pi^\star$ is not popular. To this end, let us define $\overline{\mathcal{S}} = \{\overline{F}_j : F_j \in \mathcal{S}\}$. We are going to show that $\Lambda(\pi, \pi^\star) > 0$ where the partition

$$\pi = \{A \cup \overline{U} \cup \overline{\mathcal{S}}, A' \cup B \cup \overline{\mathcal{F}} \setminus \overline{\mathcal{S}}\} \tag{2}$$

is obtained from $\pi$ by a deviation of agents in $\overline{\mathcal{S}}$; this implies that $\pi^\star$ is not popular. Note that each element $u_i \in U$ is contained in some set $F_j \in \mathcal{S}$, which means that each agent $\overline{u}_i \in \overline{U}$ has more friends in $\pi(\overline{u}_i)$ than in $\pi^\star(\overline{u}_i)$ (and zero enemies in both) and thus prefers $\pi$ to $\pi^\star$. By contrast, each guard agent $b \in B$ has fewer friends in $\pi(b)$ than in $\pi^\star(b)$ (and zero enemies

in both) and thus prefers $\pi^\star$ to $\pi$. Furthermore, each agent $\overline{F_j} \in \overline{\mathcal{S}}$ gains three new friends and loses all of its guard friends when switching from $\pi^\star$ to $\pi$ (and has zero enemies in both) and thus prefers $\pi^\star$ to $\pi$. Since every remaining agent is indifferent between $\pi$ and $\pi^\star$, we obtain that $\Lambda(\pi, \pi^\star) = |\overline{U}| - |B| - |\overline{\mathcal{S}}| = 3n - (2n-1) - n = 1 > 0$.

Next, we show that if $\pi^\star$ is not popular, then $I_{\text{X3C}}$ has a solution. Let $\pi$ be a partition that is more popular than $\pi^\star$, i.e, $\Lambda(\pi^\star, \pi) > 0$. Observe that for an anchor agent $a \in A \cup A'$, all friends and no enemies of $a$ are in $\pi^\star(a)$, so anchor agents cannot prefer $\pi$ to $\pi^\star$. Moreover, if some anchor agent prefers $\pi^\star$ to $\pi$, then at least $|A| = |A'| = 5n + m$ anchor agents do so; however, at most $|\overline{U}| + |\overline{\mathcal{F}}| + |B| = 3n + m + 2n - 1$ agents may prefer $\pi$ to $\pi^\star$, which contradicts $\Lambda(\pi^\star, \pi) > 0$. Thus anchor agents must be indifferent between $\pi$ and $\pi^\star$, which yields that $\pi$ contains two coalitions $C$ and $C'$ such that $C \supseteq A \cup \overline{U}$ and $C' \supseteq A' \cup B$.

Define $\overline{\mathcal{S}} = C \cap \overline{\mathcal{F}}$; note that $|\overline{\mathcal{S}}| > 0$, as otherwise, $\pi = \pi^\star$. Note that each guard agent $b \in B$ has fewer friends in $\pi(b)$ than in $\pi^\star(b)$ (and zero enemies in both) and thus prefers $\pi$ to $\pi^\star$, and so do the agents in $\overline{\mathcal{S}}$ as well (they gain three friends each from $\overline{U}$ but lose all guards from among their friends when deviating to $\pi$). Let $\overline{V}$ be the set of agents in $\overline{U}$ who prefer $\pi$ to $\pi^\star$; notice that $\overline{V} = \{\overline{u}_i : \exists \overline{F}_j \in \overline{\mathcal{S}} \text{ such that } u_i \in F_j\}$ by construction.

Then the agents preferring $\pi$ to $\pi^\star$ are those in $\overline{V}$ while those that prefer $\pi^\star$ to $\pi$ are those in $B \cup \overline{\mathcal{S}}$. Hence, $\Lambda(\pi^\star, \pi) = |\overline{V}| - |B| - |\overline{\mathcal{S}}| \geq 1$, which implies that

$$|\overline{V}| \geq (2n - 1) + |\overline{\mathcal{S}}| + 1. \tag{3}$$

Recall also that

$$|\overline{V}| \leq 3n \qquad \text{and} \qquad |\overline{V}| \leq 3|\overline{\mathcal{S}}|, \tag{4}$$

because each agent in $\overline{\mathcal{S}}$ has three friends in $\overline{V}$. Taking the linear combination of these two inequalities, we obtain that $|\overline{V}| \leq 2n + |\overline{\mathcal{S}}|$, and thus the three inequalities in (3) and (4) must all hold with equality. In particular, $|\overline{V}| = 3n$ and $|\mathcal{S}| = n$, and the former implies $\overline{V} = \overline{U}$, that is, $\bigcup_{\overline{F}_j \in \overline{\mathcal{S}}} F_j = U$. Therefore, the set family $\mathcal{S} = \{F_j : \overline{F}_j \in \overline{\mathcal{S}}\}$ is a solution to the EXACT 3-COVER instance $I_{\text{X3C}}$.

It remains to show that if $\pi^\star$ is not popular, then there is no popular partition $\pi$ at all. For the sake of contradiction, assume that $\pi$ is popular. We have already proven that $\Lambda(\pi, \pi^\star) \geq 0$ implies that $\pi$ must be of the form (2) for some $\overline{\mathcal{S}} \subseteq \overline{\mathcal{F}}$. By our assumption that $\pi^\star \neq \pi$, we also know that $\overline{\mathcal{S}}$ is nonempty. However, then the switch $\overline{F}_j \to A' \cup B \cup \overline{\mathcal{F}} \setminus \overline{\mathcal{S}}$ locally blocks $\pi$ for any agent $\overline{F}_j \in \overline{\mathcal{S}}$, because all guard agents as well as $\overline{F}_j$ votes for it, whereas at most three agents in $\overline{U}$ vote against it. This shows that no partition other than $\pi^\star$ may be popular in $I$. $\qquad \square$

Next, we show that popular partitions may fail to exist even in a symmetric *FE game* where neutral relations are not allowed and thus every agent that is not a friend of $i$ is an enemy of $i$, i.e., the friendship digraph and the enmity digraph are complements of each other. This suggests that the intractability shown in Theorem 14 does not stem from the possibility of neutrality.

**Proposition 15.** *There exists a FE game with symmetric and friend-appreciative (or, equivalently, with symmetric and enemy-averse) preferences that does not admit a popular partition.*

*Proof.* Consider the FE game $I$ with symmetric and friend-appreciative (or equivalently, symmetric and enemy-averse) preferences on agent set $N = \{0, 1, \ldots, 6\}$ where the friendship graph is a bidirected cycle of length 7: every agent $i \in N$ considers both $(i-1) \bmod 7$ and $(i+1) \bmod 7$ a friend and each other agent in $N \setminus \{i\}$ an enemy. We claim that $I$ does not admit a popular partition.

First, the grand coalition $\pi^N = \{N\}$ is not popular, as $\Lambda(\pi^N, \pi^N_{i \to \emptyset}) > 0$ for each agent $i$: even though $i$ and both friends of $i$ vote against the switch $i \to \emptyset$, all four remaining agents vote for it.

Second, if $\pi$ is a partition such that there are at least four agents $i \in N$ with $f^i_{\pi(i)} \leq 1$, then $\Lambda(\pi, \pi^N) > 0$, because these four agents prefer the grand coalition to $\pi$. It follows that any popular partition must be of the form $\pi^i = \{\{i\}, N \setminus \{i\}\}$ for some $i \in N$. However, in this case, the switch $(i+1) \bmod 7 \to \{i\}$ locally blocks $\pi^i$, because all agents vote for this switch except for the agent $(i+2) \bmod 7$; in particular, $\Lambda(\pi^i, \pi^i_{(i+1) \bmod 7 \to \{i\}}) > 0$, contradicting the popularity of $\pi^i$. Hence, no popular partition exists for $I$. $\qquad \square$

# Appendix B  Efficient implementation for LocPop and LocStab

In this section, we present an efficient implementation of LocPop and LocStab that proves Theorem 3.

**Theorem 3** ($\star$). LocPop *can be implemented to run in* $\mathcal{O}(n\Delta^3 \log n)$ *time (or* $\mathcal{O}(n\Delta^2 \log n)$ *for FEN-B games) and* LocStab *can be implemented to run in* $\mathcal{O}(n\Delta^2 \log n)$ *time.*

*Proof.* Regarding the running time of LocPop and LocStab, it is clear that each step can be performed in $\mathcal{O}(n\Delta k)$ time even in asymmetric FEN games, where $k$ denotes the number of coalitions in the current partition: to check for each agent $i \in N$ and each coalition $C \in \pi \cup \{\emptyset\}$ whether $i \to C$ locally dominates or locally blocks, it suffices to know the number of agents in $C$ and in $\pi(i)$ who regard $i$ a friend or an enemy and of those whom $i$ considers a friend or an enemy. As the number of such agents is at most $2\Delta$, each step can be done in $|N|(k+1)\mathcal{O}(\Delta) = \mathcal{O}(nk\Delta)$ time.

In the following, we show how our heuristics can be implemented much more efficiently; we will focus on symmetric FEN games due to their practical relevance. We start with a way to characterize when a switch $i \to C$ locally dominates a partition $\pi$.

**Lemma 16.** *Given a symmetric FEN game $I$ with a partition $\pi$, consider the following conditions for some agent $i$ and coalition $C \in \pi \cup \{\emptyset\}$:*

(c0)  $f_C^i - e_C^i \geq f_{\pi(i)}^i - e_{\pi(i)}^i + 1$;       (c3)  $2f_C^i - e_C^i > 2f_{\pi(i)}^i - e_{\pi(i)}^i$;

(c1)  $f_C^i - e_C^i \geq f_{\pi(i)}^i - e_{\pi(i)}^i + 2$;       (c4)  $f_C^i - 2e_C^i > f_{\pi(i)}^i - 2e_{\pi(i)}^i$;

(c2)  $f_{\pi(i)}^i - e_{\pi(i)}^i + 1 \geq f_C^i - e_C^i \geq f_{\pi(i)}^i - e_{\pi(i)}^i$.

*If $I$ is a FEN-B / FEN-AF / FEN-AE game, then the switch $i \to C$ locally dominates $\pi$ if and only if condition (c0) / condition (c1) $\vee$ ((c2) $\wedge$ (c3)) / condition (c1) $\vee$ ((c2) $\wedge$ (c4)), respectively, hold.*

*Proof.* Statement (b) of Lemma 1 yields

$$\Lambda(\pi, \pi_{i \to C}) = f_C^i - e_C^i - f_{\pi(i)}^i + e_{\pi(i)}^i + \text{vote}_i(\pi, \pi_{i \to C}). \tag{5}$$

We first show that the respective conditions imply that $i \to C$ locally dominates $\pi$.

First, if (c1) holds, then $\Lambda(\pi, \pi_{i \to C}) \geq 2 + \text{vote}_i(\pi, \pi_{i \to C}) > 0$ due to (5), so the switch $i \to C$ locally dominates $\pi$ even if $i$ votes against it, irrespective of the type of $I$. If $I$ is a FEN-B game, then (c0) already suffices to ensure that $i \to C$ locally dominates $\pi$, because it implies $\text{vote}_i(\pi, \pi_{i \to C}) \geq 1$ by the definition of balanced preferences, which in turn yields $\Lambda(\pi, \pi_{i \to C}) \geq 2$ due to (5).

If (c2) holds, then $\Lambda(\pi, \pi_{i \to C}) \geq \text{vote}_i(\pi, \pi_{i \to C})$ due to (5) and, hence, it suffices to show that (c3) implies that $i \to C$ is a Nash deviation under friend-appreciating preferences, whereas (c4) implies that $i \to C$ is a Nash deviation under enemy-averse preferences:

- Assume first that $I$ is a FEN-AF game where (c2) and (c3) hold, but $i \to C$ is not a Nash deviation. Then either $f_{\pi(i)}^i = f_C^i$ and $e_{\pi(i)}^i \leq e_C^i$, or $f_{\pi(i)}^i > f_C^i$. The former case cannot happen, as it contradicts (c3). Hence, we must have $f_{\pi(i)}^i \geq f_C^i + 1$; however, adding this inequality to $f_{\pi(i)}^i - e_{\pi(i)}^i + 1 \geq f_C^i - e_C^i$, we get a contradiction to (c3) again.

- Assume now that $I$ is a FEN-AE game where (c2) and (c4) hold, but $i \to C$ is not a Nash deviation; our reasoning mirrors the previous case. Then either $e_{\pi(i)}^i = e_C^i$ and $f_{\pi(i)}^i \geq f_C^i$, or $e_{\pi(i)}^i < e_C^i$. The former case cannot happen, as it contradicts (c4). Hence, we must have $e_C^i \geq e_{\pi(i)}^i + 1$; however, adding this inequality to $f_{\pi(i)}^i - e_{\pi(i)}^i + 1 \geq f_C^i - e_C^i$, we get a contradiction to (c4) again.

For the other direction, suppose that $i \to C$ locally dominates $\pi$, i.e., $\Lambda(\pi, \pi_{i \to C}) > 0$. Then $\Lambda_{-i}(\pi, \pi_{i \to C}) \geq 0$; recall that $\Lambda_{-i}(\pi, \pi_{i \to C}) = f_C^i - e_C^i - f_{\pi(i)}^i + e_{\pi(i)}^i$ by statement (b) of Lemma 1, so we get

$$f_C^i - e_C^i \geq f_{\pi(i)}^i - e_{\pi(i)}^i. \tag{6}$$

If $\Lambda_{-i}(\pi, \pi_{i \to C}) \geq 2$, then (c1) is satisfied. Else, $\Lambda_{-i}(\pi, \pi_{i \to C}) \in \{0, 1\}$, which implies (c2), and we also get $\text{vote}_i(\pi, \pi_{i \to C}) \geq 0$ due to (5) and the assumption that $i \to C$ locally dominates $\pi$. In

fact, $\text{vote}_i(\pi, \pi_{i \to C}) = 0$ is not possible, as that would mean $f^i_{\pi(i)} - e^i_{\pi(i)} = f^i_C - e^i_C$ implying $\Lambda(\pi, \pi_{i \to C}) = 0$ by (5); a contradiction to our assumption that $\Lambda(\pi, \pi_{i \to C}) > 0$. Therefore, $i \to C$ must be a Nash deviation; using this, we now show that the required conditions hold.

- For balanced preferences, this means that (5) implies $\Lambda(\pi, \pi_{i \to C}) \geq 2$ which, due to (5), implies (c0).

- For friend-appreciating preferences, we get $f^i_C > f^i_{\pi(i)}$, or $f^i_C = f^i_{\pi(i)}$ and $e^i_C < e^i_{\pi(i)}$; taking (6) into account, both of these cases yield $2f^i_C - e^i_C > 2f^i_{\pi(i)} - e^i_{\pi(i)}$, i.e., (c3) holds.

- For enemy-averse preferences, we get $e^i_C < e^i_{\pi(i)}$, or $e^i_C = e^i_{\pi(i)}$ and $f^i_C > f^i_{\pi(i)}$; taking (6) into account, both of these cases imply $f^i_C - 2e^i_C > f^i_{\pi(i)} - 2e^i_{\pi(i)}$, i.e., (c4) holds.

This completes the proof of Lemma 16. $\qquad\qquad\qquad\qquad\qquad\qquad\qquad\qquad\qquad\quad\square$

**Implementation.** Let us describe the implementation of heuristics LocPop and LocStab in detail.

The key idea is that for each agent $i \in N$, we maintain two vectors $x^1_i$ and $x^2_i$ that are used to store the following values corresponding to each coalition $C \in \pi \cup \{\emptyset\}$. The first vector stores the values $x^1_i(C) = f^i_C - e^i_C - f^i_{\pi(i)} + e^i_{\pi(i)}$, while the values stored in the second vector depend on the type of the game: If condition (c2) of Lemma 16 holds, then $x^2_i$ stores $x^2_i(C) = 2f^i_C - e^i_C - 2f^i_{\pi(i)} + e^i_{\pi(i)}$ if preferences are friend-appreciating, and $x^2_i(C) = f^i_C - 2e^i_C - f^i_{\pi(i)} + 2e^i_{\pi(i)}$ if preferences are enemy-averse, while we set $x^2_i(C) = -\infty$ if (c2) fails. For balanced preferences, the vector $x^2_i$ is not used.

Notice that maintaining the values $x^1_i(C)$ and $x^2_i(C)$ for each agent $i$ and for *all coalitions* $C \in \pi \cup \{\emptyset\}$ for the current partition $\pi$ would suffice to determine whether there exists a switch $i \to C$ for some coalition $C \in \pi \cup \{\emptyset\}$ that locally dominates or locally blocks $\pi$, due to Lemma 16. However, it is not hard to observe that it suffices to focus for each agent $i \in N$ on the set $\mathcal{C}^i$ of coalitions in $\pi$ in which $i$ has at least one friend or enemy. Then it is sufficient to maintain the values $x^1_i(C)$ and $x^2_i(C)$ for each agent $i$ only for coalitions $C \in \mathcal{C}^i$, together with the values $f^{\pi(i)}_i$ and $e^{\pi(i)}_i$. Indeed, for every coalition $C \in \pi \cup \{\emptyset\} \setminus \mathcal{C}^i$ we know $e^C_i = f^C_i = 0$, and thus the values $x^1_i(C)$ and $x^2_i(C)$ can be calculated easily: first, we have $x^1_i(C) = -f^i_{\pi(i)} + e^i_{\pi(i)}$; second, to determine $x^2_i(C)$, we first check whether condition (c2) holds (using $e^C_i = f^C_i = 0$ again) and set $x^2_i(C) = -\infty$ if (c2) fails and, otherwise, set $x^2_i(C) = -2f^i_{\pi(i)} + e^i_{\pi(i)}$ in FEN-AF games and $x^2_i(C) = -f^i_{\pi(i)} + 2e^i_{\pi(i)}$ in FEN-AE games. Notice that the values $x^1_i(C)$ and $x^2_i(C)$ therefore are the same for each coalition $C$ in $\pi \cup \{\emptyset\} \setminus \mathcal{C}^i$.

Therefore, the vectors $x^1_i$ and $x^2_i$ are indexed by the set of all coalitions in $\mathcal{C}^i$ together with one dummy coalition representing all coalitions in $\pi \cup \{\emptyset\} \setminus \mathcal{C}^i$.

To facilitate an efficient search for a locally dominating or locally blocking switch, we keep the entries in the vectors $x^1_i$ and $x^2_i$ ordered. Furthermore, we additionally order the set $X^1 = \{x^1_i : i \in N\}$ of vectors according to their first coordinates, and order the vectors in $X^2 = \{x^2_i : i \in N\}$ similarly. Observe that the largest entry occurring in any vector of $X^1$ is therefore the first coordinate of the first vector in $X^1$ and, similarly, the largest entry occurring in any vector of $X^2$ is the first coordinate of the first vector in $X^2$. We will maintain this property throughout the algorithm; call it property ($\star$).

**The initialization.** To initialize $\mathcal{C}^i$ and the vectors $x^1_i$ and $x^2_i$ for the starting partition $\pi_0$, we start by computing the values $f^i_C$ and $e^i_C$ for all agents $i \in N$ and coalitions $C \in \pi_0$ in which $i$ has a friend or an enemy. This takes $\mathcal{O}(n\Delta)$ time in total for all agents using the friendship and enmity digraphs.

Next, we order the $\mathcal{O}(\Delta)$ values in each of the $2n$ vectors in $X^1 \cup X^2$; this takes $\mathcal{O}(n\Delta \log \Delta)$ time. Then we order the set $X^1 = \{x^1_i : i \in N\}$ of vectors according to their first coordinates, and order the vectors in $X^2 = \{x^2_i : i \in N\}$ similarly in $\mathcal{O}(n \log n)$ time. The required running time for the initializations is therefore $\mathcal{O}(n\Delta + n\Delta \log \Delta + n \log n) = \mathcal{O}(n\Delta \log n)$ as $1 \leq \Delta \leq n$.

**The computation of a step in the heuristic.** We check if the largest entry in the vectors in $X^1$ is at least 2 by looking at the first coordinate of the first vector in $\mathcal{O}(1)$ time. If yes, we have found a locally dominating switch $i \to C$; note that if the coalition achieving the largest enty in $X^1$ is a dummy coalition representing all coalitions in $\pi \cup \{\emptyset\} \setminus \mathcal{C}^i$ for some agent $i$, then a coalition $C \in \pi \cup \{\emptyset\} \setminus \mathcal{C}^i$ can be found in $\mathcal{O}(|\mathcal{C}^i|) = \mathcal{O}(\Delta)$ time.[2] To maintain the vectors $x_i^1$ and $x_i^2$ as well as the sets $\mathcal{C}^i$ for all agents $i \in N$, we only need to update these vectors and sets for the $\mathcal{O}(\Delta)$ agents in $\{i\} \cup E(i) \cup F(i)$ and, regarding the vectors, only at the coordinates corresponding to $\pi(i)$ and to $C$; note that this update may result in adding a new coordinate to the vector or deleting one from it. Then, in order to maintain property $(\star)$, we have to reorder $X^1$ and $X^2$ again—but we do not have to perform a complete ordering from scratch; we just have to find the correct places of the updated elements in the already existing ordering. Hence, this can be implemented in $\mathcal{O}(\Delta(\log \Delta + \log n))$ time, as there are $\mathcal{O}(\Delta)$ elements to update and an insertion of an element to an ordered list of length $\ell$ can be done in $\mathcal{O}(\log \ell)$ time.

If the largest entry in the vectors of $X^1$ is 0 or 1, then we check the largest entry in the vectors of $X^2$ (recall that $x_i^2$ stores a nonnegative entry for a coalition $C$ only if condition (c2) holds for $i$ and $C$) and check if the first coordinate is larger than 0 or not. If not, then $\pi$ is locally popular by Lemma 16. Otherwise, we again have found a locally dominating switch; we then perform this switch and update the sets $\mathcal{C}^i$ and the vectors in $X^1$ and $X^2$ in $\mathcal{O}(\Delta \log n)$ time as in the previous case.

Checking whether $i \to C$ is a Nash deviation for LocStab can also be implemented in the same running time. We just have to append the condition that $i$ improves to each case in Lemma 16, which is also a condition that can be checked in $\mathcal{O}(1)$ time. In fact, we can redefine the values $x_i^1(C)$ and $x_i^2(C)$ to be $-\infty$ if $i$ has no incentive to join $C$. This way, it remains true that the value of $x_i^1(C)$, and also of $x_i^2(C)$, coincides for all coalitions where $i$ has neither a friend nor an enemy; thus, our approach still works.

Summarizing all of the above, we get the running time of $\mathcal{O}(n\Delta \log n + K_{\#\text{steps}} \cdot \Delta \log n)$ where $K_{\#\text{steps}}$ is the number of steps (i.e., number of switches) performed by the algorithm. Combining this with Theorem 2 proves Theorem 3. $\qquad \square$

---

[2]We can take $C = \emptyset$ unless we aim for a variant of the algorithm that keeps the number of coalitions constant; in this case, we can iterate over the coalitions of the current partition until we find one that is *not* in $\mathcal{C}^i$ in $\mathcal{O}(\Delta)$ time.

# Appendix C  Additional material for the simulations

We remark that all codes used for our simulations are available in the supplementary material files of our submission. In Section C.1, we present our simulation result for community detection in the form of some figures; we do the same for clustering in Section C.2. Finally, we present a detailed evaluation of our results in Section C.3.

## C.1  Community detection: omitted figures

This section contains all figures depicting our simulation results for the Karate club (Figure 5), Jazz musicians (Figure 6), Cora (Figure 7), and Random-25 (Figure 8) datasets, comparing our heuristics LocPop and LocStab with the Louvain and Leiden algorithms in terms of Rand index and modularity. As the Jazz dataset has no true labels, we did not compute the Rand index for it.

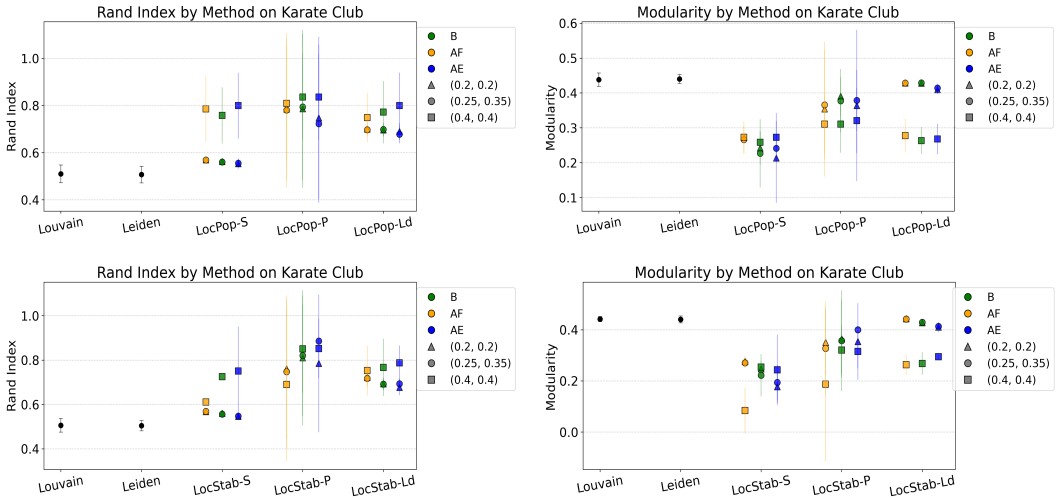

Figure 5: Comparing Rand index (left) and modularity (right) of Louvain, Leiden, and LocPop (top) or LocStab (bottom) variants for the Karate club dataset.

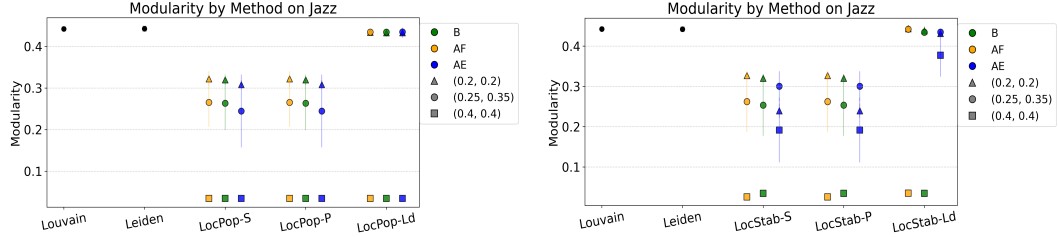

Figure 6: Comparing modularity of Louvain, Leiden, and LocPop (left) or LocStab (right) variants for the Jazz dataset.

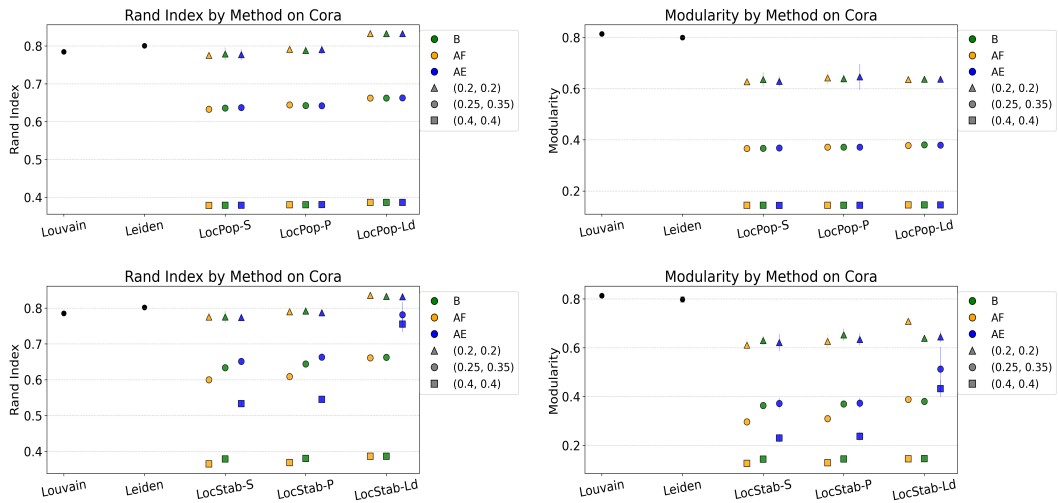

Figure 7: Comparing Rand index (left) and modularity (right) of Louvain, Leiden, and LocPop (top) or LocStab (bottom) variants for the Cora dataset.

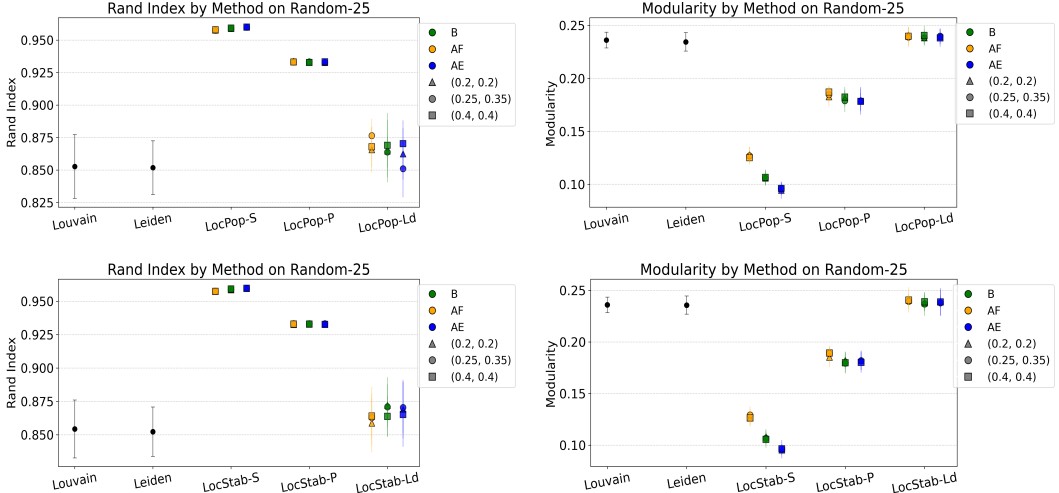

Figure 8: Comparing Rand index (left) and modularity (right) of Louvain, Leiden, and LocPop (top) or LocStab (bottom) variants for the Random-25 dataset.

## C.2 Clustering: omitted figures

This section contains all figures depicting our simulation results on the Iris (Figure 9), Breast Cancer Wisconsin (abbreviated as *Cancer*, Figure 10), Moons (Figure 11), and 3-Circles (Figure 12) datasets, comparing our heuristics LocPop and LocStab with the $k$-means and DBSCAN algorithms in terms of Rand index and silhouette score.

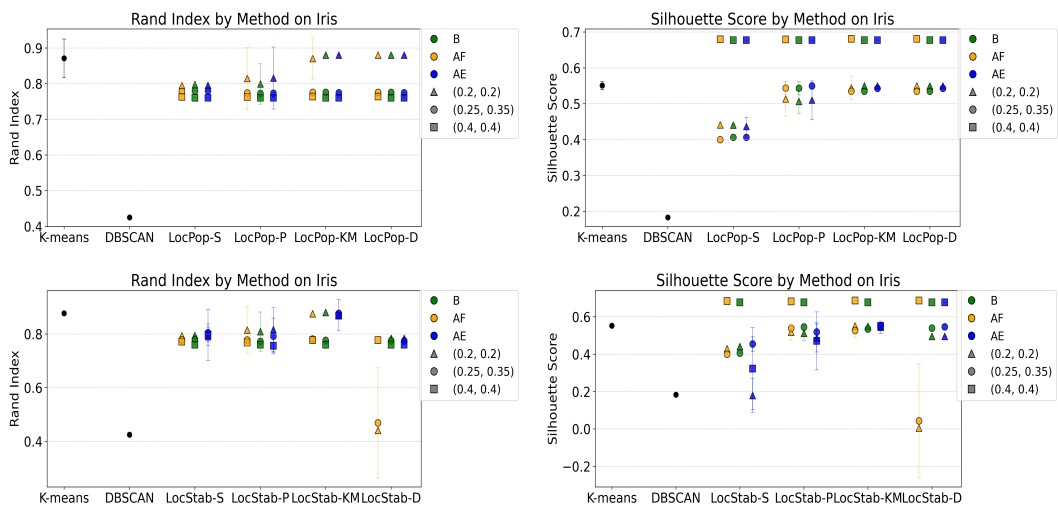

Figure 9: Comparing Rand index (left) and silhouette score (right) of $k$-means, DBSCAN and LocPop (top) or LocStab (bottom) variants for the Iris dataset.

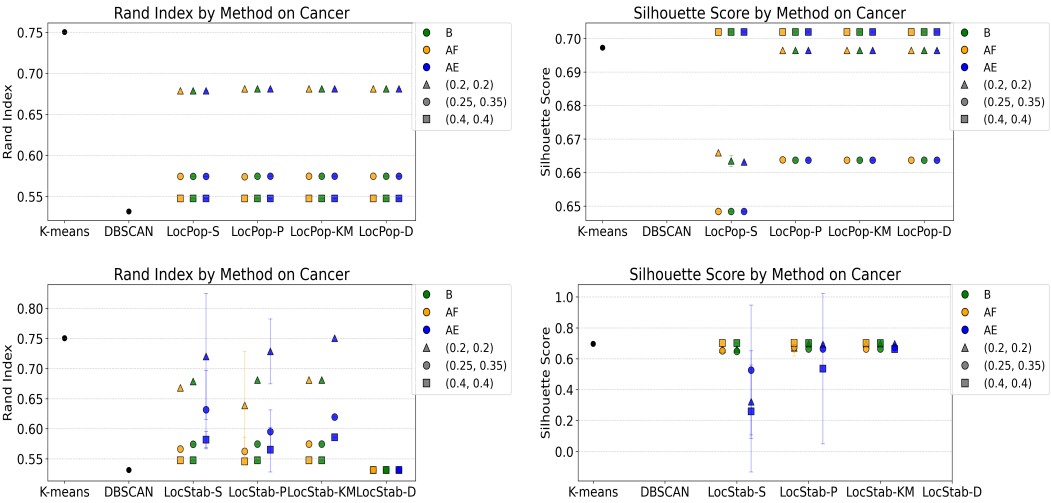

Figure 10: Comparing Rand index (left) and silhouette score (right) of $k$-means, DBSCAN and LocPop (top) or LocStab (bottom) variants for the Cancer dataset.

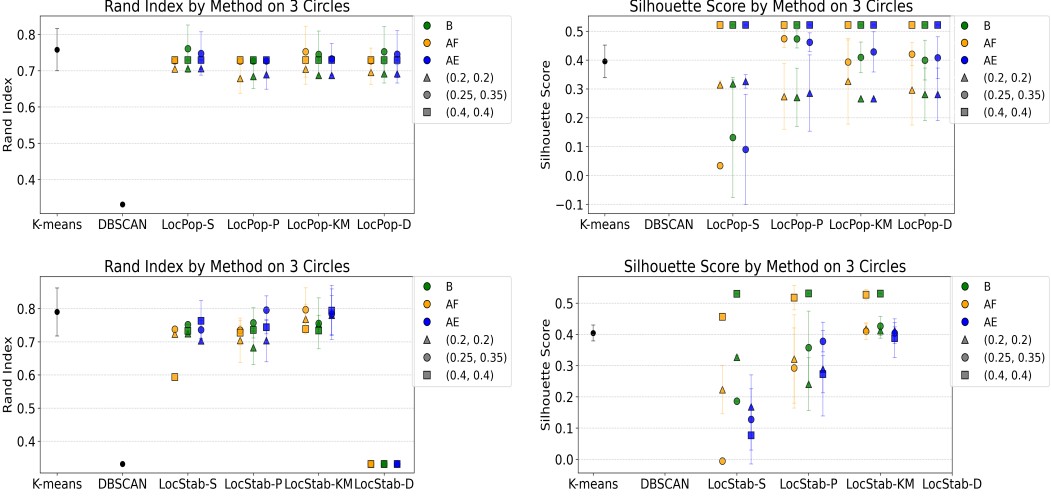

Figure 12: Comparing Rand index (left) and silhouette score (right) of $k$-means, DBSCAN and LocPop (top) or LocStab (bottom) variants for the 3-Circles dataset.

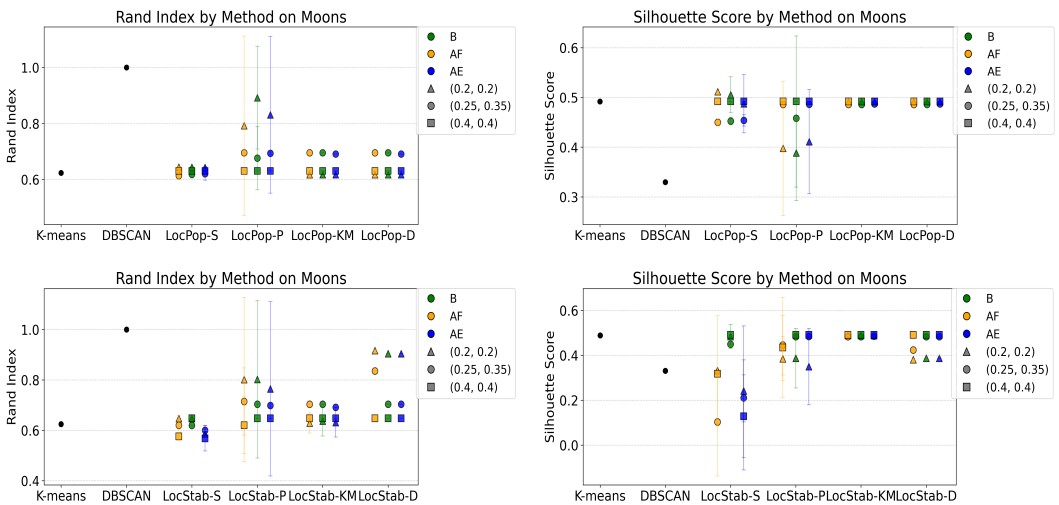

Figure 11: Comparing Rand index (left) and silhouette score (right) of $k$-means, DBSCAN and LocPop (top) or LocStab (bottom) variants for the Moons dataset.

## C.3 Evaluation of results

**General takeaways for LocPop and LocStab.** LocPop and LocStab produced very similar results, although the scores of LocStab were more dependent on the preference domains. The best threshold parameters were also mostly the same for LocPop and LocStab. In LocPop variants, the three different preferences domains (AF, AE, and B) lead to quite similar results across all instances. This is probably due to the fact that only the agent who switches may vote differently in these cases, and a single vote usually has no substantial influence. In the case of local stability, where a Nash deviation is needed, a larger differentiation was observed. The simulations also showed that—while no preference domain was consistently performing better than the two others—the *balanced* one performed best or close to the best for both LocPop and LocStab. Hence, in light of Theorems 3 and 4, using balanced preferences seems most preferable, as we have a faster runtime guarantee, and it is the only preference domain where the LocStab heuristic converges in polynomial time for asymmetric graphs. Hence, we suggest that future research focuses on the balanced preference domain.

In general, the -P and -S variants both performed well. However, the -S variant often requires much more steps to reach an outcome, which is expected, as it starts with $n$ clusters. Hence, if there is a good estimate (or even an upper bound) on the expected number of clusters, then the -P variant may be the optimal. The LocPop-P and LocStab-P variants we implemented also had a constraint that new coalitions could only be created if the current number of clusters was smaller than their predicted number.

In most cases, the $(0.4, 0.4)$ or the $(0.2, 0.2)$ parameters lead to the best results, but these parameters also often lead to the worst results; while the parameter $(0.25, 0.35)$ performed consistently between them. This suggests that parameter tuning can have a large effect on achieving the best possible outcome.

Since LocPop and LocStab are not deterministic (randomness comes from (i) the initial clustering, (ii) the indexing of the data points, and (iii) the choice of the implemented switches), we did 10 repetitions, each time randomly permuting the data points and with a random initial partition with the predicted number of clusters for LocPop-P and LocStab-P (for the other variants the initial partition is fixed) to estimate the mean and standard deviation of the scores. The algorithm was implemented in a way such that switches involving smaller-indexed agents were preferred, but by randomly permuting the indices, this bias was eliminated. The standard deviations observed were at most 0.01-0.02 on average after doing 10 repetitions, except for the Karate club dataset ($\sim 0.05$). Hence, for that dataset, we increased the number of repetitions to 30 for more robust results. We also note that in many cases, the standard deviations were too small to be visible in the figures (e.g., $< 10^{-3}$).

**Comparison with mainstream algorithms.** Let us compare the performance of our algorithms with the most widely used mainstream methods, both for community detection and for clustering.

**Community detection.** Using the output of Leiden, variants of LocPop-Ld (and LocStab-Ld) achieved a similar value in modularity, and slightly outperformed it w.r.t. the Rand index. Hence, starting from the output of Leiden was very beneficial in terms of achieving better modularity scores. For the Random-25 dataset, LocPop-S and LocPop-S were able to significantly outperform Louvain and Leiden (by more than $10\%$) w.r.t. the Rand index, but LocPop-P and LocPop-S were close to a $10\%$ improvement too. For the Karate club dataset, LocPop and LocStab variants were able to outperform Louvain and Leiden by more than $50\%$ in Rand index.

**Clustering.** In clustering, with the parameter $(0.4, 0.4)$, LocPop performed up to $20\%$ better than $k$-means w.r.t. the silhouette score in the Iris and 3 Circles datasets (and performed similarly for the others), while the other parameters led to (sometimes significantly) worse results. DBSCAN performed the worst in terms of the silhouette score, and it often put all points together, in which case the silhouette score is undefined. Regarding the Rand index, $(0.4, 0.4)$ turned out to be the worst parameter in general, which highlights that silhouette score may often not be the best measure to asses the quality of a clustering with regards to label prediction. This was most apparent in the Cancer dataset.

With respect to the Rand index, the best performing variants of LocPop and LocStab, which were the ones with the parameter $(0.2, 0.2)$ (except in the 3 Circles case) were within $10\%$ of $k$-means, often matching it, LocPop-P and LocStab-D outperformed it by roughly $50\%$ for the Moons dataset (LocStab-P being close behind them, but all variants performed better than $k$-means), where DBSCAN was able to find the true clusters completely. As opposed to community detection, for LocPop, here we found no clear benefit of using the output of DBSCAN or $k$-means to start with, while starting with a correctly predicted number of random clusters helped significantly for the Moons dataset. With LocStab-D, starting with these outputs did lead to better results for the Moons dataset.

Summarizing our observations, we conclude that the variants of LocPop and LocStab performed well and consistently, often leading to the best results in both community detection and clustering.

**Other observations, runtimes and outlook.** For heuristics LocPop and LocStab started with the outputs of Leiden and $k$-means, we tracked the Rand index between their initial partition (i.e., the output of Leiden or $k$-means, respectively) and the final partition. We observed that even in these cases, both algorithms often made quite a lot of changes. For LocPop, we observed an average Rand index—here used to measure the difference between the initial and the final partition—of $0.65$ with a minimum of $0.33$ across community detection datasets; for clustering datasets, these values reached an average of $0.55$ with a minimum of $0.34$. For LocStab, the obtained numbers were slightly higher, as expected. For community detection datasets, the average Rand index was $0.67$ with a minimum of $0.33$ and for clustering datasets, an average of $0.65$ with a minimum of $0.43$.

We also tracked the running times. Our LocPop algorithms run within a couple of seconds even on a mid-range hardware, except for Cora, the largest dataset, where computation took 3-5 minutes, LocPop-S and LocStab-S being the slowest as expected. For these variants, to speed up computation, we used random clusters of size 6, instead of singleton clusters, still modeling the same behavior. We emphasize that our implementation is not the most efficient one discussed in Section 3, but a simpler approach that iterates through the agents and the current clusters in each step. Furthermore, our approach to constructing the friendship and enmity graphs did not guarantee bounded degrees, causing a limitation, since then the running time in Theorem 3 are not guaranteed to be linear. For instances of even larger size, constructions that guarantee bounded degrees—e.g., taking the $d$ nearest neighbors as friends and the $d$ farthest ones as enemies for each agent—may lead to more efficient computation. We leave this for future work.

There are many more ways to generate friendship and enmity relations among agents, and it might be interesting to see whether there are approaches that lead to improved results. Further, another limitation is that LocPop might have a problem with instances containing clusters of different sizes, since large clusters can get rid of unwanted agents simply due to their voting power. It will be interesting to test this in future work. Finally, note that LocStab might be a fix for those instances, since agents cannot be pushed out of a cluster "against their will" (and maybe that is the reason why it performs better on the Cancer dataset).

# Appendix D  Omitted proofs

This section contains all proofs omitted from the main body of our paper, except for the proof of Theorem 3 which we proved in Appendix B. Sections D.1, D.2, and D.3 contain the omitted proofs of our results on FEN-AF, FEN-B, and FEN-AE games, respectively.

As already mentioned in Example 1, throughout this paper, all relations among agents that are not explicitly mentioned are tacitly assumed to be neutral.

## D.1  Omitted proofs for our results on FEN-AF games

Before we restate and prove our intractability results concerning FEN-AF games, we state some structural observations concerning AF-gadgets in Section D.1.1. We then proceed to provide all omitted proofs of our results on FEN-AF games in Sections D.1.2–D.1.4.

### D.1.1  A structural lemma on AF-gadgets

Given an AF-gadget $G$, let us define the *distance* of agents $i$ and $j$ within $G$ as the minimum number of arcs on a path from $i$ to $j$ or from $j$ to $i$ within the friendship graph of $G$. Moreover, whenever an AF-gadget is defined over agent set $K = \{0, 1, \ldots, 2k\}$, the agent $i + c$ is always interpreted as $(i + c) \bmod (2k + 1)$ for each agent $i \in K$ and constant integer $c$.

**Lemma 17.** *Consider a FEN-AF game that admits a locally stable partition $\pi$ and contains an AF-gadget $G$ over agent set $K$. If some coalition $C \in \pi$ with $C \subseteq K$ contains two agents whose distance in $G$ is at least 4, then $C = K$.*

*Proof.* Note that any two agents whose distance in $G$ is at least 4 are mutual enemies. Hence, if two such agents $i$ and $j$ belong to some coalition $C \subseteq K$, then each of them must have a friend in $C$: indeed, assuming that one of them, say agent $i$, does not have a friend in $C$, we get that the switch $i \to \emptyset$ locally blocks $\pi$, because $i$ and $i$'s enemies in $C$ vote for it, while only $i - 1$ may vote against it. The distance between the friend of $i$ and the friend of $j$ (i.e., agents $i + 1$ and $j + 1$) within $G$ is again at least 4, so by the same reasoning, each of them must have a friend within $C$. Repeating this argument, we get that every agent of $C$ must have a friend within $C$, which implies $K \subseteq C$. By $C \subseteq K$ we get $C = K$, as required. $\qquad\square$

### D.1.2  Proof of Theorem 5

**Theorem 5 ($\star$).** *The asymmetric FEN-AF game consisting solely of an AF-gadget admits no locally popular partition.*

*Proof.* Consider an AF-gadget $G$ of size $2k + 1$ containing agents $0, 1, \ldots, 2k$. Suppose for the sake of contradiction that $\pi$ is a locally popular partition.

If $\pi$ is the grand coalition, then for an arbitrary agent $i$, the switch $i \to \emptyset$ locally dominates $\pi$, because (recalling that $k \geq 5$) exactly $2k - 4 > 3$ agents vote for it (those who consider $i$ an enemy) and only the two agents $i$ and $i - 1$ vote against it (as they both lose a friend). This contradicts the local popularity of $\pi$.

Thus $\pi$ is not the grand coalition. By Lemma 17, this implies that for any two agents $i$ and $j$ with $\pi(i) = \pi(j)$, the distance of $i$ and $j$ in $G$ is at most 3, so each coalition in $\pi$ is a subset of $\{i, i + 1, i + 2, i + 3\}$ for some agent $i$.

Suppose now that some agents $i$ and $i + 2$ are both within some coalition $C = \pi(i) = \pi(i + 2)$, but $i + 1$ is not in $C$. Then $i + 1$ has an incentive to join $C$, that is, $i + 1$ votes for the switch $i + 1 \to C$, and so does $i$. Since the distance between any two agents of $C$ is at most 3 in $G$, there can be at most one agent in $C$ who considers $i + 1$ an enemy, namely agent $i - 1$. Thus at most one agent in $C$ votes against the switch $i + 1 \to C$, and note also that no agent in $\pi(i + 1)$ votes against it. Hence, $i + 1 \to C$ locally blocks $\pi$, a contradiction. This shows that each coalition in $\pi$ must either contain at most four agents who, additionally, appear consecutively on the friendship cycle of $G$, or it must be of the form $\{i, i + 3\}$ for some $i$.

If there is a coalition of the form $\{i, i + 3\}$, then the switch $i + 3 \to \pi(i + 4)$ locally blocks $\pi$, as both $i$ as $i + 3$ vote for it and the only agent who may vote against it is $i + 7$.

If there is a coalition containing four agents $\{i, i+1, i+2, i+3\}$, then again, the switch $i+3 \to \pi(i+4)$ locally blocks $\pi$: the only agents who may vote against it are $i + 2$ and $i + 7$, whereas agents $i$, $i + 1$, and $i + 3$ all vote for it.

Therefore, every coalition must contain at most three agents who appear consecutively on the friendship cycle of $G$. Suppose there is a coalition with exactly three agents. Then, as 3 does not divide $2k + 1$, there is one coalition, say $\{i, i+1, i+2\}$, which is followed (along the friendship cycle of $G$) by a coalition of size two, namely coalition $\{i + 3, i + 4\}$. Then the switch $i + 2 \to \pi(i + 3)$ locally blocks $\pi$, as agents $i$ and $i + 2$ vote for it, and only agent $i + 1$ votes against it.

Hence, each coalition contains at most two agents. If there is one containing exactly two agents, then there is one coalition, say $\{i, i + 1\}$, which is preceded (on the friendship cycle of $G$) by a coalition of size one, namely $\{i - 1\}$, as $2k + 1$ is odd. Hence, the switch $i - 1 \to \{i, i + 1\}$ locally blocks $\pi$. It follows that each coalition in $\pi$ must be a singleton. But then the switch where any agent $i$ joins $\{i+1\}$ locally blocks $\pi$, again a contradiction. It follows that no locally popular partition exists for this hedonic game. $\qquad\square$

Note that for the hedonic game defined in the proof of Theorem 5, there does exist a unique locally stable partition: the grand coalition containing all agents. In fact, the proof of Theorem 5 even implies the following slightly stronger claim.

**Remark 1.** Suppose that a hedonic game with friend-appreciative preferences contains an AF-gadget over agent set $K$, and admits a locally stable partition $\pi$. If no coalition of $\pi$ contains agents both in and outside of $K$, then $K$ itself is a coalition in $\pi$.

### D.1.3 Proof of Theorem 6

**Theorem 6** ($\star$). *There exists an asymmetric FEN-AF game that admits no locally stable partition.*

*Proof.* As a counterexample, consider a FEN-AF game $I$ that contains an AF-gadget over a set $K = \{0, 1, \ldots, 2k\}$ of agents together with an additional set $A = \{a_1, \ldots, a_{2k-4}\}$ of agents such that

- $a_1, \ldots, a_{2k-4}$ are mutual enemies with $1, 2, \ldots, 2k + 1$,

- $a_2, \ldots, a_{2k-4}$ consider 0 an enemy,

- $0$ considers $a_1$ a friend and $a_2, \ldots, a_{2k-4}$ enemies,

- $a_1$ and $a_2$ are mutual friends, and

- $a_i$ considers $a_{i+1}$ a friend for $i = 2, 3, \ldots, 2k - 5$.

Suppose that $\pi$ is a locally stable partition for $I$. First, we claim that if a coalition $C \in \pi$ contains both an agent from $K$ and an agent from $A$, then $C$ contains 0 as well. To see this, consider the "largest" agent $j = \max_{i:i \in K \cap C}$ of $K$ in $C$. If $0 \notin C$, then the switch $j \to \emptyset$ locally blocks $\pi$, because agent $j$ and all agents in $A \cap C \neq \emptyset$ vote for it, while at most one agent (namely, agent $j - 1$) may vote against it.

Next, we claim that if some coalition $C \in \pi$ contains both 0 and $a_i$ for some $i$, then $C$ also contains $a_{i+1}$ if it exists. For $i = 1$, this follows from the fact that, otherwise, $a_1$ has an incentive to join $\pi(a_2)$, as both $a_1$ and $a_2$ vote for the switch $a_1 \to \pi(a_2)$ and only agent 0 votes against it (note that only agents in $K$ consider $a_1$ an enemy, and we have $K \cap \pi(a_2) = \emptyset$ by the previous paragraph), contradicting our assumption that $\pi$ is locally stable. For $i > 1$, we can similarly show that if $a_{i+1} \notin C$, then the switch $a_i \to \pi(a_{i+1})$ locally blocks $\pi$: both $a_i$ and 0 vote for it, and only agent $a_{i-1}$ may vote against it.

As a consequence, no coalition $C$ in $\pi$ contains both an agent from $K$ and an agent from $A$. Indeed, assuming otherwise, the coalition $C$ would contain agents 0 and $a_{2k-4}$ by the previous two claims,

in which case the switch $a_{2k-4} \to \emptyset$ locally blocks $\pi$: both $a_{2k-4}$ and $0$ vote for it, while at most one agent (namely, agent $a_{2k-3}$) may vote against it.

Due to Remark 1, it follows that the agents of $K$ must form a single partition in $\pi$. In particular, agent $0$ has one friend and $2k - 4$ enemies within $\pi(0)$. Also, agent $0$ has at least one friend and at most $2k - 5$ enemies in the coalition $\pi(a_1)$, so $0$ has an incentive to join $\pi(a_1)$. Finally, the switch $0 \to \pi(a_1)$ locally blocks $\pi$, because $2k - 4 + 1$ agents from $K$ vote for it (namely, agent $0$ and all those regarding $0$ an enemy), while only one agent from $K$ (namely, agent $2k$) and $2k - 5$ agents from $A$ (all of them except $a_1$) vote against it. This contradiction shows that $\pi$ cannot be locally stable. □

### D.1.4 Proof of Theorem 7

**Theorem 7** ($\star$). LOCAL-POPULARITY-EXISTENCE *and* LOCAL-STABILITY-EXISTENCE *are* NP-*complete for FEN-AF games.*

It is clear that both problems are in NP, as given a FEN-AF game, we can nondeterministically guess a partition $\pi$ and then verify in polynomial time that no possible switch locally dominates or locally blocks $\pi$ (note that the number of switches for $\pi$ is at most quadratic in the number of agents). We complete the proof of Theorem 7 by proving the NP-hardness of LOCAL-STABILITY-EXISTENCE and LOCAL-POPULARITY-EXISTENCE separately in Theorems 18 and 24, starting with the former.

**Theorem 18.** LOCAL-STABILITY-EXISTENCE *is* NP-*hard for FEN-AF games.*

*Proof.* We provide a reduction from 5-SAT which, given a Boolean formula $\varphi$ in conjunctive normal form with five literals per clause, asks whether $\varphi$ is satisfiable. Let $\varphi$ be our input 5-SAT formula with variables $v_1, \ldots, v_n$ and clauses $C_1, \ldots, C_m$. By adding dummy clauses if needed, we may assume that $m \equiv 1 \bmod 3$ and that $m \equiv 1 \bmod 2$.

We create an instance $I$ of LOCAL-STABILITY-EXISTENCE as follows. Let $k = 5 \cdot 11m = 55m$; then $k \equiv 1 \bmod 2$ and $k \equiv 1 \bmod 3$. For each clause $C_j$, we create a *clause gadget* $G_j$ that is an AF-gadget of size $k$. The agents of $G_j$, ordered along the friendship cycle of $G_j$, are $x_j^0, \ldots, x_j^{55m-1}$. Every $11m$-th agent $x_j^0, x_j^{11m}, \ldots, x_j^{44m}$ along this cycle corresponds to one of the literals in the clause $C_j = (l_{j_1} \vee \cdots \vee l_{j_5})$; these are called *literal agents*. We also add a set $A_1, \ldots, A_{5m}$ of $5m$ *auxiliary gadgets*, each of them an AF-gadget of size $k = 55m$. Each auxiliary gadget $A_j$ has a single distinguished agent $a_j^0$; we refer to such agents as *enforcer agents*. We also add an agent $T$. Intuitively, in a locally stable partition, the coalition containing $T$ will also include a literal agent from each clause corresponding to a true literal. We proceed by describing the relationships between the agents on different cycles:

- Every literal agent considers $T$ a friend and is considered by $T$ an enemy.

- Every enforcer agent considers $T$ a friend but is considered by $T$ as a neutral.

- $T$ and every non-literal, non-enforcer agent consider each other mutual enemies.

- Any two agents $y$ and $z$ from two different clause gadgets are mutual enemies, unless $y$ and $z$ are both literal agents that do *not* correspond to a positive and a negative literal of the same variable, in which case they are mutually neutral.

- Any two agents $y$ and $z$ from two different auxiliary gadgets are mutual enemies, unless both are enforcer agents, in which case they are mutually neutral.

- Finally, for a vertex $x$ from a clause gadget and a vertex $y$ from an auxiliary gadget,

  - if $y$ is not a literal agent or $z$ is not an enforcer agent, then they are mutual enemies;
  - if $y$ is the literal agent corresponding to the $\ell$-th literal of the $j$-th clause, then $y$ and $z$ are mutual enemies if $z = a_{5(j-1)+\ell}^0$, and they are mutually neutral if $z$ is an enforcer agent other than $a_{5(j-1)+\ell}^0$.

This completes our construction of the instance $I$ of LOCAL-STABILITY-EXISTENCE from the given 5-SAT instance $\varphi$. It remains to prove the correctness of the reduction: $I$ is a yes-instance

of LOCAL-STABILITY-EXISTENCE if and only if $\varphi$ is satisfiable. This will follow from a series of claims.

**Direction "$\Rightarrow$":** Let us first assume that there exists a locally stable partition $\pi$ for $I$.

**Claim 19.** *Coalition $\pi(T)$ contains at most $20m - 1$ agents. Consequently, $\pi(T)$ cannot contain all agents from a clause or an auxiliary gadget.*

*Proof of claim.* For a contradiction, suppose that more than $20m - 1$ agents are in the same coalition as $T$. Then at least $10m$ of them are mutual enemies of $T$, as only $10m$ agents consider $T$ a friend, and no agent considers $T$ as neutral. Recall also that $T$ considers no agent as a friend and thus has an incentive to leave $\pi(T)$. Hence, the switch $T \to \emptyset$ locally blocks $\pi$, a contradiction. $\triangleleft$

**Claim 20.** *No coalition $C \in \pi$ can contain all agents from some clause or auxiliary gadget.*

*Proof of claim.* For the sake of contradiction, assume that some coalition $C$ contains some clause or auxiliary gadget $K$. Claim 19 implies $C \neq \pi(T)$. Let $y$ be a literal or enforcer agent in $K$. Since $y$ has one friend and exactly $55m - 5$ enemies within $K$, $y$ has an incentive to join $\pi(T)$ by Claim 19. Thus the switch $y \to \pi(T)$ locally blocks $\pi$: at least $(55m - 5) + 1$ agents—all enemies of $y$ within $K$ and $y$ itself—vote for it, whereas at most $(20m - 1) + 1$ agents—enemies of $y$ in $\pi(T)$ and the unique agent in $K$ who considers $y$ a friend—vote against it, a contradiction. $\triangleleft$

**Claim 21.** *If a coalition $C \in \pi$ with $T \notin C$ contains agents from two different clause or auxiliary gadgets, then $C$ cannot contain a non-literal, non-enforcer agent.*

*Proof of claim.* Suppose for the sake of contradiction that $C \in \pi$ with $T \notin C$ contains some agent $y$ from a (clause or auxiliary) gadget $K$, some agent $y'$ from a different gadget $K'$, and a non-literal, non-enforcer agent $w$. Note that neither $K$ nor $K'$ is fully contained in $C$ due to Claim 20. Therefore, using also $T \notin C$, we get that $C$ contains an agent $z \in K$ and an agent $z' \in K'$ such that neither $z$ nor $z'$ considers any agent in $C$ a friend. Moreover, $w$ is a mutual enemy with some $z'' \in \{z, z'\}$. This implies that the switch $z'' \to \emptyset$ locally blocks $\pi$, because both $z''$ and $w$ vote for it, while at most one agent (who considers $z''$ a friend) may vote against it, a contradiction. $\triangleleft$

**Claim 22.** *Coalition $\pi(T)$ contains at least one agent from each clause gadget and each auxiliary gadget.*

*Proof of claim.* For a contradiction, suppose that $\pi(T)$ contains no agent from some clause or auxiliary gadget $K$. If no agent from $K$ is contained in the same coalition as some agent from a different gadget, then Remark 1 implies $K \in \pi$, a contradiction to Claim 20. Hence, there is a coalition $C$ that contains some agent $y$ from $K$ and some agent from a different gadget. By Claim 21, we get that $y$ is either a literal or an enforcer agent. Suppose that $y$ is a literal agent $y = x_j^i$ for some clause gadget $K = G$; the case when $K$ is an auxiliary gadget and $y$ is an enforcer agent is analogous.

Note that $x_j^{i+1}$ is the only agent $x_j^i$ considers a friend, while the only agent who considers $x_j^i$ a friend is $G_j^{i-1}$. Since both $G_j^{i-1}$ and $x_j^{i+1}$ are non-literal, non-enforcer agents, neither of them is contained in $C$ due to Claim 21. In particular, $x_j^i$ has an incentive to join $\pi(x_j^{i+1})$. Since the switch $x_j^i \to \pi(x_j^{i+1})$ cannot locally block $\pi$, we obtain that there must be some agent in $\pi(x_j^{i+1})$ who considers $x_j^i$ an enemy. Observe that $T \notin \pi(x_j^{i+1})$ by our choice of $K = G_j$, so by Claim 21 we know that all agents of $\pi(x_j^{i+1})$ belong to $G_j$. Using Lemma 17 and Claim 20, we get that $\pi(x_j^{i+1})$ cannot contain two agents whose distance along $G_j$ is more than 3; we refer to this fact as ($\spadesuit$). This implies $i' \in \{i-2, i-1, i+1, i+2, i+3, i+4\}$ for every agent $x_j^{i'} \in \pi(x_j^{i+1})$, so the agent within $\pi(x_j^{i+1})$ who considers $x_j^i$ an enemy must be $x_j^{i-2}$ or $x_j^{i+4}$. Moreover, $\{x_j^{i-2}, x_j^{i+4}\} \not\subseteq \pi(x_j^{i+1})$ also follows. We distinguish between two cases and arrive at a contradiction in both, which proves the claim.

**Case A: $x_j^{i-2} \in \pi(x_j^{i+1})$.** Then $\pi(x_j^{i+1}) \subseteq \{x_j^{i-2}, x_j^{i-1}, x_j^{i+1}\}$ due to ($\spadesuit$). If $x_j^{i-1} \notin \pi(x_j^{i+1})$, then $x_j^{i+1}$ has an incentive to join $\pi(x_j^{i+2})$. Since the switch $x_j^{i+1} \to \pi(x_j^{i+2})$ does not locally block $\pi$ but both $x_j^{i+1}$ and $x_j^{i-2}$ vote for it, it follows that at least two agents must consider $x_j^{i+1}$ an enemy in $\pi(x_j^{i+2})$. However, due to ($\spadesuit$), these agents could only be $x_j^{i-1}$ and $x_j^{i+5}$, but they cannot both be

in $\pi(x_j^{i+2})$, a contradiction. Thus it must be the case that $\pi(x_j^{i+1}) = \{x_j^{i-2}, x_j^{i-1}, x_j^{i+1}\}$. Then $x_j^i$ has an incentive to join $\pi(x_j^{i+1})$; recall that $T \notin \pi(x_j^i)$, so $x_j^i$ has no friend in $\pi(x_j^i)$. Observe that the switch $x_j^i \to \pi(x_j^{i+1})$ locally blocks $\pi$: both $x_j^i$ and $x_j^{i-1}$ vote for it, but only $x_j^{i-2}$ votes against it, a contradiction.

**Case B: $x_j^{i+4} \in \pi(x_j^{i+1})$.** Then $\pi(x_j^{i+1}) \subseteq \{x_j^{i+1}, x_j^{i+2}, x_j^{i+3}, x_j^{i+4}\}$. Then $x_j^{i+4}$ has an incentive to join $\pi(x_j^{i+5})$, so both $x_j^{i+1}$ and $x_j^{i+4}$ vote for the switch $x_j^{i+4} \to \pi(x_j^{i+5})$. Since $\pi$ is locally stable, there must be two agents who vote against this switch. If $x_j^{i+3} \notin \pi(x_j^{i+1})$, then these must be two agents in $\pi(x_j^{i+5})$ who consider $x_j^{i+4}$ an enemy. Due to (♠), these can only be agents $x_j^{i+2}$ and $x_j^{i+8}$, but at most one of them can be in $\pi(x_j^{i+5})$, a contradiction. Hence, $x_j^{i+3} \in \pi(x_j^{i+1})$. Since the switch $x_j^{i+2} \to \pi(x_j^{i+1})$ does not locally block $\pi$, it follows that $x_j^{i+2} \in \pi(x_j^{i+1})$ as well, which yields $\{x_j^{i+1}, x_j^{i+2}, x_j^{i+3}, x_j^{i+4}\} \in \pi$. Then the switch $x_j^{i+4} \to \pi(x_j^{i+5})$ locally blocks $\pi$, because three agents (namely, $x_j^{i+1}$, $x_j^{i+2}$, and $x_j^{i+4}$) vote for it and at most two may vote against it ($x_j^{i+3}$ and possibly $x_j^{i+8}$), a contradiction. ◁

**Claim 23.** *Coalition $\pi(T)$ contains no non-literal, non-enforcer agents. Also, each literal agent in $\pi(T)$ can have at most one mutual enemy in $\pi(T)$, which is necessarily an enforcer agent.*

*Proof of claim.* For the sake of contradiction, assume that $y \neq T$ is a non-literal, non-enforcer agent in $\pi(T)$. Take an agent $z \in \pi(T)$ from a clause $G_j$ not containing $y$ such that the unique friend of $z$ within $K$ is not in $\pi(T)$. By Claims 22 and 19, there must be such an agent $z$. Note that $y$ and $z$ are mutual enemies. If $z$ is not a literal agent, then $z$ has no friend in $\pi(T)$, so the switch $z \to \emptyset$ locally blocks $\pi$, a contradiction. If $z = x_j^i$ is a literal agent, then $z$ is mutual enemies with every agent from at least one auxiliary gadget by construction, so $z$ has at least two mutual enemies in $\pi(T)$.

It now suffices to show that the assumption that $z$ has at least two mutual enemies in $\pi(T)$ leads to a contradiction.

Assume first that $x_j^i$ has an incentive to join $\pi(x_j^{i+1})$. Then $x_j^i$, $T$, and the two mutual enemies of $x_j^i$ in $\pi(T)$ vote for the switch $x_j^i \to \pi(x_j^{i+1})$ and at most one agent in $\pi(T)$ votes against it. Moreover, due to Lemma 17 and Claim 20, each two agents in $\pi(x_j^{i+1})$ are at distance at most 3 from each other along $G_j$, and thus the only agents in $\pi(x_j^{i+1})$ who consider $x_j^i$ an enemy can be $x_j^{i-2}$ and $x_j^{i+4}$, so fewer than four agents vote against the switch, a contradiction.

It follows that $x_j^i$ has no incentive to join $\pi(x_j^{i+1})$, which happens only if $x_j^i$ regards at least two agents in $\pi(x_j^{i+1})$ as enemies. This implies that $\pi(x_j^{i+1})$ contains at least two agents from $\{x_j^{i+2}, x_j^{i+3}, x_j^{i+4}\}$. As $\pi$ is locally stable, it follows that $\{x_j^{i+2}, x_j^{i+3}\} \subseteq \pi(x_j^{i+1})$. Suppose that $x_j^{i+4} \in \pi(x_j^{i+1})$ as well. Then the switch $x_j^{i+4} \to \pi(x_j^{i+5})$ locally blocks $\pi$, because $x_j^{i+4}$, $x_j^{i+2}$, and $x_j^{i+1}$ all vote for it, whereas only $x_j^{i+3}$ and $x_j^{i+8}$ may vote against it. Otherwise, $\pi(x_j^{i+1}) = \{x_j^{i+1}, x_j^{i+2}, x_j^{i+3}\}$. The switch $x_j^{i+3} \to \pi(x_j^{i+4})$ locally blocks $\pi$ unless $\pi(x_j^{i+4}) = \{x_j^{i+5}, x_j^{i+6}, x_j^{i+7}, x_j^{i+8}\}$. In this case, however, the switch $x_j^{i+8} \to \pi(x_j^{i+9})$ locally blocks $\pi$ by the argument used in the previous case. ◁

We are now ready to prove that $\varphi$ is satisfiable.

By Claim 22, at least one literal agent from each clause gadget $G_j$ must be contained in $\pi(T)$. We claim that the corresponding literals can be set to true at the same time. For a contradiction, suppose that there are two literal agents, $y$ and $z$, in $\pi(T)$ such that one of them corresponds to literal $v_i$ and the other one to its negation $\overline{v}_i$. By Claim 23, $a_j^0 \in \pi(T)$ for any auxiliary gadget $A_j$, which means that every literal agent is mutual enemies with at least one enforcer agent in $\pi(T)$. Since $y$ and $z$ are also mutual enemies by construction, each of these two literal agents have at least two mutual enemies, which is a contradiction to Claim 23. Hence, $\varphi$ is satisfiable.

**Direction "⇐":** Let us now assume that $\varphi$ is satisfiable.

Consider a truth assignment that makes $\varphi$ true. Create a partition $\pi$ as follows. For each auxiliary gadget $A_j$, group $s_j^0$ together with $T$. Also, from each clause gadget, choose a literal agent $x_j^i$

corresponding to a true literal and group it together with $T$, too. Finally, for each clause or auxiliary gadget $K$, create coalitions of size two from the agents of $K$ not put into $\pi(T)$ such that each of these coalitions contains two agents appearing consecutively on $K$.

We claim that $\pi$ is locally stable. Agent $T$ has an incentive to join every coalition $C$ in $\pi \cup \{\emptyset\}$ other than $\pi(T)$. However, at least $6m$ agents vote against such a switch $T \to C$ and at most one votes for it. Note that a literal or enforcer agent in $\pi(T)$ has one friend in $\pi(T)$ and at most one enemy: Indeed, each enforcer agent is mutual enemies with exactly one literal agent—which may or may not be in $\pi(T)$—and vice versa; moreover, two literal agents are mutual enemies only if they correspond to a literal $v_i$ and its negation $\overline{v}_i$, respectively. This, however, cannot happen, as only literal agents corresponding to true literals are in $\pi(T)$. Since each literal or enforcer agent has at most one friend in every coalition $C \neq \pi(T)$, and if they do have a friend in $C$ then they also have an enemy in $C$, so they have no incentive to join $C$.

For a non-literal, non-enforcer agent $y$, let $\{y, y'\}$ denote the coalition $\pi(y)$. On the one hand, if $y$ regards $y'$ as a friend, then $y$ has no incentive to join any coalition in $\pi \cup \{\emptyset\}$. If, on the other hand, $y'$ regards $y$ as a friend, then $y$ has an incentive to join the unique coalition containing its sole friend $y''$, but the switch $y \to \pi(y'')$ does not locally block $\pi$, because only $y$ votes for it but $y'$ votes against it.

We conclude that $\pi$ is locally stable, proving the correctness of our reduction. $\qquad\square$

We now turn to the existence problem for local popularity and its computational complexity.

**Theorem 24.** LOCAL-POPULARITY-EXISTENCE *is* NP-*hard for FEN-AF games.*

*Proof.* We again provide a reduction from 5-SAT by slightly modifying the reduction presented in the proof of Theorem 18.

Let $\varphi$ be a 5-SAT formula with variables $v_1, \ldots, v_n$ and clauses $C_1, \ldots, C_m$. By adding dummy clauses if needed, we may again assume that $m \equiv 1 \bmod 3$ and $m \equiv 1 \bmod 2$.

From $\varphi$ we create an instance $I$ of LOCAL-POPULARITY-EXISTENCE as follows. Again, let us set $k = 5 \cdot 11m = 55m$; then $k \equiv 1 \bmod 2$ and $k \equiv 1 \bmod 3$. For each clause $C_j$, we create a *clause gadget* $G_j$ that is an AF-gadget of size $k$. The agents of $G_j$ are again referred to as $x_j^0, \ldots, x_j^{55m-1}$, ordered along the friendship cycle of $G_j$, and every $11m$-th agent $x_j^0, x_j^{11m}, \ldots, x_j^{44m}$ along this cycle corresponds to one of the literals in $C_j = (l_{j_1} \vee \cdots \vee l_{j_5})$. These are again called *literal agents*. Note that we do not create the auxiliary gadgets and enforcer agents used in the proof of Theorem 18, but we do add an agent $T$. As before, a locally popular partition will always group $T$ together with an agent corresponding to a true literal from each clause. We proceed by describing the relationships between agents on different gadgets:

- Every literal agent considers $T$ a friend, whereas $T$ considers everyone an enemy and is considered an enemy by every non-literal agent.

- Furthermore, two agents $y$ and $z$ from two different clause gadgets are mutual enemies, unless $y$ and $z$ are both literal agents that do *not* correspond to a positive and a negative literal of the same variable, in which case they are mutually neutral.

We are going to show that the constructed instance $I$ of LOCAL-POPULARITY-EXISTENCE admits a locally popular partition if and only if $\varphi$ is satisfiable.

**Direction "$\Rightarrow$":** Let us first assume that there exists a locally popular partition $\pi$ for $I$.

**Claim 25.** *No coalition $C \in \pi$ may contain all agents from some clause gadget.*

*Proof of claim.* For a contradiction, suppose that a coalition in $\pi$ contains all agents of a clause gadget $G_j$. Then the switch $x_j^i \to \emptyset$ for an arbitrary agent $x_j^i$ in $G_j$ locally dominates $\pi$, because at least $55m - 5$ agents vote for such a switch and at most one agent votes against it, a contradiction. $\lhd$

**Claim 26.** *If a coalition $C \in \pi$ with $T \notin C$ contains agents from two different clause gadgets, then $C$ cannot contain a non-literal agent.*

*Proof of claim.* Exactly the same arguments as used in the proof of Claim 21 prove the claim. $\lhd$

**Claim 27.** *Coalition $\pi(T)$ contains at least one literal agent from each clause gadget.*

*Proof of claim.* For a contradiction, suppose that there is a clause gadget $G_j$ that contains no agent in $\pi(T)$. If no agent of $G_j$ is in the same coalition as some agent from a different gadget, then $\pi$ cannot be locally popular, because we have shown in Theorem 5 that an AF-gadget does not admit a locally popular partition.

By Claim 26, it follows that there must be a literal agent $x_j^i$ in $G_j$ such that $C = \pi(x_j^i)$ contains only literal agents, with at least one of them not in $G_j$. By $T \notin C$, we know that $x_j^i$ has an incentive to join $\pi(x_j^{i+1})$. Hence, as no agent considers $x_j^i$ a friend in $C$, there must be an agent in $\pi(x_j^{i+1})$ who considers $x_j^i$ an enemy, as otherwise, the switch $x_j^i \to \pi(x_j^{i+1})$ locally blocks $\pi$. Note that $T \notin \pi(x_j^{i+1})$ by our choice of $G_j$, and since $x_j^{i+1}$ is a non-literal agent, we know that $\pi(x_j^{i+1}) \subseteq G_j$ by Claim 26. Therefore, by Lemma 17 and Claim 25, we know that all agents in $\pi(x_j^{i+1})$ are at distance at most 3 from $x_j^{i+1}$ along $G_j$. Hence, the agent in $\pi(x_j^{i+1})$ who considers $x_j^i$ an enemy must be $x_j^{i-2}$ or $x_j^{i-2}$, and they cannot both be contained in $\pi(x_j^{i+1})$. Distinguishing between the cases $x_j^{i-2} \in \pi(x_j^{i+1})$ and $x_j^{i+4} \in \pi(x_j^{i+1})$ and using exactly the same arguments as in Cases A and B in the proof of Theorem 18 we obtain a contradiction to the local stability and, hence, to the local popularity of $\pi$. ◁

We are now ready to prove that $\varphi$ is satisfiable.

By Claim 27, $\pi(T)$ contains at least one literal agent from each clause gadget. We claim that the corresponding literals can be made true at the same time. For a contradiction, suppose that there are two literal agents, $y$ and $z$, corresponding to some variable $v_i$ and its negation $\overline{v}_i$, respectively, that are both in $\pi(T)$. Then $y$ and $z$ are mutual enemies. Hence, the switch $y \to \emptyset$ locally dominates $\pi$, as both $T$ and $z$ vote for it and only $y$ votes against it, a contradiction. Hence, this yields a truth assignment satisfying $\varphi$.

**Direction "$\Leftarrow$":** Assume now that there is a truth assignment satisfying $\varphi$.

Create a partition $\pi$ as follows. For each clause gadget $G_j$, choose a literal agent corresponding to a literal set to true and group it with $T$. Then for each clause gadget $K$, create coalitions of size two from the agents of $K$ not put into $\pi(T)$ such that each of these coalitions contains two agents appearing consecutively on $K$.

We claim that $\pi$ is locally popular. Let $C \in \pi \cup \{\emptyset\}$. Note that the only agent who may vote for a switch $T \to C$ is $T$, while at least $m$ agents vote against it. For a literal agent $x_j^i \in \pi(T)$, the only agents who vote for a switch $x_j^i \to C$ can be $T$ and possibly agent $x_j^{i-1}$, while $x_j^i$ itself votes against it, and moreover, if $x_j^{i-1} \in C$ votes for the switch, then $x_j^{i-2} \in C$ votes against it. Hence, $x_j^i \to C$ does not locally dominate $\pi$. For an agent $y$ in a coalition $\{x_j^i, x_j^{i+1}\} \in \pi$ of size two, only one agent may vote for the switch $y \to C$, and this happens only if either $y = x_j^{i+1}$ and $y$ has a friend in $C$, in which case $x_j^i$ votes against the switch, or if $y = x_j^i$ and $C = \{x_j^{i-2}, x_j^{i-1}\}$, in which case both $x_j^{i-2}$ and $x_j^i$ vote against the switch. Hence, no switch locally dominates $\pi$, so $\pi$ is locally popular, which completes the proof of correctness for our reduction. □

## D.2 Omitted proofs for our results on FEN-B games

We start with restating and proving Theorem 4, stating the polynomiality of LocStab on FEN-B games, in Section D.2.1. We move on with some structural observations concerning B-gadgets in Section D.2.2. We then proceed to provide all omitted proofs of our results on FEN-B games in Sections D.2.3–D.2.6.

### D.2.1 Polynomiality of LocStab

**Theorem 4** ($\star$)**.** *In a possibly asymmetric FEN-B game, the total balanced utility of all agents strictly increases in each step of heuristic* LocStab*; consequently, heuristic* LocStab *always converges in $\mathcal{O}(n\Delta)$ steps to a locally stable partition, irrespective of the initial partition.*

*Proof.* Let $N$ denote the set of agents. Consider a step of the heuristic where a partition $\pi$ is replaced by $\pi_{i\to C}$ for some agent $i$ and coalition $C \in \pi \cup \{\emptyset\}$ is such that $C \cup \{i\} \succ_i \pi(i)$ and $\Lambda(\pi, \pi_{i\to C}) > 0$. As $\Lambda(\pi, \pi_{i\to C}) > 0$ we have $\Lambda_{-i}(\pi, \pi_{i\to C}) \geq 0$. So, by statement (a) of Lemma 1,

$$\sum_{j\in N\setminus\{i\}} u_j^{\text{B}}(\pi_{i\to C}) - u_j^{\text{B}}(\pi) = \Lambda_{-i}(\pi, \pi_{i\to C}) \geq 0.$$

Finally, $u_i^{\text{B}}(C) - u_i^{\text{B}}(\pi(i)) > 0$ because the switch is performed by heuristic LocStab based on the condition that it is a Nash deviation, i.e., $i$ has an incentive to join $C$ (recall that preferences are balanced). Hence, $f(\pi_{i\to C}) - f(\pi) = \sum_{j\in N} u_j^{\text{B}}(\pi_{i\to C}) - u_j^{\text{B}}(\pi) > 0$, which proves the first statement of the theorem. Since the total balanced utility is between $-n\Delta$ and $n\Delta$, the claimed bound on the number of steps follows as well. $\square$

### D.2.2    A structural lemma on B-gadgets

Let us say that a coalition $C$ in a B-gadget over agent set $K = \{0, 1, \ldots, 6\}$ is *consecutive* if $C = \{i, i+1, \ldots, i+\ell\}$ for some integer $\ell \in [6]$.

**Lemma 28.** *Let $\pi$ be a partition for a B-gadget, and let $C \in \pi$ be a coalition with either $|C| \geq 4$, or $|C| = 3$ and $C$ is not consecutive. Then $i \to \emptyset$ locally dominates $\pi$ for some agent $i \in C$.*

*Proof.* Let $\pi$ be a partition for the B-gadget over agent set $K = \{0, 1, \ldots, 6\}$ and let $C \in \pi$.

If $C$ is the grand coalition, then $i \to \emptyset$ locally dominates $\pi$, because $i$, $i+2$, $i+3$, and $i+4$ all vote for it, while only $i-1$ and $i-2$ vote against it.

If $4 \leq |C| \leq 6$, then let $i \in C$ be an agent such that $i - 1 \notin C$ and, if $C$ is not consecutive, then $\{i, i+1, i+2\} \not\subseteq C$; observe that such an agent $i$ exists. Consider the switch $i \to \emptyset$. If $i - 2 \notin C$, then at least two agents in $\{i+2, i+3, i+4\} \cap C$ vote for this switch, while $i+1$ is indifferent between $\pi$ and $\pi_{i\to\emptyset}$. Hence, even if $i$ votes against it, we get $\Lambda(\pi, \pi_{i\to\emptyset}) \geq 1$. If $i - 2 \in C$ and $C$ is consecutive, then $C = K \setminus \{i-1\}$, in which case agents $i$, $i+2$, $i+3$, and $i+4$ all vote for the switch $i \to \emptyset$, and only $i - 2$ votes against it. If $i - 2 \in C$ but $C$ is not consecutive, then due to our choice of $i$, we know that $C$ does not contain both $i+1$ and $i+2$, and thus $f_C^i = 1$, implying also $e_C^i \geq 2$ (because $|C| \geq 4$ and $i - 1 \notin C$). Hence, $i$ has an incentive to join $\emptyset$. Moreover, at least one agent in $\{i+2, i+3, i+4\} \cap C$ votes for $i \to \emptyset$, while only $i-2$ votes against it. Hence, we get $\Lambda(\pi, \pi_{i\to\emptyset}) \geq 1$ again, which means that $i \to \emptyset$ locally dominates $\pi$, as required.

It remains to consider the case when $|C| = 3$ and $C$ is not consecutive. If $\{i, j, j+1\} \subseteq C$ for some agent $j$, then $j \notin \{i-2, i-1, i+1\}$. If $j \in \{i+2, i+3\}$, then both $j$ and $j+1$ regard $i$ as an enemy, and thus they both vote for the switch $i \to \emptyset$. If $j = i + 4$, then both $i$ and $i+4$ vote for the switch $i \to \emptyset$. Hence, we get $\Lambda(\pi, \pi_{i\to\emptyset}) \geq 1$ in both cases. Finally, if $\{j, j+1\} \not\subseteq C$ for any agent $j \in K$, then $C$ must be of the form $\{i, i+2, i+4\}$ for some $i \in K$. Then both $i+2$ and $i+4$ vote for $i \to \emptyset$, yielding again a switch that locally dominates $\pi$. $\square$

### D.2.3    Proof of Theorem 8

**Theorem 8** ($\star$)**.** *The asymmetric FEN-B game consisting solely of a B-gadget admits no locally popular partition.*

*Proof.* Assume for the sake of contradiction that $\pi$ is a locally popular partition for the B-gadget over agent set $K = \{0, 1, \ldots, 6\}$. By Lemma 28 we know that $\pi$ contains no coalition of size more than 3, and moreover, all coalitions of size 3 must be consecutive. If $\pi$ contains a nonconsecutive coalition $C$ of size 2, then $C$ must be of the form $\{i, i+2\}$, as otherwise, its two agents are mutual enemies, and thus either of them switching to $\emptyset$ locally dominates $\pi$. We may further assume that $i - 1 \notin \pi(i+1)$ (as otherwise, we can pick the coalition $\{i-1, i+1\}$ instead of $C$); hence, either $\{i+1\}$ is a singleton in $\pi$ or $\{i+1, i+3\} \in \pi$. Consider the switch $i+2 \to \pi(i+1)$: both $i+1$ and $i+2$ vote for it, while only $i$ votes against it (note that $i+3$ is indifferent between $\pi$ and $\pi_{i+2\to\pi(i+1)}$). Hence, this switch locally dominates $\pi$, a contradiction that shows that all coalitions in $\pi$ must be consecutive.

If $C = \{i, i+1\} \in \pi$, then both $i$ and $i+1$ vote for the switch $i+2 \to C$, and only $i+2$ (but neither $i+3$ nor $i+4$) may vote against it, contradicting again the local popularity of $\pi$. Thus $\pi$ can

only contain singletons and (consecutive) coalitions of size 3. If $i$ and $i + 1$ both form singletons in $\pi$, then $i$ has an incentive to join $\{i + 1\}$, and $i + 1$ does not vote against it, so $i \to \{i + 1\}$ locally dominates $\pi$. Hence, the only remaining possibility is that $\pi$ contains a singleton and two coalitions of size 3, say $\{i - 3, i - 2, i - 1\}$ and $\{i, i + 1, i + 2\}$ for some agent $i$. However, in this case the switch $i \to \pi(i - 1)$ locally dominates $\pi$, because agents $i - 1$, $i - 2$, and $i + 2$ vote for it, while only $i$ and $i - 3$ vote against it. This contradiction completes the proof. $\quad\square$

### D.2.4 Proof of Theorem 9

Before proving Theorem 9, we start with a simple lemma that gives the bases of our reductions for Theorems 9 and 11.

**Lemma 29.** *Consider a FEN-B game $I$ containing three* special *agents, $s_1$, $s_2$, and $s_3$, a set $D_1 \cup D_2 \cup D_3$ of dummy agents, and a set $A$ of vertex agents with $|D_1| = |D_2| = |D_3| = 3|A|$. For each $i \in [3]$, let $D_i^+ = D_i \cup \{s_i\}$ and let the friendship and enmity relations in $I$ be such that*

- *every agent in $D_i^+$ regards all other agents in $D_i^+$ a friend and regards every remaining special or dummy vertex a friend, and*

- *vertex agents are mutually neutral with every dummy agent.*

*Then every locally stable partition $\pi$ contains three coalitions, $C_1$, $C_2$, and $C_3$, such that $D_i^+ \subseteq C_i$ for each $i \in [3]$.*

*Proof.* First, assume that there exists a coalition $C \in \pi$ such that $C$ contains agents both from $D_i^+$ and $D_j^+$ for some indices $i \neq j$; we will call such coalitions *mixed*. Pick an index $i$ such that $|D_i^+ \cap C| \leq |D_j^+ \cap C|$. If $D_i^+ \cap C$ contains a dummy agent $d$, then the switch $d \to \emptyset$ locally blocks $\pi$, because all agents in $D_j^+$ as well as $d$ vote for it, while only the agents in $D_i^+ \setminus \{d\}$ vote against it; a contradiction. Hence, we know that $C \cap D_i^+ = \{s_i\}$. In this case we must have $|C \cap D_j^+| \leq |A|$, as otherwise, $s_i \to \emptyset$ locally blocks $\pi$, because only vertex agents in $C$ may vote against it, while all agents in $D_j^+$ as well as $s_i$ vote for it. Thus each mixed coalition $C$ contains dummy agents from at most one set $D_j$ among $D_1$, $D_2$, and $D_3$ (and then must contain a special vertex other than $s_j$), and moreover, a mixed coalition can contain at most $|A|$ agents from $D_j^+$.

Fix some index $j \in [3]$, and assume now that there are coalitions $C$ and $C'$ in $\pi$ such that both contain agents from $D_j$, $|C \cap D_j^+| \leq |C' \cap D_j^+|$, and $C'$ is non-mixed. Let us pick an arbitrary agent $d \in C \cap D_j^+$. Then the switch $d \to C'$ locally blocks $\pi$, as all agents in $(C' \cap D_j^+) \cup \{d\}$ vote for it, while the only agents who vote against it are those in $(C' \cap D_j^+) \setminus \{d\}$ because $C'$ is non-mixed. This contradiction shows that there is at most one non-mixed coalition that contains agents from $D_j$, and moreover, it contains fewer agents from $D_j^+$ than any mixed coalition containing an agent of $D_j$.

If $\pi$ contains at least one mixed coalition $C$ with $C \cap D_j \neq \emptyset$, then the unique (or possibly nonexistent) non-mixed coalition that contains agents from $D_j$ must contain fewer than $|A|$ agents from $D_j^+$. However, since there can be at most two mixed coalitions containing agents of $D_j$ (as each of them must contain a special agent other than $s_j$), we obtain that a total of at most $3|A| - 1$ agents of $D_j$ can be contained in coalitions of $\pi$, contradicting $|D_j| = 3|A|$. This proves that there are no mixed coalitions containing agents of $D_j$ and hence all agents of $D_j$ must be contained in a unique non-mixed coalition $C_j$. Note also that $s_j \in C_j$ follows, as otherwise, $s_j \to C_j$ locally blocks $\pi$. This proves the claim. $\quad\square$

**Theorem 9** $(\star)$. LOCAL-POPULARITY-EXISTENCE *is NP-complete for FEN-B games.*

*Proof.* It is clear that this problem is in NP, as given a FEN-B game, we can nondeterministically guess a partition $\pi$ and then verify in polynomial time that no possible switch locally dominates $\pi$ (note again that the number of switches for $\pi$ is at most quadratic in the number of agents).

To show NP-hardness, we reduce from the NP-hard 3-COLORING problem: given an undirected graph $G = (V, E)$, the task is to decide whether $G$ admits a coloring $\chi : V \to [3]$ that is *proper*, i.e.,

there is no edge $\{u, v\} \in E$ with $\chi(u) = \chi(v)$. We will call a proper coloring *tight* if each vertex has two neighbors whose colors differ from each other.

Given an instance $G = (V, E)$ of 3-COLORING, we start by creating a graph $H$ as follows: for each vertex $v \in V$, we add new vertices $v^1$, $v^2$, and $v^3$ to $G$ forming a triangle, and we connect $v^1$ and $v^2$ to $v$ by an edge. Observe that a proper coloring of $G$ can be extended into a tight coloring of $H$ and, in fact, every proper coloring of $H$ is tight. Let $H = (V', E')$ denote the constructed graph and let $n = |V'|$.

Next, we create a FEN-B game $I$ based on $H$ as follows. We introduce a set $D_1 \cup D_2 \cup D_3$ of *dummy agents* with $|D_1| = |D_2| = |D_3| = 21n$, three *special agents*, $s_1$, $s_2$, and $s_3$, and for each vertex $v \in V'$, a set $A_v = \{v_i : i \in \{0, 1, \ldots, 6\}\}$ of seven *vertex agents* who form a B-gadget. We will use the notation $D_i^+ = D_i \cup \{s_i\}$ for each $i \in [3]$, so the set of agents is $N = \bigcup_{i \in [3]} D_i^+ \cup \bigcup_{v \in V'} A_v$.

The remaining relationships between the agents are as follows:

- For each $i \in [3]$, every agent in $D_i^+$ considers each other agent in $D_i^+$ a friend and every remaining special or dummy agent an enemy.

- Two vertex agents, $u_i \in A_u$ and $v_j \in A_v$, are mutual enemies if and only if $\{u_i, v_j\} \in E'$; otherwise, they are mutually neutral.

- Each vertex agent is mutual friends with every special agent and mutually neutral with every dummy agent.

This completes our construction of the instance $I$ of LOCAL-POPULARITY-EXISTENCE from the given 3-COLORING instance $G$. We are now going to show that the constructed FEN-B game $I$ admits a locally popular partition if and only if $G$ admits a proper 3-coloring.

**Direction "$\Rightarrow$":** Assume first that there exists a locally popular partition $\pi$ for $I$.

Let us fix an arbitrary vertex $v \in V'$ and consider the corresponding B-gadget on $A_v$. Let $\mathcal{C}_v$ denote the set of all coalitions $C \in \pi$ with $C \cap A_v \neq \emptyset$. Let $I_v$ be the FEN-B game comprising solely of $A_v$, and let $\pi^{|v} = \{C \cap A_v : C \in \mathcal{C}_v\}$ denote the partition for $I_v$ that can be thought of as the projection of $\pi$ onto $A_v$. Let the *level of a coalition* $C \in \mathcal{C}_v$ be

$$\mathrm{lev}(C) = |F(v_i) \cap (C \setminus A_v)| - |E(v_i) \cap (C \setminus A_v)|$$

for an arbitrary $v_i \in C \cap A_v$; note that this notion is well-defined, as all vertex agents in $A_v$ have the same friends and enemies outside $A_v$.

Observe now that $I$ satisfies the conditions in Lemma 29: hence, as $\pi$ is locally stable, we get that there exist coalitions $C_1, C_2, C_3 \in \pi$ such that $D_i^+ \subseteq C_i$ for each $i \in [3]$. Since vertex agents in $A_v$ regard only special agents as friends outside $A_v$, it follows that the level of any coalition in $\mathcal{C}_v$ is at most 1.

**Claim 30.** *Each coalition $C \in \mathcal{C}_v$ has level at least* 0.

*Proof of claim.* For the sake of contradiction, assume that $C \in \mathcal{C}_v$ has level at most $-1$.

If $C \cap A_v$ satisfies the condition of Lemma 28 (i.e., either $|C \cap A_v| \geq 4$, or $|C \cap A_v| = 3$ and $C \cap A_v$ is not consecutive), then we know that there exists an agent $v_i$ in $C \cap A_v$ such that the switch $v_i \to \emptyset$. Notice that agents in $A_v \setminus \{v_i\}$ do not consider their relationships outside $A_v$ when voting for or against the switch $v_i \to \emptyset$, so they vote the same way in $I$ as in $I_v$. Moreover, we have

$$\mathrm{vote}_{v_i}(\pi, \pi_{v_i \to \emptyset}) \geq \mathrm{vote}_{v_i}(\pi^{|v}, \pi^{|v}_{v_i \to \emptyset})$$

because $v_i$ has more enemies in $C \setminus A_v$ than friends (since $C$ has level at most $-1$). Finally, both friendship and enemy relationships between two agents who do not belong to the same B-gadget are symmetric, so we know that

$$\sum_{a \in C \setminus A_v} \mathrm{vote}_a(\pi, \pi_{v_i \to \emptyset}) = -\mathrm{lev}(C) \geq 1.$$

Summing up all these insights, we obtain that $\Lambda(\pi, \pi_{v_i \to \emptyset}) \geq 1 - \mathrm{lev}(C) \geq 2$, contradicting the local popularity of $\pi$.

Assume now that $|C \cap A_v| = 3$ and $C \cap A_v$ is consecutive, that is, $C \cap A_v = \{v_i, v_{i+1}, v_{i+2}\}$ for some $i \in [6]$ (henceforth, we treat indices within $A_v$ modulo 7). Then $v_i \to \emptyset$ locally dominates $\pi$: agent $v_i$ votes against it, $v_{i+2}$ votes for it, $v_{i+1}$ does neither, while the total vote of all agents in $C \setminus A_v$ sums up to $-\mathrm{lev}(C) \geq 1$, yielding $\Lambda(\pi, \pi_{v_i \to \emptyset}) \geq 1$.

If $|C \cap A_v| = 2$ and $C \cap A_v$ is nonconsecutive, then the switch $v_i \to \emptyset$ for any $v_i \in C$ locally dominates $\pi$, because at least one agent in $C \cap A_v$ votes for it, while the total vote of all agents in $C \setminus A_v$ sums up to $-\mathrm{lev}(C) \geq 1$, yielding $\Lambda(\pi, \pi_{v_i \to \emptyset}) \geq 1$.

If $C \cap A_v = \{v_i, v_{i+1}\}$ for some agent $v_i \in C$, then again $v_i \to \emptyset$ locally dominates $\pi$: by $u^{\mathrm{B}}_{v_i}(\pi) = 0$, both $v_i$ and $v_{i+1}$ are indifferent between $\pi$ and $\pi_{v_i \to \emptyset}$, and the total vote of all agents in $C \setminus A_v$ sums up to $-\mathrm{lev}(C) \geq 1$, yielding $\Lambda(\pi, \pi_{v_i \to \emptyset}) \geq 1$.

Finally, if $C \cap A_v = \{v_i\}$, then again $v_i \to \emptyset$ locally dominates $\pi$, because $v_i$ votes for it, and the total vote of all agents in $C \setminus A_v$ sums up to $-\mathrm{lev}(C) \geq 1$, yielding $\Lambda(\pi, \pi_{v_i \to \emptyset}) = 2$. $\triangleleft$

We next show that it is not possible for all coalitions in $\mathcal{C}_v$ to have level exactly 0. Assume the contrary. By Theorem 8, we know that $\pi^{|v}$ cannot be locally popular in $I_v$, so there exists an agent $v_i$ and a coalition $C \in \pi^{|v} \cup \{\emptyset\}$ such that the switch $v_i \to C$ locally dominates $\pi^{|v}$ in $I_v$. Let $C' \in \pi$ be the coalition satisfying $C = C' \cap A_v$. We will show that the switch $v_i \to C'$ locally dominates $\pi$ in $I$. First, agents in $A_v \cap (\pi(v_i) \cup C') \setminus \{v_i\}$ vote for or against the switch without regard for the agents outside $A_v$. Second, the contribution of all agents outside $A_v$ to the balanced utility of $v_i$ remains the same in $C'$ as in $\pi(v_i)$ due to $\mathrm{lev}(C') = \mathrm{lev}(\pi(v_i)) = 0$. Third, the total vote of all agents regarding the switch $v_i \to C'$ sums up to exactly $-\mathrm{lev}(\pi(v_i) + \mathrm{lev}(C) = 0$. Summing up all this, we get $\Lambda(\pi, \pi_{v_i \to C'}) \geq 1$, as promised.

By Claim 30, this shows that for each $v \in V'$, there exists some $C \in \mathcal{C}_v$ with $\mathrm{lev}(C) = 1$; if there are multiple such coalitions, then fix one arbitrarily. Notice that $C$ must contain a unique special agent $s_i$ due to $\mathrm{lev}(C) = 1$; we then define $\chi(v) = i$. We claim that $\chi$ is a proper coloring of $H$: Assuming that the endpoints of an edge $\{u, v\} \in E'$ (recall that $E'$ is the edge set of $H$) have the same color $i$, we get that $C_i = \pi(s_i)$ contains agents both from $A_u$ and $A_v$; however, as they are mutual enemies and the only friend of vertex agents outside their gadget must be a special agent, we obtain that agents in $A_v \cap C_i$ have at least as many enemies as friends in $C_i \setminus A_v$, a contradiction to $\mathrm{lev}(C) = 1$. It follows that $H$ and, hence, its subgraph $G$ admits a proper 3-coloring.

**Direction "$\Leftarrow$":** Assume now that $G$ admits a proper coloring, so $H$ admits a proper and tight coloring $\chi : V' \to [3]$. Create the partition $\pi$ consisting of three coalitions

$$C_i = D_i^+ \cup \{A_v : v \in V', \chi(v) = i\}$$

for $i = 1, 2, 3$. We claim that $\pi$ is locally popular. For the sake of contradiction, assume that some switch $a \to C$ locally dominates $\pi$.

First, if $a$ is a dummy or special agent in $C_i$ for some $i \in [3]$, then all agents in $D_i^+$ vote against $a \to C$, and the only agents that may vote for it are vertex agents (in case $a = s_i$); hence, $\Lambda(\pi, \pi_{a \to C}) \leq 0$ by $|D_i^+| > 7n$.

Second, if $a$ is a vertex agent in $A_v$, then three agents in $A_v$ vote for $a \to C$, while two agents in $A_v$ vote against it. We also know that $u^{\mathrm{B}}_a(\pi) = 0$, because it has three enemies in $A_v$ and two friends in $A_v$ plus $s_i$. Moreover, no vertex agent in $A_v$ votes for the switch $a \to C$: since $\chi$ is a proper coloring, there are no enemies of $a$ in $C_i \setminus A_v$; the special agent $s_i$ votes against it, and dummies in $C_i$ are indifferent between $\pi$ and $\pi_{a \to C}$. This already shows that $a \to \emptyset$ does not locally dominate $\pi$, because $\Lambda(\pi, \pi_{a \to \emptyset}) = \sum_{a' \in C_i} \mathrm{vote}_{a'}(\pi, \pi_{a \to \emptyset}) = 0$. To deal with the case when $C = C_j \in \pi$ for some $j \neq i$, recall that since $\chi$ is tight, there is at least one vertex $u \in V'$ among the neighbors of $v$ with $\chi(u) = j$. This means that all seven agents in $A_u \subseteq C_j$ vote against $a \to C_j$; while the only agent in $C_j$ that votes for it is $s_j$. Therefore, $\Lambda(\pi, \pi_{a \to C_j}) \leq -6$. This contradiction proves that $\pi$ is indeed locally popular. $\square$

### D.2.5 Proof of Proposition 10

**Proposition 10** ($\star$). *There exists a symmetric FEN-B game that admits a partition that is popular but does not maximize the agents' total balanced utility.*

*Proof.* Consider the symmetric FEN-B game $I$ on agent set $N = A \cup B$ where $A = \{a_i : i \in [7]\}$ and $B = \{b_i : i \in [7]\}$. To create the (undirected) friendship graph $G^F$ of $I$, we add a clique on $A$, a clique on $B$, plus the three edges in $\{\{a_i, b_i\} : i \in [3]\}$. The (undirected) enmity graph $G^E$ of $I$ contains only the four edges in $\{\{a_7, b_j\} : j = 4, 5, 6, 7\}$.

We claim that the grand coalition $\pi^N = \{N\}$ is popular for $I$. Assume for the sake of contradiction that there is a partition $\pi$ with $\Lambda(\pi^N, \pi) > 0$. Note first that no agent in $N \setminus \{a_7, b_4, b_5, b_6, b_7\}$ may prefer $\pi$ to $\pi^N$ as they have all their friends with them and no enemies in the coalition $N$. Thus at most five agents may prefer $\pi$ to $\pi^N$. Second, if some agent prefers $\pi$ to $\pi^N$, then $\pi$ must put $a_7$ and at least one agent in $\{b_4, b_5, b_6, b_7\}$ in different coalitions.

If there is no coalition in $\pi$ containing $A$, then all six agents in $A \setminus \{a_7\}$ prefer $\pi^N$ to $\pi$.

If some agent other than $a_7$ prefers $\pi$ to $\pi^N$, then $B$ must also be contained in some coalition of $\pi$. Otherwise, no agent in $\{b_4, b_5, b_6, b_7\}$ prefers $\pi$ to $\pi^N$, as they lose at least one friend and lose at most one enemy when switching from $\pi^N$ to $\pi$. Hence, in this case only $\pi = \{A, B\}$ is possible, but then all six agents in $\{a_i, b_i : i \in [3]\}$ prefer $\pi^N$ to $\pi$.

Finally, if only $a_7$ prefers $\pi$ to $\pi^N$, then $\pi$ must contain a coalition $C$ with $C \subsetneq B$. However, then all agents $b_i$, $i \in [3]$, prefer $\pi^N$ to $\pi$.

This proves that $\pi^N$ is indeed popular.

The total balanced utility of $\pi^N$ is $f(\pi^N) = 4 \cdot \binom{7}{2} + 2 \cdot 3 - 2 \cdot 4$, but this is lower than the total balanced utility of the partition $\{A, B\}$ by $f(\{A, B\}) = 4 \cdot \binom{7}{2} < f(\pi^N)$. This proves our statement. $\qquad\square$

### D.2.6 Proof of Theorem 11

**Theorem 11** ($\star$). *Given a symmetric FEN-B game $I$ and an integer $t$, the problem of deciding whether $I$ admits a partition $\pi$ whose total balanced utility is at least $t$ is NP-complete.*

*Proof.* It is clear that the problem is in NP, as we can compute the total utility of any partition efficiently. To prove NP-hardness, we again present a reduction from 3-COLORING based on the construction in Lemma 29. Given an input graph $G = (V, E)$ over $n$ vertices, let us create a symmetric FEN-B game $I$ as follows. We introduce a set $D_1 \cup D_2 \cup D_3$ of *dummy agents* with $|D_1| = |D_2| = |D_3| = 3n$, three *special agents*, $s_1$, $s_2$, and $s_3$, and a *vertex agent* $v'$ for each vertex $v \in V$. We will again use the notation $D_i^+ = D_i \cup \{s_i\}$ for each $i \in [3]$.

The friendship and enmity relationships between the agents are as follows:

- For each $i \in [3]$, every agent in $D_i^+$ considers each other agents in $D_i^+$ a friend and every remaining special or dummy agent an enemy.

- Two vertex agents, $u'$ and $v'$, are mutual enemies if and only if $\{u, v\} \in E$; otherwise, they are mutually neutral.

- Each vertex agent is mutual friends with every special agent and mutually neutral with every dummy agent.

Additionally, we set our target value as $t = 6\binom{n}{2} + 2n$.

This completes our construction of the instance $I$ from the given 3-COLORING instance $G$. We are going to show that the constructed FEN-B game $I$ admits a partition $\pi$ with $f(\pi) \geq t$ if and only if $G$ admits a proper 3-coloring.

**Direction "$\Rightarrow$":** Assume first that there exists a popular partition $\pi$ for $I$ with $f(\pi) \geq t$; without loss of generality, we may assume that $\pi$ maximizes the total balanced utility over all partitions for $I$. In particular, this implies that $I$ is locally stable: otherwise, a switch that locally blocks $\pi$ would result in a strict increase in $f(\pi)$, as shown in Theorem 4; a contradiction. Therefore, Lemma 29 can be applied, and we obtain that $\pi$ contains three coalitions $C_1, C_2, C_3 \in \pi$ such that $D_i^+ \subseteq C_i$ for each $i \in [3]$.

Since each vertex agent can belong to at most one of the sets $C_1$, $C_2$, and $C_3$, the total balanced utility of all dummy and special agents is at most $6\binom{n}{2} + n$. Moreover, the balanced utility of any

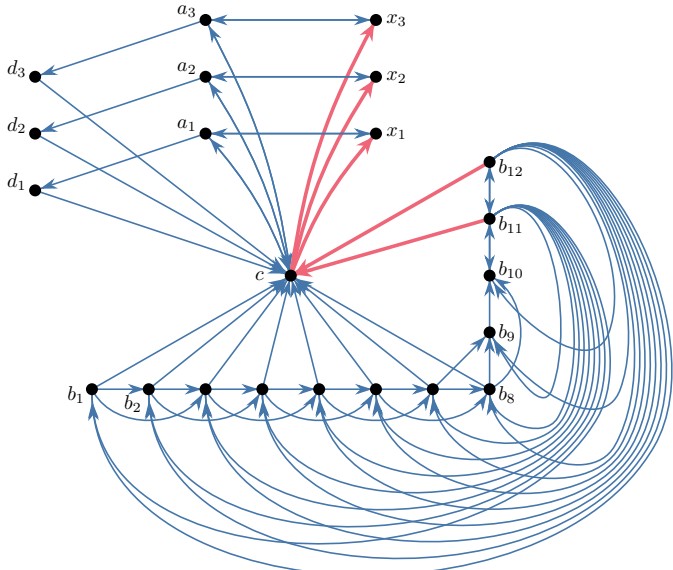

Figure 13: Illustration of the AE-gadget as defined in Definition 4 (repeated from Figure 2). We denote friendship arcs as thin, blue lines and enemy arcs as thick, red lines.

vertex agent $v'$ is at most 1, as the only agents regarded as friends by $v'$ are special agents, and $v'$ can share a coalition with at most one of them. Hence, $f(\pi) \le 6\binom{n}{2} + 2n = t$, and equality can only be achieved if each vertex agent $v$ is in coalition $C_i$ for some $i \in [3]$ and has no enemies in $\pi(v') = C_i$. However, this implies that each set $\{v \in V : v' \in C_i\}$ is an independent set in $G$ (i.e., no two of its vertices are connected by an edge of $G$), proving that $G$ admits a proper 3-coloring.

**Direction "$\Leftarrow$":** Assume now that $G$ admits a proper 3-coloring $\chi : V \to [3]$. Define coalitions $C_i = D_i^+ \cup \{v' : v \in V \text{ and } \chi(v) = i\}$ for $i \in [3]$. Then the partition $\pi = \{C_1, C_2, C_3\}$ has total balanced utility exactly $f(\pi) = 6\binom{n}{2} + 2n = t$, proving the correctness of our reduction. $\qquad\square$

### D.3 Omitted proofs for our results on FEN-AE games

After stating some structural observations in Section D.3.1, we provide all omitted proofs for FEN-AE games in Sections D.3.2–D.3.3. See Figure 13 for an illustration of an AE-gadget, our building block for all of our nonexistence and hardness results concerning FEN-AE games.

#### D.3.1 A structural lemma on AE-gadgets

**Lemma 31.** *Consider a FEN-AE game I that contains an AE-gadget as described in Definition 4 over agent set $K = \{c\} \cup \{b_i : i \in [12]\} \cup \{a_i, d_i, x_i : i \in [3]\}$, and let $\pi$ be a locally stable partition for I. Assume that no agent outside $K$ considers anyone within $K$ a friend, and among agents of $K$, only $x_1$, $x_2$, and $x_3$ may have friends outside $K$. Furthermore, assume that coalitions $\pi(c)$, $\pi(b_{11})$, and $\pi(b_{12})$ only contain agents from $K$. Then*

    *(a) $\{a_1, a_2, a_3, d_1, d_2, d_3\} \subseteq \pi(c)$;*

    *(b) if, for some $i \in [3]$, agent $x_i$ considers no agent in $\pi(x_i) \setminus K$ a friend, then $x_i \in \pi(c)$; and*

    *(c) $\{b_i : i \in [12]\} \subseteq \pi(b_1) \ne \pi(c)$.*

*Proof.* To show (a), we first show that $d_i \in \pi(c)$ for each $i \in [3]$. By our assumptions, the only agents who consider $d_i$ a friend are $a_i$ and $c$, whereas $d_i$ has only $c$ as a friend and has no enemies in $\pi(c)$. Therefore, if $d_i \notin \pi(c)$, then $d_i$ has an incentive to join $\pi(c)$, and thus the switch $d_i \to \pi(c)$ locally blocks $\pi$: both $d_i$ and $c$ vote for it and only $a_i$ may vote against it. This proves $d_i \in \pi(c)$.

This implies that among the three agents whom $a_i$, for some $i \in [3]$, considers friends, two are contained in $\pi(c)$, and due to our assumptions, $a_i$ has no enemies in $\pi(c)$. Hence, if $a_i \notin \pi(c)$, then $a_i$ has an incentive to join $\pi(c)$. The switch $a_i \to \pi(c)$ then locally blocks $\pi$: both $a_i$ and $c$ vote for it, while only $x_i$ (the only agent who considers $a_i$ a friend) may vote against it. This contradiction proves $\{a_1, a_2, a_3, d_1, d_2, d_3\} \subseteq \pi(c)$, as desired.

To show (b), assume for the sake of contradiction that $x_i \notin \pi(c)$ and $x_i$ has no friends in $\pi(x_i) \setminus K$. Then $x_i$ has no friends in $\pi(x_i)$, because the only agent whom $x_i$ considers a friend is $a_i \in \pi(c)$. Since $x_i$ has no enemies in $\pi(c) \subseteq K$ due to our assumptions, $x_i$ has an incentive to join $\pi(c)$. Hence, $x_i \to \pi(c)$ locally blocks $\pi$, as both $x_i$ and $a_i$ vote for it, and only $c$ votes against it; a contradiction proving (b).

It remains to show (c). First observe that $c \notin \pi(b_{12})$ must hold, as otherwise, the switch $b_{12} \to \emptyset$ would locally block $\pi$, as both $c$ and $b_{12}$ would vote for it (as they are mutual enemies) and only $b_{11}$ may vote against it. Similarly, $c \notin \pi(b_{11})$ also holds, as otherwise, the switch $b_{11} \to \emptyset$ would locally block $\pi$, since both $c$ and $b_{11}$ would vote for it and only $b_{10}$ may vote against it (note that $b_{12} \notin \pi(c)$ as we have just shown).

Next, we claim that $\pi(b_{11}) = \pi(b_{12})$. Suppose the contrary: $\pi(b_{11}) \neq \pi(b_{12})$. Choose $j$ and $j'$ such that $\{j, j'\} = \{11, 12\}$ and $\pi(b_j)$ contains at least as many agents from $\{b_i : i \in [10]\}$ as $\pi(b_{j'})$. Since $b_{11}$ and $b_{12}$ consider each other a friend and, apart from each other, they have the same set of friends, and their only enemy $c$ is not in either of $\pi(b_{11})$ or $\pi(b_{12})$, it follows that the switch $b_{j'} \to \pi(b_j)$ locally blocks $\pi$, as both $b_{11}$ and $b_{12}$ vote for it and only $b_{10}$ may vote against it.

It follows that $b_{10} \in \pi(b_{11})$, as otherwise, $b_{10}$ has an incentive to join $\pi(b_{11})$, and the switch $b_{10} \to \pi(b_{11})$ locally blocks $\pi$: agents $b_{10}$, $b_{11}$, and $b_{12}$ all vote for it and only $b_8$ and $b_9$ may vote against it. Similarly, we obtain $b_9 \in \pi(b_{11})$. Finally, once we know that $b_{i'} \in \pi(b_{11})$ for all $i' > i$ for some $i \in [8]$, then $b_i \in \pi(b_{11})$ follows, as otherwise, $b_i \to \pi(b_{11})$ would locally block $\pi$: among the three agents whom $b_i$ considers friends, $\pi(b_{11})$ contains two and contains no enemies of $b_i$ by $\pi(b_{11}) \subseteq K$, so $b_i$ together with $b_{11}$ and $b_{12}$ vote for $b_i \to \pi(b_{11})$, while only $b_{i-1}$ and $b_{i-2}$ may vote against it. Hence, $\{b_i : i \in [12]\} \subseteq \pi(b_{11}) = \pi(b_1) \neq \pi(c)$. $\qquad\square$

### D.3.2 Proof of Theorem 12

**Theorem 12** ($\star$)**.** *The asymmetric FEN-AE game consisting solely of an AE-gadget admits no locally stable partition.*

*Proof.* Consider a FEN-AE game $I$ that solely consist of an AE-gadget as described in Definition 4 over agent set $K = \{c\} \cup \{b_i : i \in [12]\} \cup \{a_i, d_i, x_i : i \in [3]\}$. By Lemma 31, the only partition that may be locally stable for $I$ is $\pi = \{\{a_i, d_i, x_i, c : i \in [3]\}, \{b_i : i \in [12]\}\}$. Observe that $c$ has two enemies and zero friends in $\pi(b_1)$, whereas $c$ has three enemies in $\pi(c)$, and hence, $c$ prefers $\pi(b_1)$ to $\pi(c)$. Therefore, the switch $c \to \pi(b_1)$ locally blocks $\pi$: nine agents (namely, $c$ and $b_j$ for $j \in [8]$) vote for it, while only eight agents (namely, $a_i$ and $d_i$ for $i \in [3]$ as well as $b_{11}$ and $b_{12}$) vote against it. Thus no partition for $I$ is locally stable. $\qquad\square$

### D.3.3 Proof of Theorem 13

**Theorem 13** ($\star$)**.** LOCAL-POPULARITY-EXISTENCE *and* LOCAL-STABILITY-EXISTENCE *are* NP-*complete for FEN-AE games.*

*Proof.* It is clear that both problems are in NP, as given a FEN-AE game, we can nondeterministically guess a partition $\pi$ and then verify in polynomial time that no possible switch locally dominates or locally blocks $\pi$ (recall that the number of switches for $\pi$ is at most quadratic in the number of agents).

To show the NP-hardness of these problems, we present a reduction from 3-SAT that works for both problems. Let $\varphi = C^1 \wedge \cdots \wedge C^m$ be an instance of 3-SAT, i.e., a Boolean formula over variables $v_1, \ldots, v_n$ with three literals per clause; without loss of generality, we assume that $m \geq 2$.

We create a FEN-AE game $I$ of LOCAL-POPULARITY-EXISTENCE and LOCAL-STABILITY-EXISTENCE as follows. For each clause $C^j$, we create an AE-gadget $G^j$ as described in Definition 4 over agent set $K^j = \{c^j\} \cup \{b_i^j : i \in [12]\} \cup \{a_i^j, d_i^j, x_i^j : [i \in [3]\}$. We will call $c^j$ a *clause agent*

and agents $x_1^j$, $x_2^j$, and $x_3^j$ *literal agents*, corresponding to the three literals in $C^j$. We further add an agent $T$.

The friendship and enmity relations between agents from different AE-gadgets are as follows:

- $x_i^j$ considers $T$ a friend for each $i \in [3]$ and $j \in [m]$, while all other agents consider $T$ an enemy.

- $T$ considers $x_i^j$ a neutral for each $i \in [3]$ and $j \in [m]$ but considers every other agent an enemy.

- Two agents, $u$ and $v$, from different AE-gadgets are mutual enemies unless both of them are literal agents corresponding to literals that are not negations of each other, in which case they are mutually neutral.

We prove the correctness of the reduction via Claims 32 and 33.

**Claim 32.** *If $I$ admits a locally stable partition, then $\varphi$ is satisfiable.*

*Proof of claim.* Let $\pi$ be a locally stable partition for $I$. For the sake of contradiction, suppose that an agent $v \in \{c^j, b_{11}^j, b_{12}^j\}$ is contained in the same coalition of $\pi$ as some agent $u$ from a different AE-gadget $G^h$. Let $\ell$ denote the largest integer such that both $u$ and $v$ have at least $\ell$ enemies in $\pi(u) = \pi(v)$. Since $u$ and $v$ are mutual enemies, we know $\ell \geq 1$, and thus both $u$ and $v$ have an incentive to join $\emptyset$. However, none of the switches $u \to \emptyset$ and $v \to \emptyset$ can locally block $\pi$, so there must exist a set $F_u$ of at least $\ell + 1$ agents in $\pi(u) = \pi(v)$ who consider $u$ a friend and, similarly, a set $F_v$ of at least $\ell + 1$ agents who consider $v$ a friend in $\pi(v) = \pi(u)$. In particular, $u$ is not a literal agent $x_i^h \in K^h$ for some $i \in [3]$, as $x_i^h$ is regarded as a friend only by $a_i^h$. Moreover, as all friendship relations lie within some AE-gadget, we get that $F_u \subseteq K^h$ and $F_v \subseteq K^j$. However, $u$ must then be mutual enemies with all $\ell + 1$ agents in $F_v \subseteq \pi(v)$, and $v$ mutual enemies with all $\ell + 1$ agents in $F_u \subseteq \pi(v)$, which contradicts the definition of $\ell$.

Again for the sake of contradiction, suppose now that an agent $v \in \{c^j, b_{11}^j, b_{12}^j\}$ is contained in $\pi(T)$. Since $v$ and $T$ are mutual enemies but $v \to \emptyset$ does not locally block $\pi$, there must be at least two agents in $\pi(T)$ who consider $v$ a friend. These two agents cannot be literal agents by the choice of $v$, so they are also mutual enemies with $T$. Hence, at least four agents vote for the switch $T \to \emptyset$ including $T$. However, we have already shown that no agent from an AE-gadget other than $G^j$ can be contained in $\pi(v) = \pi(T)$; therefore, at most three agents in $\pi(T)$ consider $T$ a friend, implying that $T \to \emptyset$ locally blocks $\pi$, a contradiction.

We conclude that coalitions $\pi(c^j)$, $\pi(b_{11}^j)$, and $\pi(b_{12}^j)$ only contain agents from $G^j$. This means that all conditions of Lemma 31 are satisfied. Hence, if no agent $x_i^j \in \{x_1^j, x_2^j, x_3^j\}$ considers at least one agent in $\pi(x_i^j) \setminus K^j$ a friend, then by Lemma 31 we get $\pi(c^j) = \{x_i^j, a_i^j, d_i^j, c^j : i \in [3]\}$ and $\pi(b_1^j) = \{b_i^j : i \in [12]\}$. However, then it follows from Theorem 12 that some switch, namely the switch $c^j \to \pi(b_1^j)$, locally blocks $\pi$, a contradiction.

This means that for each AE-gadget $G^j$, there exists a literal agent $x_i^j \in K^j$ that considers at least one agent in $\pi(x_i^j) \setminus K^j$ a friend; by construction, this agent can only be $T$. Let $X_T$ denote the set of literal agents in $\pi(T)$; then we have just proven that $X_T \cap K^j \neq \emptyset$ for each $j \in [m]$. Suppose that two agents in $X_T$ are mutual enemies (which happens if they correspond to literals $v_i$ and $\overline{v}_i$ for some variable $v_i$). Then both of them have an incentive to join $\emptyset$, and thus both of them vote for the corresponding switches, whereas at most one agent may vote against them (because for each literal agent $x_i^j$, there is only one agent, namely $a_i^j$, who considers $x_i^j$ a friend); thus, either of these switches locally blocks $\pi$, a contradiction.

Therefore, we can create a truth assignment by setting $v_i$ to true if and only if there is a literal agent corresponding to $v_i$ in $X_T$. By the above arguments, this is a consistent truth assignment and yields a true literal in each clause, thus satisfying $\varphi$. ◁

**Claim 33.** *If $\varphi$ is satisfiable, then $I$ admits a locally popular partition.*

*Proof of claim.* Suppose that $\varphi$ is satisfiable by a truth assignment $\mu$. We create a partition $\pi$ as follows. We let $\pi(T)$ contain—besides $T$—all literal agents that correspond to true literals. Additionally, we create a coalition $\{b_i^j : i \in [12]\}$ and a coalition $\{a_i^j, d_i^j, x_i^j, c^j : i \in [3]\} \setminus \pi(T)$ for each AE-gadget $G^j$.

We claim that $\pi$ is locally popular. Since $\mu$ satisfies $\varphi$, each clause agent $c^j$ has at most two enemies and six friends in $\pi(c^j)$, has two enemies and zero friends in $\pi(b_1^j)$, and has at least $m + 1 \geq 3$ enemies in $\pi(T)$. Hence, the only possible coalition in $\pi \cup \{\emptyset\}$ that $c^j$ has an incentive to join is $\emptyset$ via the switch $c^j \to \emptyset$. Regarding the remaining (i.e., all non-clause) agents, observe that none of them has enemies in its coalition under $\pi$. Using this, we get the following:

- Neither $T$ nor any agent $d_i^j$ votes for any possible switch.

- An agent $b_i^j$ may only vote for the switch $c^j \to \pi(b_1^j)$. However, such a switch does not locally dominate $\pi$, as all agents in $\{c^j, a_i^j, d_i^j : i \in [3]\} \cup \{b_{11}^j, b_{12}^j\}$ vote against such a switch, whereas only the agents in $\{b_i^j : i \in [8]\}$ vote for it.

- As we have argued, a clause agent $c^j$ votes only for the switch $c^j \to \emptyset$; however, such a switch does not locally dominate $\pi$, as only $c^j$ votes for it, whereas all six agents in the set $\{a_i^j, d_i^j : i \in [3]\}$ vote against it.

- An agent $a_i^j$ votes only for switches of the form $x_i^j \to \pi(a_i^j)$, where $x_i^j \in \pi(T)$. However, such a switch does not locally dominate $\pi$, as only $a_i^j$ votes for it, while $c^j$ votes against it.

- Finally, an agent $x_i^j$ may only vote for a switch of the form $T \to \pi(x_i^j)$ or $a_i^j \to \pi(x_i^j)$. A switch $T \to \pi(x_i^j)$ for some $x_i^j \notin \pi(T)$ is voted for only by the at most two literal agents in $\pi(x_i^j)$, while it is voted against by $T$ and all remaining seven agents in $\pi(x_i^j)$. A switch $a_i^j \to \pi(x_i^j)$ for some $x_i^j \in \pi(T)$ is voted for only by the at most three literal agents in $\pi(T) \cap K^j$, while $T$, $c^j$, and at least $m - 1 \geq 1$ literal agents from $\pi(T) \setminus K^j$ vote against it. Thus none of these switches locally dominates $\pi$.

We obtain that no switch for which at least one agent votes locally dominates $\pi$, that is, $\pi$ is locally popular. ◁

As a locally popular partition is locally stable by definition, NP-hardness of both LOCAL-STABILITY-EXISTENCE and LOCAL-POPULARITY-EXISTENCE follows. □

