# OpenReview forum: "Clustering via Hedonic Games: New Concepts and Algorithms"
_NeurIPS.cc/2025/Conference — NeurIPS 2025 spotlight_

### Official Review · Reviewer_Wf17 · 2025-06-26

**Clarity:** 4
**Significance:** 2
**Originality:** 3
**Rating:** 4
**Confidence:** 2

**Summary:**

This paper adopts concepts from FEN (friends–enemies–neutrals) hedonic games to study clustering, drawing a parallel between the two areas: data points are seen as agents with preferences over coalitions, similarities/dissimilarities are represented as friendship/enmity graphs. The authors extend the concept of local stability from FE games to FEN games, and introduce a stronger stability notion called local popularity. They develop clustering heuristics based on local improvements of these stability criteria and prove polynomial-time convergence results for various setting of the clustering game, including symmetric, asymmetric, FEN-AF, FEN-AE, and FEN-B settings. They complement their algorithmic results by showing NP-completeness of the decision problem on existence of locally stable or popular partitions in asymmetric cases, and demonstrate their algorithms with experiments.

**Questions:**

1. Regarding weakness item 1, can the authors please provide some motivation for local popularity?
2. Could the authors provide some intuition and explanation on why the hardness results for FEN-AF and FEN-AE differ? It seems to me that they are ``symmetric'' settings of each other.

**Ethical Concerns:**

["NO or VERY MINOR ethics concerns only"]

**Final Justification:**

I remain positive about the paper and will keep my rating as is.

**Limitations:**

Yes.

**Quality:**

3

**Strengths And Weaknesses:**

Strengths:
1. The paper bridges concepts from hedonic games and clustering, where the stability concepts are meaningful as local interactions and preferences, and they develop provably efficient algorithms based on these local stability/popularity properties to produce clusterings that are robust against local deviations.
2. The authors thoroughly analyzed the two algorithms LocPop and LocStab under various settings based on preference domains (AF, AE, B), symmetric/asymmetric relationships, and provide comprehensive theoretical results for both upper bounds (polynomial-time convergence) and matching hardness results (NP-completeness), offering a complete picture.

Weaknesses:
1. A stronger stability notion of local popularity was introduced, but the practical benefits are unclear to me. It increases the computational complexity by a factor of $\Delta$ (and especially for asymmetric FEN-B it is a difference of P v.s. NP), but as pointed out by the authors, LocPop and LocStab have very similar empirical performances with respect to the tested evaluation metrics.
2. The paper provides guarantees of convergence to locally stable or popular clustering irrespective of the initial partition, but the high variances in some of the experiment results suggest the quality of the final clustering, in terms of the selected metrics, is very sensitive to initial partitions.
3. To define friendship and enmity graphs in the experiments, the parameters $f$ and $e$ requires fine tuning, and as pointed out by the authors themselves, these parameters can significantly affect experiment performances, and which values perform well are arbitrary.

---

> ### Author Rebuttal · Authors · 2025-07-30
>
> Thank you very much for your helpful comments and questions, which we answer below.
>
> ******
> **Comment**: The paper provides guarantees of convergence to locally stable or popular clustering irrespective of the initial partition, but the high variances in some of the experiment results suggest the quality of the final clustering, in terms of the selected metrics, is very sensitive to initial partitions.
>
> **Answer**: We agree that the initial partition can play an important role in the quality of the final clustering. Hence, it should be emphasized that starting from a good solution---e.g., from the output of another clustering algorithm, as we have done in some of the experiments---can be beneficial for our method. In general cases, we used a random initial partition to neutralize this dependency, leading to higher variance.
>
> **********
> **Comment**: **weakness item 1**. A stronger stability notion of local popularity was introduced, but the practical benefits are unclear to me. It increases the computational complexity by a factor of $\Delta$ (and especially for asymmetric FEN-B it is a difference of P v.s. NP), but as pointed out by the authors, LocPop and LocStab have very similar empirical performances with respect to the tested evaluation metrics.
>
> **Question**: 1. Regarding **weakness item 1**, can the authors please provide some motivation for local popularity?
>
> **Answer**: First of all, as mentioned in Theorem 2, the additional factor of $\Delta$ does not appear in the symmetric balanced preference domain, and by our experiments, the balanced version performs just as good as the others. Hence, we believe that local popularity in the balanced case is the notion that provides the best balance between flexibility and efficient computation. Local stability may give too much power to single agents, as they can reject a switch if they do not prefer it, which can sacrifice flexibility in some cases.
>
> ******
> **Question**: 2. Could the authors provide some intuition and explanation on why the hardness results for FEN-AF and FEN-AE differ? It seems to me that they are ``symmetric'' settings of each other.
>
> **Answer**: FEN-AF and FEN-AE games provide different dynamics. In FEN-AF games, coalitions that are desirable can contain both enemies and friends. However, in FEN-AE games the aversion of enemies dictates that only coalitions with no enemies can be desirable to being alone.
> For this reason, in a reduction to the problem of finding a locally stable or locally popular partition in FEN-AE games, the enmity relations needed to be constructed with much more attention and technical finesse, and even examples with no locally stable or locally popular solutions required more complex components.

---

### Official Review · Reviewer_sZgp · 2025-06-29

**Clarity:** 3
**Significance:** 2
**Originality:** 2
**Rating:** 4
**Confidence:** 4

**Summary:**

The paper proposes the usage of algorithms and solution concepts from the hedonic games literature for clustering. For this, they introduce two new solution concepts in a subclass of hedonic games and provide algorithms that are guaranteed to produce an outcome fulfilling these concepts (by showing the guaranteed convergence of the best response dynamics via a potential function algorithm). They also show that finding stable outcomes in slightly more general subclasses of games is computationally hard. They demonstrate in experiments that their method outperforms several clustering algorithms in clustering tasks.

**Questions:**

1.	Can you please elaborate on the relationship between your paper and the “Hedonic clustering games” paper referenced above?
2.	Is your theoretical contribution needed to apply hedonic games to clustering? Are alternative solution concepts / subclasses of hedonic games either computationally intractable or non-applicable?
3.	Could you please expand on the relation between your solution concepts and the classic quality measures from the clustering community?
4.	Is the complexity of your solution concepts in general FEN-games (for arbitrary parameters) an open question?

**Ethical Concerns:**

["NO or VERY MINOR ethics concerns only"]

**Final Justification:**

I have increased my score to borderline accept, in light of the fact that the authors have addressed my concern related to missing related work.
To increase my score further, I would need to see a revised version of the paper with a reworked framing that would argue in a more convincing way why the theoretical contributions were necessary to achieve the experimental results.

**Limitations:**

yes

**Quality:**

3

**Strengths And Weaknesses:**

Currently, I view the submission as a borderline paper, but I am very open to increasing my score and supporting acceptance if the authors respond satisfactorily to my questions.

### Reasons to Accept

- Using hedonic games for clustering is a mostly original idea (see below) that provides a contribution to both communities and might open up new directions for fruitful future work. I fully agree with the statement of the authors that the paper “can spark a stronger cooperation between these fields to explore the applicability of coalition formation games in clustering.”
- The paper makes a good conceptual, technical, and experimental contribution, which makes it a generally appealing read.
- The theoretical contribution is solid and original (using standard techniques).
- The paper is generally well-written and clear.
- The paper shows that the solution approach via hedonic games leads to better performance in experiments.
- The paper also contributes to a new line of work on the convergence of best response dynamics in hedonic games.

### Reasons to Reject

 ***Missing Related Paper***

The following paper feels conceptually closely related to the submission, but is not cited or discussed:
*Moran Feldman, Liane Lewin-Eytan, and Joseph (Seffi) Naor. 2015. Hedonic clustering games. ACM Trans. Parallel Comput. 2015.*

***Framing***

Reading the paper, I was struggling with its framing and goal. The paper feels like a mix of two papers: one that introduces how to use clustering for hedonic games (which feels quite original), and a second, standard hedonic games paper with some new results on two stability concepts. In particular, given the paper’s clustering framing, I would have expected that every contribution would be tied to the clustering application, and a clearer discussion on why the studied solution concepts are desirable in the clustering domain. I think the paper would get a lot stronger if the authors could argue that their work was necessary to apply hedonic games to clustering.

***Clarity Issues***

There is some room for improvement in the introduction section:

- The first part of the introduction switches multiple times between motivation and contribution (e.g., Lines 40–45 feel like a contribution section) and then has a separate contribution section at the end. Separating these aspects more clearly could make the introduction easier to digest and less redundant.
- I would have appreciated a clearer discussion on how your solution concepts relate to classic quality measures from the clustering world. Do you perceive your methods as proxies to optimize existing quality measures, or do you rather view them as appealing in themselves?
- The introduction feels pretty dense in terms of technical jargon, descriptions, and discussions. Especially for readers from the clustering community, I fear that the initial part of the paper will be quite hard to follow.
- Over the whole paper, at the beginning of a section and subsections, I was oftentimes missing a roadmap of what follows, and was thus unsure what to expect in the following. Generally, the writing in Section 3 feels very statement-heavy with limited intuition.

***Some Questions I Had When Reading the Paper*** (no need to address them in the rebuttal)

- I would be curious how far standard clustering solutions are from being stable.
- I found the name “local stability” quite confusing, as the popularity condition is still baked into this solution concept but not reflected in the name at all.
- Is the NP-completeness of your problems a practical hurdle for computing solutions fulfilling the concept, e.g., using heuristics that empirically often converge or ILPs?
- Any intuition on whether there is also a more direct/constructive algorithm to compute LocPop and LocStab?
- I would have appreciated a slightly more detailed discussion of the idea behind the Rand index and silhouette score in Section 5.

---

> ### Author Rebuttal · Authors · 2025-07-30
>
> Thank you very much for your helpful comments and questions, which we answer below.
> *******
> **Question**: 1. Can you please elaborate on the relationship between your paper and the “Hedonic clustering games” paper referenced above?
>
> **Answer**:
> We thank the reviewer for leading us to the work by Feldman et al. Their topic is, of course, closely related to ours (as it also connects social choice theory and clustering), but their approach is quite different.
> Notwithstanding that we very much appreciate the results of this interesting paper, we provide a structured comparison of the two approaches, and we will include a condensed version of this discussion in the final version of the paper.
>
>
> 1. The central contribution of our paper is a novel clustering framework rooted in _cooperative_ game theory, whereas Feldman et al. adopt a _noncooperative_ game-theoretic approach centered around **Nash equilibria**.
>
> 2. Feldman et al. analyze the **price of anarchy** and **price of stability**, quantifying the efficiency loss due to selfish agent behavior in equilibrium. In contrast, we focus on _cooperative_ fairness-driven solutions derived from social choice theory.
>
> 3. Their closest model to ours is ``correlation clustering'' where the players' utilities depend on the similarities with other agents in their own cluster and the dissimilarities with agents from other clusters. Here, this similarity is measured by a real number between 0 and 1. Both this and our framework are special cases of additively separable hedonic games.
>
> 4. A major limitation of the Nash equilibrium approach is its restrictive applicability: existence and convergence are guaranteed only in special metric spaces (e.g., agents on a line or a tree). Our framework, by contrast, operates over arbitrary metrics and can be applied to a much broader class of clustering scenarios.
>
> 5. While Feldman et al. establish theoretical existence and convergence results, they do not provide practical or efficient algorithms. For the closest case, the correlation clustering case, Feldman et al. show that finding a Nash equilibrium is computationally hard (namely, PLS-complete), which is a serious bottleneck if it were to be used for larger clustering tasks. Our work however, does not have this issue, as we design algorithms that are both efficient and scalable.
>
> 6. Unlike our work, Feldman et al. do not benchmark their method against existing clustering algorithms, nor do they claim general applicability to standard clustering tasks. They do not discuss how Nash equilibria are expected to perform according to traditional metrics. One of the key takeaways from our paper is that, despite originating in cooperative game theory and social choice theory, our framework performs comparably to---or sometimes even better than---classical clustering algorithms on standard datasets.
>
> 7. Both models allow agents to express preferences over cluster memberships. However, in the paper by Feldman et al., agents move unilaterally between clusters based on individual utility, whereas in our setting, cluster changes must be _collectively_ acceptable, reflecting a more community-aware and more global notion of stability.
> That is the great advantage offered by _cooperative_ game theory.
>
> 8. A distinguishing feature of our model is the use of **FEN games**, which reduces the complexity of utility representation by classifying relationships into discrete categories: friend, enemy, or neutral. This abstraction simplifies the input space without compromising the effectiveness of the resulting clusterings.
>
>
> We hope that this detailed comparison clarifies the complementary nature of the two approaches and highlights the distinct contributions of our work.
>
> ********
> **Question**: 2. Is your theoretical contribution needed to apply hedonic games to clustering? Are alternative solution concepts/subclasses of hedonic games either computationally intractable or non-applicable?
>
> **Answer**: There are very few other notions for cooperative FEN games or, more generally, for additively separable hedonic games that have both guaranteed existence and can be computed efficiently. One such notion is contractional individual stability, which requires that there is no player who wants to switch coalitions, and this switch does not make anyone worse off. This leads to overly restrictive notions compared to our notions of local stability and local popularity, so it lacks the flexibility necessary for being useful in clustering.
>
> Restricted to FEN games, a few more notions become tractable like ``weak core stability.'' However, they either lack the flexibility that our framework has (e.g., weak core partitions can only be found efficiently by computing the strongly connected components of the friendship graph as clusters, which most often would lead to just one single cluster), or they pose too few restrictions (e.g., Pareto optimality).
>
> We refer to Table 1 in a paper by Aziz, Brandt, Seeding, ``Computing desirable partitions in additively separable hedonic games'' (Artificial Intelligence 195:316--334, 2013), for a tractability map of different concepts in ASHGs.
>
> To the best of our knowledge, there has been no paper previously that provides a method for clustering stemming from social choice theory that has the following two traits:
>
> 1. Relies on a concept that is intuitive, always exists in important, nontrivial cases, is efficiently computable, and flexible enough to be generalizable to different kinds of clustering tasks.
> Note further that our algorithm is polynomial even for _asymmetric_ preferences (in LocStab, balanced case), whereas all known best response dynamics convergence results so far seem to depend on _symmetric_ preferences. This is another useful flexibility factor.
>
> 2. Provides large-scale simulation studies, not just for instances where the solution concepts are most natural, but for very distinct and seemingly unrelated tasks like classifying cancerous images, observing impressive results in all cases.
>
> Hence, we strongly believe that the notions studied in our paper are the first ones that make hedonic games truly applicable for clustering tasks.
>
> *********
> **Question**: 3. Could you please expand on the relation between your solution concepts and the classic quality measures from the clustering community?
>
>
> **Answer**:
> We do not claim any formal relationship between our solution concepts and any classic quality measures for clustering; we are not aware of any such direct connections.
> However, we would like to stress that despite the lack of formal relations between clustering measures and our solution concepts, the underlying ideas are fundamentally very similar: In the FEN model, we are looking for a coalition structure where coalitions contain as many friendships and as few enmities as possible; this is very similar to the situation in clustering where we aim for similar data points to be allocated to the same cluster, and very different data points to be allocated to separate clusters---which is the main idea behind many of the classic measures such as Rand index. The array of various clustering measures tries to capture this somewhat vague notion of a _good clustering_, and our solution concept tries to do the same in the context of coalition formation where, apart from similarities and dissimilarities (expressed as friendships/enmities), we may also care about fairness and collective decision-making.
>
> ******
> **Question**: 4. Is the complexity of your solution concepts in general FEN-games (for arbitrary parameters) an open question?
>
>
> **Answer**: Things that we proved to be NP-hard are of course NP-hard for more general cases too. Some of our positive results, such as the existence reults, may extend to symmetric ASHGs, but we expect that their computation will be hard (e.g. PLS-hard) as the potential function may not be polynomially bounded, so the restriction to FEN games is likely a necessary step for the needed efficiency guarantees for clustering.
>
> We refer to the referenced papers by Brandt and Bullinger [10], by Brandt, Bullinger, and Tappe [11], and by Bullinger and Gilboa [13] for an overview of complexity results for various similar solution concepts.
>
>
> *******
> **Small Comments**:
>    -  I would be curious how far standard clustering solutions are from being stable.
>
>    - I found the name “local stability” quite confusing, as the popularity condition is still baked into this solution concept but not reflected in the name at all.
>
> - Is the NP-completeness of your problems a practical hurdle for computing solutions fulfilling the concept, e.g., using heuristics that empirically often converge or ILPs?
>
>    - Any intuition on whether there is also a more direct/constructive algorithm to compute LocPop and LocStab?
>
>     - I would have appreciated a slightly more detailed discussion of the idea behind the Rand index and silhouette score in Section 5.
>
> **Answers**:
> - This is an interesting future direction and we will look at it.
>
> - We agree that the name is not optimal, and we consider switching it. A better name could be _``Nash-stable local popularity.''_
>
> - We do not believe ILPs would be able to provide efficient solutions, especially for large datasets. Heuristics may be able to converge fast for datasets; this is again an interesting future direction.
>
> - We are not aware of any more direct/constructive algorithm to compute LocPop and LocStab. Since maximizing total balanced utility is NP-hard by Theorem 11 we expect that local search may be the best efficient method.
>
> - We will try to add more details on Rand index and Silhouette score if space allows.

---

> > ### Comment · Reviewer_sZgp · 2025-08-02
> >
> > Thanks for the thoughtful response!
> > Q1: I agree with your assessment that your paper makes substantial contributions beyond the Feldman et al. paper. Adding this discussion to the paper and featuring it early on would be important to accurately depict the novelty of the paper.
> > Q2: That makes sense. I see the value of the introduced concepts (beyond the already existing ones) for the clustering application. Adding this discussion would strengthen the connection between the theoretical and experimental parts. If you are interested in further strengthening this link and the paper, it could be valuable to include, in future versions, solution methods based on other tractable clustering solution concepts as baselines to the experiments. If these baselines were to perform significantly worse, that would provide a compelling argument in favor of the solution concepts proposed in the paper and the practical relevance of the theoretical contribution.

---

> > > ### Author Response · Authors · 2025-08-04
> > >
> > > We sincerely thank you for your helpful suggestions! We will add a short discussion on the Feldman et al. paper to the final version. We will also extend our experimental study with new concepts in the journal version.

---

### Official Review · Reviewer_mtec · 2025-07-02

**Clarity:** 3
**Significance:** 2
**Originality:** 2
**Rating:** 3
**Confidence:** 4

**Summary:**

This paper studies the notions of local popularity and local stability in friends-enemies-neutrals (FEN) Hedonic Games (HGs) and possible applications to clustering. FEN games are additively separable HGs in which every agent divides the others into friends enemies and neutrals, and assigns values $\alpha_f, \alpha_e$ to friends and enemies and 0 to neutrals. These games can be represented via a friendship digraph and an enmity one and they are symmetric if the relations are symmetric. Local popularity means that there is no deviation of a single agent s.t. the number of agents gaining from that is strictly more than the rest. Local stability (which was already introduced) requires a deviation to be a Nash one. The authors study 2 simple dynamics LocStab and LocPop and show convergence in the symmetric case. They show convergence of LocStab also in the asymmetric case and then they show possible non-existence and NP-hardness results for the different classes of FEN games considered. They then perform clustering experiments using the mentioned procedures and compare them with usual clustering algorithms.

**Questions:**

I have listed my concerns above. I am curious on how the authors chose the values $(f,e)$ for the experiments and if they have any explanation for the reasons why some parameters work better than others.

**Ethical Concerns:**

["NO or VERY MINOR ethics concerns only"]

**Limitations:**

yes

**Paper Formatting Concerns:**

No formatting concern.

**Quality:**

2

**Strengths And Weaknesses:**

The paper is generally well written and easy to follow.  The results obtained by the authors seem sound and the hardness results non-trivial. I personally am not particularly surprised by the convergence of the simple dynamics in the symmetric games: the very definitions of local stability and popularity seem amenable to the use of a potential argument.

 My main concern about the paper, however, is that, from my perspective, it is hardly about clustering. This is a paper introducing a slightly new concept of stability (local popularity) and studying it along with local stability, in a specific class of Hedonic Games. The relation between coalitional/Hedonic games and clustering is not new (for instance see Hedonic Clustering Games by Feldman Lewin-Eytan and Naor, which is not in the references of the work) and the authors really deal with clustering only in the experimental results. There is no formal comparison between the results of other clustering procedures and the one presented in the paper. This way, I am unsure if I should evaluate this paper as a HGs one or a clustering one.
All the comments above motivate my score.

---

> ### Author Rebuttal · Authors · 2025-07-30
>
> Thank you very much for your helpful comments and questions, which we answer below.
>
> *******
> **Comment**: My main concern about the paper, however, is that, from my perspective, it is hardly about clustering. We also agree that multidisciplinarity and fundamental connections may be oversold in the paper. This is a paper introducing a slightly new concept of stability (local popularity) and studying it along with local stability, in a specific class of Hedonic Games. The relation between coalitional/Hedonic games and clustering is not new (for instance see Hedonic Clustering Games by Feldman Lewin-Eytan and Naor, which is not in the references of the work) and the authors really deal with clustering only in the experimental results. There is no formal comparison between the results of other clustering procedures and the one presented in the paper. This way, I am unsure if I should evaluate this paper as a HGs one or a clustering one. All the comments above motivate my score.
>
> **Answer**: We agree that the paper focuses more on hedonic games than clustering.
> We thank the reviewer for leading us to the paper by Feldman, Lewin-Eytan, and Naor. However, Feldman et al. take a very different route from ours (see a more detailed comparison below), and so we believe that our submission is the first work that shows the applicability of hedonic games and social choice concepts to general clustering tasks. To the best of our knowledge, there is no previous paper that provides a method for clustering stemming from social choice theory with the following two traits:
>
> 1. Relies on a concept that is intuitive, always exists in important, nontrivial cases, is efficiently computable, and flexible enough to be generalizable to different kinds of clustering tasks.
> Note further that our algorithm is polynomial even for _asymmetric_ preferences (in LocStab, balanced case), whereas all known best response dynamics convergence results so far seem to depend on _symmetric_ preferences. This is another useful flexibility factor.
>
> 2. Provides large-scale simulation studies, not just for instances where the solution concepts are most natural, but for very distinct and seemingly unrelated tasks like classifying cancerous images, observing impressive results in all cases.
>
> We explain the similarities and dissimilarities with the paper by Feldman et al. in detail below:
>
>
> 1. The central contribution of our paper is a novel clustering framework rooted in _cooperative_ game theory, whereas Feldman et al. adopt a _noncooperative_ game-theoretic approach centered around Nash equilibria.
>
> 2. Feldman et al. analyze the **price of anarchy** and **price of stability**, quantifying the efficiency loss due to selfish agent behavior in equilibrium. In contrast, we focus on _cooperative_ fairness-driven solutions derived from social choice theory.
>
> 3. Their closest model to ours is ``correlation clustering'' where the players' utilities depend on the similarities with other agents in their own cluster and the dissimilarities with agents from other clusters. Here, this similarity is measured by a real number between 0 and 1. Both this and our framework are special cases of additively separable hedonic games.
>
> 4. A major limitation of the Nash equilibrium approach is its restrictive applicability: existence and convergence are guaranteed only in special metric spaces (e.g., agents on a line or a tree). Our framework, by contrast, operates over arbitrary metrics and can be applied to a much broader class of clustering scenarios.
>
> 5. While Feldman et al. establish theoretical existence and convergence results, they do not provide practical or efficient algorithms. For the closest case, the correlation clustering case, Feldman et al. show that finding a Nash equilibrium is computationally hard (namely, PLS-complete), which is a serious bottleneck if it were to be used for larger clustering tasks. Our work however, does not have this issue, as we design algorithms that are both efficient and scalable.
>
> 6. Unlike our work, Feldman et al. do not benchmark their method against existing clustering algorithms, nor do they claim general applicability to standard clustering tasks. They do not discuss how Nash equilibria are expected to perform according to traditional metrics. One of the key takeaways from our paper is that, despite originating in cooperative game theory and social choice theory, our framework performs comparably to---or sometimes even better than---classical clustering algorithms on standard datasets.
>
> 7. Both models allow agents to express preferences over cluster memberships. However, in the paper by Feldman et al., agents move unilaterally between clusters based on individual utility, whereas in our setting, cluster changes must be _collectively_ acceptable, reflecting a more community-aware and more global notion of stability.
>   That is the great advantage offered by _cooperative_ game theory.
>
> 8. A distinguishing feature of our model is the use of **FEN games**, which reduces the complexity of utility representation by classifying relationships into discrete categories: friend, enemy, or neutral. This abstraction simplifies the input space without compromising the effectiveness of the resulting clusterings.
>
>
> We hope that this detailed comparison clarifies the complementary nature of the two approaches and highlights the distinct contributions of our work.
>
> *****
> **Question**: I have listed my concerns above. I am curious on how the authors chose the values $(f,e)$ for the experiments and if they have any explanation for the reasons why some parameters work better than others.
>
> **Answer**: We chose these based on preliminary experimenting, searching for parameters that seemed most robust across all datasets simultaneously. We think that, intuitively, if the $f$ and $e$ values are close to the average ratio between the desired cluster size and total number of points, then our algorithms perform better; however, this is a mere conjecture we have not yet verified.

---

> > ### Comment · Reviewer_mtec · 2025-08-06
> >
> > I thank the authors for their response and clarifications. My point in naming that reference was intended as a suggestion to expand the related works section and the relative discussion in the paper. I see the differences between the two works! I also think that the paper could benefit by highlighting the connections to clustering and expanding the experimental part.

---

> > > ### Author Response · Authors · 2025-08-06
> > >
> > > Thank you very much! We will seek to implement your helpful suggestions as far as space permits. We plan to expand the experimental part for the journal version.

---

### Official Review · Reviewer_NTLN · 2025-07-02

**Clarity:** 3
**Significance:** 3
**Originality:** 4
**Rating:** 5
**Confidence:** 4

**Summary:**

This paper approaches clustering problems as the problem of finding clusters by setting up a "friends-and-enemies" game where the objects to be clustered become the agents, who want to join friends and avoid enemies.
When used to cluster objects that are given by pairwise distances, two objects are declared friends when their distance is below some fixed fraction $f\in [0,1]$ of the largest distance $D$ between the objects, while they are declared as enemies when their distance is greater than $e D$ for some value of $f<e\in [0,1]$.
Two clustering objectives are defined: One is to find a partitioning of the agents (the common node set of the friends and enemies graphs) such that the partition is *locally popular*, the other is that the partition is *locally stable*. Here we call a partition locally stable if no agent can join a group or create its own group in a way that every agent would prefer this switch. Further, we call a partition locally stable if there is no agent that could join a group or create its own group such that the agent would prefer this switch, while everyone else also prefers this switch. Here, the degree of preference of an agent towards a group is $\alpha \cdot f - \beta \cdot e$ where $f$ ($e$) is the number of friends (respectively, enemies) the agent has in the considered group and $\alpha,\beta>0$ are fixed parameters.

In terms of the choice of $(\alpha,\beta)$, three cases are considered: With $N$ agents, these are $(N,-1)$ ("appreciation of friends"), $(1,-N)$ ("aversion of enemies") and $(1,1)$ ("balanced"). In addition, both symmetric and asymmetric graphs are considered.

The main technical contributions are complexity results (algorithms when appropriate) for the existence of a locally stable/popular partitions for the 3x2=6 types of "games". In short, for symmetric graphs these problems are in P (and simple local heuristics solve the associated search problems, efficiently), for asymmetric graphs, only the problem corresponding to the balanced case is in P. The remaining problems are proven to be NP-complete.

Experimental results corroborate the theory, where the two local search methods are compared with some standard methods (k-means, DBSCAN). The new methods appear to be reasonable.

**Questions:**

Can we motivate the approach from "first principles"? Or at least relate to other "first principles" approaches to clustering.

**Ethical Concerns:**

["NO or VERY MINOR ethics concerns only"]

**Final Justification:**

I maintain this is a fine paper, that requires minor work only which the authors are willing to do.
I remain positive.

**Limitations:**

Yes

**Quality:**

4

**Strengths And Weaknesses:**

Strengths:

1. The approach is intuitive; I like that there is a clear (intuitive?) definition of the objective for finding clusterings.
2. The paper feels quite complete; I appreciate the work that went into developing these ideas. It is appealing that the local search methods provably work in the "nice" cases.
3. The text is clear. No issues with clarity.

Weaknesses:

1. While there is an intuitive appeal to the approach, overall, as typical in clustering, it is unclear what are the settings where this approach is expected to be "good".
2. While there is no issue with clarity, things feel a bit more complicated than what they need to be. There is quite a bit of terminology that is not necessary and is not helping the readers. A more straightforward approach to presenting the results could be better. Relatedly, the abstract promises that "fundamental connections" between coalition formation games and clustering will be studied. This is not true. What happens is that ideas from coalition formation games are used to derive clustering algorithms. To study connections one would need some kind of definitions concerned with clustering objectives, that one could perhaps relate to coalition formation. I think the multidisplinarity is also oversold. Similarly, I could not relate the last sentence of the abstract to the paper.

---

> ### Author Rebuttal · Authors · 2025-07-30
>
> Thank you very much for your helpful comments and questions, which we answer below.
>
> *******
> **Comment**: While there is an intuitive appeal to the approach, overall, as typical in clustering, it is unclear what are the settings where this approach is expected to be ``good''.
>
> **Answer**: We believe that settings where data points are agents and fairness is important are the main applications, such as detecting communities in social networks, crowdsourcing, etc. We will emphasize possible applications in more detail.
>
> Furthermore, our simulations suggest that our algorithms perform best when the expected cluster sizes are similar.
> ********
> **Comment**: Relatedly, the abstract promises that ``fundamental connections'' between coalition formation games and clustering will be studied. This is not true. What happens is that ideas from coalition formation games are used to derive clustering algorithms. To study connections one would need some kind of definitions concerned with clustering objectives, that one could perhaps relate to coalition formation. I think the multidisciplinarity is also oversold. Similarly, I could not relate the last sentence of the abstract to the paper.
>
> **Answer**:
> We agree that we may have oversold the interdisciplinary nature of our work, as we indeed have not unearthed any strong bidirectional formal connections between the two fields. The fundamental connection is more on the intuitive level---between our solution concepts in hedonic games and the classic quality measures in clustering---and not on the conceptual level of definitions or theorems. The purpose of the paper (besides our strong theoretical interest in FEN games) was to try to see whether this underlying intuitive connection can lead to efficient clustering algorithms that rely on solution concepts from hedonic games but are useful in clustering too. In view of our experiments, this seems to be the case.
>
> Regarding the admittedly too much exaggerating last sentence of the abstract, we are happy to delete it.
>
> **********
> **Question**: Can we motivate the approach from "first principles''? Or at least relate to other "first principles'' approaches to clustering.
>
> **Answer**:
> We think that our approach cannot be formulated as a ``first principles'' approach to clustering, since it requires the construction of the friendship/enmity graph over the data points. Hence, this part of our approach is inherently not data-driven. However, it is not impossible to conceive some method which sets the similarity/dissimilarity thresholds that define friendships and enmities via some data-driven, perhaps iterative, process.
> Nevertheless, in its current form, our method does not work from first principles.

---

> > ### Comment · Reviewer_NTLN · 2025-08-09
> >
> > First principles: What I meant is some sort of axioms that lead to the clustering approach considered here.
> > Other than that, we are on the same page!

---

### Decision · Program_Chairs · 2025-09-17

**Decision:**

Accept (spotlight)

**Comment:**

The paper proposes a well defined and quite intuitive definition of an objective for clustering. It then provides a thorough investigatio of the computational complexity of optimizing this objective in several cases. The writeup is clear and teh results are clean. The main criticism the reviewers express is that the paper is oversold, bith in terms of how fundamental the proposed objective is and in terms of the applicability of the results. In their rebuttals, the authors promis to address these issues in tehir final version.